# The response of culturally important plants to experimental warming and clipping in Pakistan Himalayas

**Saira Karimi[1], Muhammad Ali Nawaz[2], Saadia Naseem[1], Ahmed Akrem[3], Hussain Ali[4], Olivier Dangles[5], Zahid Ali[1] ***

**1** Department of Biosciences, Plant Biotechnology & Molecular Pharming Lab, COMSATS University Islamabad (CUI), Islamabad, Pakistan, **2** Department of Biological and Environmental Sciences, Qatar University, Doha, Qatar, **3** Department of Botany, Institute of Pure and Applied Biology, Bahauddin Zakariya University, Multan, Pakistan, **4** Department of Zoology, Quaid-i-Azam University, Islamabad, Pakistan, **5** CEFE, Univ Montpellier, CNRS, EPHE, IRD, Univ Paul Valéry Montpellier 3, Montpellier, France

* zahidali@comsats.edu.pk

**Data Availability Statement:** All relevant data are within the paper and its Supporting Information files.

## Abstract

The relative effects of climate warming with grazing on medicinally important plants are not fully understood in Hindukush-Himalaya (HKH) region. Therefore, we combined the indigenous knowledge about culturally important therapeutic plants and climate change with experimental warming (open-top chambers) and manual clipping (simulated grazing effect) and compared the relative difference on aboveground biomass and percent cover of plant species at five alpine meadow sites on an elevation gradient (4696 m-3346 m) from 2016–2018. Experimental warming increased biomass and percent cover throughout the experiment. However, the interactive treatment effect (warming x clipping) was significant on biomass but not on percent cover. These responses were taxa specific. Warming induced an increase of 1 ± 0.6% in *Bistorta officinalis* percent cover while for *Poa alpina* it was 18.7 ± 4.9%. Contrastingly, clipping had a marginally significant effect in reducing the biomass and cover of all plant species. Clipping treatment reduced vegetation cover & biomass by 2.3% and 6.26%, respectively, but that was not significant due to the high variability among taxa response at different sites. It was found that clipping decreased the effects of warming in interactive plots. Thus, warming may increase the availability of therapeutic plants for indigenous people while overgrazing would have deteriorating effects locally. The findings of this research illustrate that vegetation sensitivity to warming and overgrazing is likely to affect man–environment relationships, and traditional knowledge on a regional scale.

## Introduction

Climate change is modifying the structure and function of high elevation ecosystems. Mountains are splendidly diverse ecosystems that hold high proportions of endemic species [1–5]. These systems are exceptionally vulnerable to climate change with most species distribution models projecting drastic changes in community composition and distribution range [6].

**Funding:** Snow Leopard Foundation (SLF) and Forest & Wildlife Department of Gilgit-Baltistan, Pakistan provided financial support to Muhammad Ali Nawaz and Saira Karimi to carry out surveys and data collection from KNP. For collection of medicinal plants from KNP for bio-exatracts and their functional profiling, financial support was taken from Pakistan Sciences Foundation, PSF-NSLP Project No. 663. Higher Education Commission (HEC) Pakistan provided scholarship to Ms. Saira Karimi under the umbrella of HEC-IRSIP program for 6 months study abroad.

**Competing interests:** Authors declare that there is no competing professional or financial conflict of interests. We confirm that the manuscript has been read and approved by all named authors and that there are no other persons who satisfied the criteria for authorship but are not listed. Moreover, we confirm that the order of author names has been approved by all of us.

Relevantly, high altitude regions are already under the stress of land and water degradation, climate change exerts extra pressure and makes them highly exposed [7]. These stressors have intense effects on the habitability of the mountainous ecosystem and the human communities that depend on them (IPCC 2019), in the inter-tropical region (Himalayas, Andes, Eastern Africa). In the Himalayas, an increase of 3.7°C is expected in mean surface temperatures between 2018–2100 (relative to 1986–2005). That is expected to cause substantial effects on the water cycle, biodiversity, and livelihood of the local population [8]. This concern is reinforced by increasing evidence that the rate of warming is magnified with elevation, ecosystems at high altitudes (> 4000 m) showing more rapid changes in temperature than those at lower elevation [4]. In addition to the warming of mean temperatures, there is often a greater increase in the daily temperature variations [9] with potential effects on wildlife and humans.

Overgrazing disturbs plant species composition, biomass, and other vegetation characteristics [10]. Repeated grazing can initiate exploitation and reduction of highly palatable species which are less resistant to severe grazing. This leads to changes in plant availability and nutrient content for grazing animals' feed and forages in natural ecosystems. Hence, some plants are selected repeatedly as forage and cause exploitation by overgrazing. It may cause a significant reduction in the species and attribute to selective grazing of highly palatable species that are intolerant to severe grazing and flattening [11, 12]. It may lead to selective palatability issues for grazing animals and changes in nutrient availability [13]. At higher elevations, the climatic zone does not permit tillage crop farming. Therefore, animals provide food to an increasingly dense human population in this area of Pakistan by transforming grass and herbage into milk and meat [5]. In recent years; larger herds of cows, horses, and sheep have been introduced in many high-altitude regions in the entire world [14, 15], which previously did not support large grazing herds [16, 17]. Changes in the agricultural system by the introduction of new crops and the replacement of native species of grazers by species from other environments could be complex in the removal of certain endogenous plants as well as overall plant cover reduction.

Warming and grazing both have substantial impacts on the natural elements of high-altitude ecosystems, with critical consequences for local livelihoods [18]. In cold habitats such as the high Himalayas, leaves are generally rich in nutrients (high N content) to circumvent the detrimental effects of the adverse environment on the overall physiology of plants [19]. These attributes and specific characteristics are making them more desirable and palatable for grazers [20]. Other than grazing, plant species are ubiquitously utilized by indigenous communities as traditional medicines as well. The therapeutic effect of some mountain plant species is widely accepted and utilized in many well-known regions [14, 21–24], yet detailed ethnobotanical studies are scarce [25].

Ecosystem change assessment implies measuring how and to what extent a change can affect people's life [26]. While a great deal of research has been conducted to predict warming and grazing effects on alpine biodiversity [20], studies on medicinal plants are still lacking and indigenous knowledge is marginalized, though mountainous communities still rely on the traditional system of medicines for medical treatments [22]. The incorporation of indigenous knowledge of medicinal plants with the current biological conservational practices is necessary to assess the future face of biodiversity and associated threats [27–29]. To address these shortcomings, here, we provide an example of integrating plant community responses to grazing and warming with ecological and socioeconomics characteristics to assess how people's approach to culturally important plant species may be affected in the future.

The HKKH (Hindukush-Karakoram-Himalaya) region harbors a rich indigenous knowledge that serves as a source of sustainable rangeland management [27]. The region is also home to diverse cultures, languages, traditional wisdom, and religions. The significant aspect

of livelihood in HKH (Hindukush-Himalaya) is inherent to mountain specificities (limited accessibility, unique plant species, greater fragility socioeconomic inequalities, indigenous knowledge, and vulnerability). It is home to native people who are most marginalized socially and economically and vulnerable to ongoing environmental changes [22, 30] in particular global warming scenarios [1, 31–35]. Traditional knowledge may help local people to find solutions, helping them to cope with impending changes [32, 36, 37].

For indigenous communities, the medicinal plants of the Eastern Himalaya are an invaluable resource for herbal remedies, income, and forage for the local communities in Pakistan [38]. In the recent scenario of climate change, average temperatures have increased by 1.5˚C, which is more than twice the global average increase articulated by World Economic Forum 2020 [39]. The mountainous areas of Khunjerab National Park (KNP) are extensively used as feeding grounds for both wild ungulates (yaks) and domestic animals (goats, sheep, and cattle) with most families keep mixed herds [40]. Merging traditional knowledge with manipulative experiments is a helpful approach to test the sensitivity of such ecosystems to climate change [19]. Therefore, the specific objectives of this study were to 1) investigate culturally and medicinally important plant communities' responses to climate warming and clipping (simulated grazing) by manipulative experiments, and 2) to link these ecological factors to the socio-economic wellbeing of local inhabitants by assessing the dependence and utilization of these resources through ethnobotanical surveys.

## Materials and methods

### Study area

The current study was conducted at the Khunjerab National Park (KNP) which is situated in the HKKH (Hindukush-Karakoram-Himalaya) mountain ranges near the border with China (36.37˚ N, 74.41˚E). The KNP is spread over an area of 4455 km$^2$ with altitudes ranging from 2439 m to 4878 m above sea level. [41, 42]. The study was established in the alpine zone where the climate is defined by warm summers starting from May till August. The maximum temperature in May recorded in the warm year goes up to 25˚C [43] while in winter it drops down below 0˚C (up to –10˚C) from October at 3346 m [17, 44, 45]. Valleys of KNP are characterized by stony beds, water channels (Nullah), vegetation with grasses, and large plants like bushes along slopes. Khunjerab is covered with grasses and has green pastures. Genus *Artemisia* and *Poa* exist here predominantly. The current state of knowledge is based on the analyzed climatic trends in HKKH region from data available from 1980 to 2009 [46], which indicates the rapid declines in the glaciers of the Great Himalaya which attributes to global warming [47]. The reports of the weather station at Gilgit, illustrate the significant increase in mean annual temperature (average 0.65˚C) which is higher than the global average (0.17˚C) during the last 35 years [48].

The alpine KNP region includes the following vegetation altitudinal zones: 1) The *sub nival zone (> 4500 m)* is made of snow and bare desert, covering about 30% of the national park area. Characteristic plant species are *Saussurea simpsoniana*, *Primula macrophylla*, *Oxytropis macrophylla*, *Potentilla multifida*, and *Hedinia tibetica* [49]. 2) *Alpine meadows (3500–4500 m)* are rich in herb biomass and therefore serve as an important habitat for livestock (sheep, goats, cattle, and yaks) and wild herbivores such as ibex (*Capra ibex sibrica)*, golden marmot (*Marmota caudata aurea*) and Marco Polo sheep (*Ovis ammon polii*). Dominant plant taxa are *Primulla macrophylla*, *Plantago lanceolata*, *Saxifraga* spp., *Potentilla hololeuca*, *Poa alpina*, and *Carex* spp. 3) The *sub alpine Steppe (< 3500 m)* is vegetated mainly with *Artemisia* and *Primula* plant genera. Some grasses such as *Poa* and *Carex* spp. are also found in relatively moist places [50].

The summer pastures of KNP are subjected to grazing management systems by indigenous communities because they are rich in herbaceous biomass [5]. In 2013, an analysis conducted to document the changing aspects of above ground biomass (AGB) in high altitude rangelands of Pakistan, proclaimed the number and types of livestock in KNP pastures featuring more than 5000 individuals including sheep, goat cattle yak, and others (horses & donkeys) [51].

## Ethnobotanical survey

We formulated questionnaires (S5 Table) to elicit facts about how accessible and/or useful medicinal plant species for local people in the region. We visited three distinct valleys of KNP (Khunjerab, Ghunjerab, Shimshal) and negotiated with local community coordinators to enlist the people, interested in the survey. Plant species particular to the family *Asteraceae*, *Brassicaceae*, and *Elaeagnaceae* were cited considerably. Based on this initial screening survey, sites for manipulative experiments were selected where these species were in abundance as excerpted by the informants. Questions were asked and data were gathered about local medicinal plants, their traditional uses, methods of crude drug preparation, and climate change responses, etc. Socio-economic and demographic data about interviewed people are given in the S2 Table. Partially structured interviews were formulated to collect information about key medicinal properties of plants, frequency of use, and the perceived impacts of climate change on their life cycle, growing period, and occurrence.

In this specific study, we involved local inhabitants living in the area to inquire about indigenous plant species. The consent of agreement was taken verbally by 1st and 2nd authors. The work has been always performed with ethical oversight by an ethics committee and was reviewed and approved before the start of the work, by Institutional Ethics Review Board (No. CIIT/BIO/ERB/17/53) and Institutional Biosafety Committee (No. CIIT/BIO/ERB/17/04) of COMSATS University Islamabad (former COMSATS Institute of Information Technology (CIIT) Islamabad.

## Warming and clipping experiments

Based on this initial screening survey, sites for manipulative experiments were designated comprised of the predominant species as excerpted by informants and a three-year experiment was established in 2015 and visited annually for maintenance and collection of data. The experiments were designed in a randomized block design (RBD) and involved the installation of a manipulative experimental at five different sites along an altitudinal gradient ranging from 3590 to 4696 m, which lies in the alpine meadow vegetation zone of KNP. Each site consisted of a fenced area (un-grazed) of 5 x 5 m and laid out in four subplots of 2.5 x 2.5 m (Fig 1). We used standard cattle 1–2 wire fencing (1.7 m height with 15 x 20 cm mesh size). The experimental sites included summer pastures of alpine meadows. After fencing, the vegetation was observed in both fenced sites and surrounding open sites subjected to continuous grazing during the peak season. The herbaceous summer pastures of KNP are attributed to a grazing management system adopted by the local community that allows them to bring their livestock for grazing during peak season which usually starts from late March and lasts till August. A summary of the number and type of grazing animals in selected pastures of KNP was reported earlier making the sheep, yok, goats, and cattle the potential grazers. [51].

Inside the fenced area, we selected two subplots for warming treatment using hexagonal fiberglass open-top chambers (OTC1 & OTC2) (Fig 1). OTC1 plots served as only warming treatments while clipping was done from one of the OTCs (inside 1m x 1m area) which corresponded to warming x clipping (W x CL) treatment. They were hexagonal with a 32 cm height a central portion of 1 x 1 m open at the top [52]. The OTCs were kept on the subplots in each

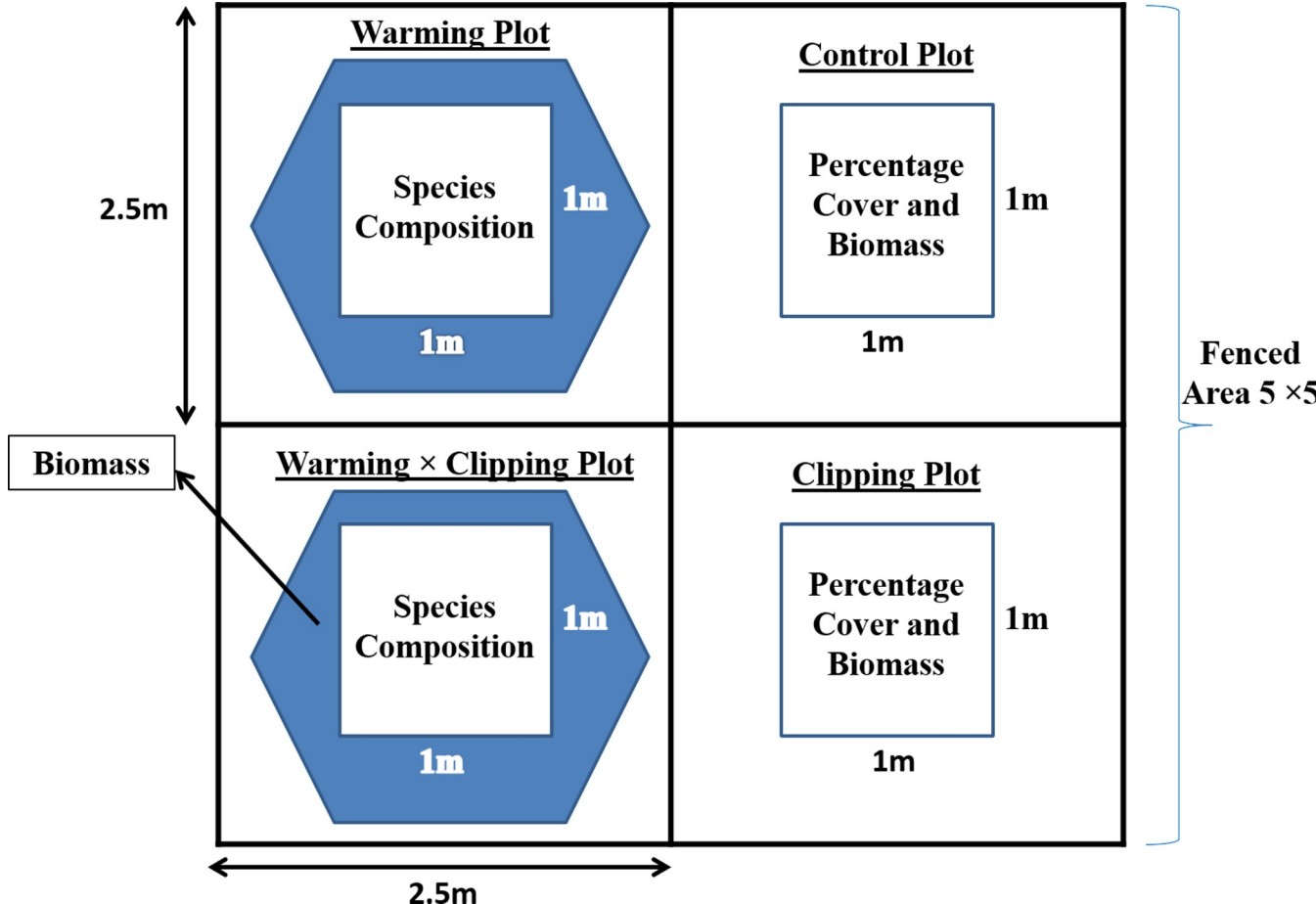

**Fig 1. Schematic representation of a typical exclosure with open-top warming chambers (OTC) for passive heating, manual clipping plot for simulated grazing and control plots.**

summer season. In general, the fenced area with OTCs acted as an exclosure and grazing barrier for animals, however, it did not eliminate small rodents and birds. But their effect on the plant destruction was not that significant because not only OTCs but as well as control and clipping plots were also subjected to their infringement.

The warming effect of OTC at 10 cm above the soil surface was between 1.7–1.9˚C on an average [53, 54]. This temperature enhancement is within the range of expected degrees of warming in the recent scenario of future climate change in the region [55].

## Clipping treatment

The experiment consisted of two treatments: one 2.5m$^2$ subplot inside exclosure subjected to clipping treatment, one similar dimensioned plot outside the fenced area as a reference to estimate the herbivory level on plant species to mimic the grazing and their control inside exclosure (Fig 1). The warming x clipping (W x CL) treatment was employed by clipping the 1m$^2$ area inside OTC2. The clipping was done one time each year to effectively simulate herbivory. The simulated grazing experiment involved: annual clipping plot inside the exclosure where the manual clipping of vegetation was done to the average height of vegetation outside, its control, and the area that is grazed by herbivores outside the fence and (un-warmed, un-grazed). At the start of the experiment, the grazing level of herbivores was estimated and clipped inside

the exclosure in clipping treatment, then each year plant growth was recorded from that clipped point. In the clipped plots the aboveground part of all selected plant species was clipped to about 2–4 cm on an average above the entire plot. There was no disruption in the unclipped plots (OTC, control).

A 1x1m quadrat was placed randomly within each plot. For clipping plots, this location is fixed for all sampling years and permanently marked using pegs. For OTCs, biomass estimation was done under one of the six sides on a rotational basis. Species compositions indicate the area where species were identified, and percent cover was estimated.

## Data sampling

In the present study, all recordings of the plant community composition and abundance were performed during the peak blooming season which starts from late March and lasts until late August [17]. During the experimental years (2016–2018), we used the quadrat sampling method (1m x 1m) [56] to measure plant metrics such as percentage cover, and aboveground biomass. The percentage cover of the selected plant species was recorded individually at the start of the experiment (2016) by visualizing the numbers of grids of quadrat covered by plant species.

The plant species were selected based on their medicinal and cultural potential mentioned by informants, citation frequency, occurrence in alpine meadows, and the palatability as described by local inhabitants (Table 1). In each 1 x 1 m subplot, the percent cover for each plant species was estimated to the nearest 10% for each species rooted inside the plot (1 x 1 m %%, 10cm grid points) using the Daubenmire method [57]. Implementing this method, we measured the cover; defined as vertical projections of vegetation that include the area of a quadrat. In all experimental sites, individual plant species were identified and sampled carefully for taxonomic classification. Inventory was updated each year to record the new observations (2016–2018). We measured the aboveground biomass of each species in each treatment by clipping method and weighed it in the laboratory by air drying at 37°C for 72 hours.

The percent cover and biomass were averaged across all the experimental years and the % increase and decrease were calculated by working out the difference between control and last year data points.

## Statistical analysis

The composed ethnomedicinal data was quantitatively reviewed by the index of Relative Frequency Citation (RFC), calculated as the ratio between the number of informants mentioning the use of a given species and the total number of informants participating in the survey [50]. The species-specific responses within each treatment at each site and their significance were estimated through one-way ANOVA (S3a Table). First linear models were selected to analyze if there was a difference in species, site, and treatment effects (S3B Table, S2 Fig) on the percentage cover and aboveground biomass. The individual effect of each factor was also determined (S3C and S3D Table) to record the significant increase or decrease in cover and biomass.

Response variables of percentage cover and biomass were modeled using lme4 package [58] with fixed-effect predictors of the warming treatment, the clipping treatment, and their two-way interaction(warming x clipping, W x CL), and random effects for site and species (Table 2). We separately ran models with species as a fixed effect to check the effect of treatment for each species S4 Table. The following model formulation was used.

*Perc cover ~ warming + clipping + warming* clipping + (1|site) + (1|species) (Table 2)*
*Perc cover ~Factor*species: (1|site)          (S4 Table)*
*\*Here the factors are the treatments, (warming, clipping and warming x clipping). The model formulation was to check the individual response or species-specific response to each treatment.*

**Table 1. Ecological and ethnobotanical properties of selected medicinal plant species.**

| Plant species and authority | Family | Local name | Altitudinal range (m) | Type | Part used | Medicinal properties | Climate change effect (according to local people) | Citation Frequency (FC%) | RFC* |
|---|---|---|---|---|---|---|---|---|---|
| | | | | | | | | | FC/N (0<RFC<1) N = No. of informants) |
| *Artemisia santolinifolia Turcz.ex Krasch* | Asteraceae | *Roon* | 3343–4039 | Perennial | Whole plant | Worm, digestion, diarrhea, malaria | Increased availability due to extended growth period | 75 | 0.9375 |
| **Artemisia rupestris* L.* | Asteraceae | *Khich* | 4059–4676 | Perennial | Whole plant | Digestion, insect repellent elevates immunity, skin diseases | Late-blooming | 60 | 0.9375 |
| **Taraxacum afficinale* L.* | Asteraceae | *Yamook* | 3346–4059 | Perennial | Stems, leaves, dried twigs | Asthma, cardiac diseases | No effect | 35 | 0.4375 |
| *Smelowskia calycina (Steph.)* | Brassicaceae | *Zakh* | 4096 | Perennial | Flower, leaves | Blood pressure troubles and diabetes | Berries are not that juicy | 15 | 0.1875 |
| *Silene gonosperma (Rupr.) Bocquet* | Caryophyllaceae | *Gulch* | 3359–4693 | Perennial | Whole plant | Fever | Vegetation increased. | 17 | 0.2125 |
| **Astragalus penduncularis Royle ex Benth* | Fabaceae | *Zhoop* | 3346–4696 | Perennial | Leaves and Flower | Diabetes, antiaging, anti-inflammatory | Vegetation increased | 23 | 0.2875 |
| *Oxytropis glabra DC* | Fabaceae | *Zarth sprag* | 4034 | Perennial | Leaves | Internal wound, anti-aging | No effect | 41 | 0.5125 |
| **Comastoma pulmonarium (Turcz) Toyok* | Gentianaceae | *Shalay Char* | 4696 | Perennial | Leaves and flower | Pneumonia, sore throat, and fever. | No effect | 33 | 0.4125 |
| *Peganum harmala L.* | Nitrariaceae | *Ispandur* | 3343 | Perennial | Seeds | Fever and joint pain | No effect | 8 | 0.1 |
| *Plantago major L* | Plantaginaceae | *Sepgilk* | 4059 | Perennial | Leaves and Seeds | Ulcers, skin wounds &dysentery | Ripe late | 38 | 0.475 |
| **Poa alpina L.* | Poaceae | *Noz* | 3346–4076 | Perennial | Whole plant | ***Indigestion*** | Vegetation increased | 45 | 0.5625 |
| *Bistorta officinalis* | Polygonaceae | *Onbu* | 4690 | Perennial | Root | Wounds | Less vegetation | 8 | 0.1 |
| **Primula macrophylla D.Don* | Primulaceae | *Banafsha* | 4676 | Perennial | Leaves and Flower | Alleviate pain, analgesic | Forage available | 9 | 0.175 |
| **Potentilla hololeuca (Boiss)Hook.f* | Rosacea | *Zatsprig* | 3343–4696 | Perennial | Leaves and Flower | Menstrual cramps | Increased vegetation | 29 | 0.3625 |
| *Saxifraga sp* | Saxifragaceae | *Sitbark* | 4693 | Perennial | Leaves | Diarrhea, and minor skin problems | No effect | 12 | 0.15 |
| *Pedicularis kashmiriana Pennell* | Scrophulariaceae | *Push* | 4690 | Perennial | Whole plant | Stomach aches | Shifted towards high altitude. | 14 | 0.1125 |
| *Hedinia tibetica (Thomson) Oestenf* | Scrophulariaceae | *Tibet* | 4690 | Perennial | Whole plant | Pain and swelling, muscle weakness | Distributed at all sites. | 13 | 0.1625 |

(*) represents the most preferable forage species as mentioned local names by informants during the survey, RFC: Relative frequency citation indicates *Artemisia species* having a relatively high frequency of citation among other species. The frequency is expressed as the percentage of a plant is mentioned by the informants.

**Table 2. Summary statistics of linear mixed model to determine the fixed effects of treatments and random effects of site & species.**

| Predictors | Percentage Cover | | | | Above Ground Biomass | | | |
|---|---|---|---|---|---|---|---|---|
| | Estimates | CI | t-value | p-value | Estimates | CI | t-value | p-value |
| (Intercept) | 7.78 | 4.25 – 11.31 | 4.572 | <0.001*** | 2.71 | 1.57 – 3.86 | 4.631 | <0.001*** |
| Factor [Warming] | 7.86 | 5.16 – 10.56 | 5.618 | <0.001*** | 3.64 | 2.46 – 4.83 | 6.035 | <0.001*** |
| Factor [Clipping] | 1.78 | -0.93 – 4.48 | 1.170 | 0.242 | -1.68 | -2.87 – -0.50 | -2.789 | 0.005** |
| Factor [Warm*Clip] | -0.32 | -3.02 – 2.39 | -0.381 | 0.718 | -1.56 | -2.75 – -0.38 | -2.587 | 0.037* |
| **Random Effects** | **Percentage Cover** | | | | **Above Ground Biomass** | | | |
| $\sigma^2$ | 80.83 | | | | 15.40 | | | |
| $\tau_{00}$ Species | 17.68 | | | | 2.38 | | | |
| $\tau_{00}$ Site | 6.26 | | | | 0.10 | | | |
| ICC | 0.23 | | | | 0.14 | | | |
| N Site | 5 | | | | 5 | | | |
| N Species | 17 | | | | 17 | | | |
| Observations | 340 | | | | 340 | | | |
| **Marginal R$^2$ / Conditional R$^2$** | **0.094 / 0.301** | | | | **0.207 / 0.317** | | | |

Significance codes: 0 '***' 0.001 '**' 0.01 '*' 0.05 '.' 0.1 ' ' 1.

*To estimate the fixed effects of the treatments, we used linear mixed effect models using lme4 package (Bates, Mächler, Bolker, & Walker, 2015 [58]) in R (The R Foundation, 2018 [59]). The model summary provides the estimation, standard error, and t values of each of the factors as shown in Table 2. It is shown that inferences for the fixed-effects parameters in linear mixed models are based on t-value distributions with suitably adjusted degrees of freedom. In mixed effect models, the main rationale is that while p-values can be misleading because of unclear null distribution, t values can still be useful as standardized parameters. The t-test used to evaluate the significance of overall treatments, the variance between the treatments and the response of each species to the specific treatment.*

High ethnomedicinal value (Table 1) and the significant responses to the treatments illustrated by mixed effect models, the vulnerability and susceptibility of important plant species were determined. These species included *Artemisia rupetris*, *Poa alpina*, *Primula macrophylla*, *Potentilla hololeuca* and, *Astragulus penduncularis*. The combined effect of treatments on these species was illustrated as susceptibility zones to show the significant increase or decrease in percent cover and biomass. The significance of the effect of treatments was assessed by linear mixed effect models using lme4 package function lme() [58] in R [59]. The plot was made by ggplot2, function geom_polygon, in package tidyverse.

To assess the change in the relative distribution of plant species within the plant communities, we used non-metric multidimensional scaling (NMDS) ordination analyses [60] on plant species that were present across the plot at each site (3590 m–4696 m) so that we could characterize plant community changes for each warming and clipping treatment groups. The Bray–Curtis dissimilarity measure was used to deal with relative abundance data as it allowed using both presence/absence and abundance data. All the analyses were performed in RStudio 2.4–4. (R Core Team 2017) using vegan packages [61].

## Results

### Ethnobotanical survey

The utmost objective of this study was to gather the indigenous knowledge about high mountainous medicinal plants from the local community and evaluate the climate variation effects

on culturally and medicinally valued plants. We interviewed a total of 80 informants who identified approximately 50 medicinal plant species widely spread in the area (see S1 Table). Based on ethnobotanical survey results, some highly cited species were selected; those species are palatable, essential for therapeutic remedies, and abundant on experimental sites thus prone to the change in the environment. Predominant botanical families comprised of *Asteraceae 25%*, *Fabaceae* 8%, *Solanaceae* 6%, *Tamaricaceae* 4%, and others count for 2% or less (see Table 1). The subsequently selected species were characterized by three or more informants as a general criterion for the reliability of medicinal plant use [62]. Those interviews also allowed to collect detailed information about the therapeutic properties of plant species along with their altitudinal range of occurrence, time of blooming, partly used for remedies, and potential climate change effect on their availability (Table 1). Based on the collected information, we screened out 17 highly important and common plant species, which were categorized based on their percentage use. *Artemisia rupestris* (AR) was the most cited plant due to its availability and medicinal potential. Other widely cited plant genera include *Poa alpina* (PA) and *Oxytropis glabra* (OX), *Taraxacum officinale* (TM), and *Plantago major* (PM).

The frequently reported and treated ailments were high blood pressure, diarrhea, skin allergies (40%), followed by eye irritation, fever, gastrointestinal diseases (25–30%) including stomachache, worm treatment, acidity (<30%), etc. Ailments related to heart, diabetes, joint pains, obesity, and metabolism were also reported (see S1 Fig). A variety of methods were reported by informants to prepare remedies such as i.e., decoction, infusion, tincture, juice, or ground to powder, etc. The dominant parts used in the preparation of medicine were leaves, flowers, and/or the whole plant. The characteristic cultural significance of plant species was recorded as a source of feed and forage to livestock and wild animals. Local communities rely on livestock to generate income. Summer pastures are important, and the mentioned species are frequently available there.

## Effects of warming and clipping on percentage cover and above ground biomass of representative plant species

**Linear mixed effect model for fixed and random effects of variables.** The individual factor impact on the percentage cover and biomass is shown in Fig 2 and Table 2, illustrating the impact of warming (W), clipping (CL), and interactive treatment of both warming x clipping (W x CL) keeping the site as a random effect. There were strong significant and positive effects (*p<0.001*) of factor warming on the overall percentage cover and biomass tested by linear mixed model contrastingly clipping exerted significant negative effect on biomass and non-significant negative effect on percentage cover. Plant species decreased their aboveground biomass in clipping plots while there was a significant increase in biomass in warming plots. This positive effect was overridden by the clipping effect in interaction plots where the combined impact of both warming and clipping was found to be significantly negative on aboveground biomass (*p<0.05*) (Table 2).

A mixed effect model summary showing estimate, standard error, and t statistics is described in Table 2 explaining the significant fixed effects of experimental warming, clipping, and the interaction treatment (warm*clip) on plant species. Marginal $R^2$ value represent variance by fixed effects and conditional $R^2$ explained variance by the entire model (fixed and random). Overall, the effect of warming treatment is positive as compared to clipping and the two-way interaction of both treatments (warm*clip). Species specific response to each treatment while keeping the effect of the site as random showed that elevated warming has significantly increased the percentage cover and aboveground biomass as shown in Fig 2, S4 Table.

Site effect on percentage cover was significant at M1(*4696 m*) where the cover was significantly increased than control plot and at M5(*3346 m*) cover was decreased (S3b, S3C Table, S2A Fig). M1 had higher biomass than control while at M5 there is no significant variation

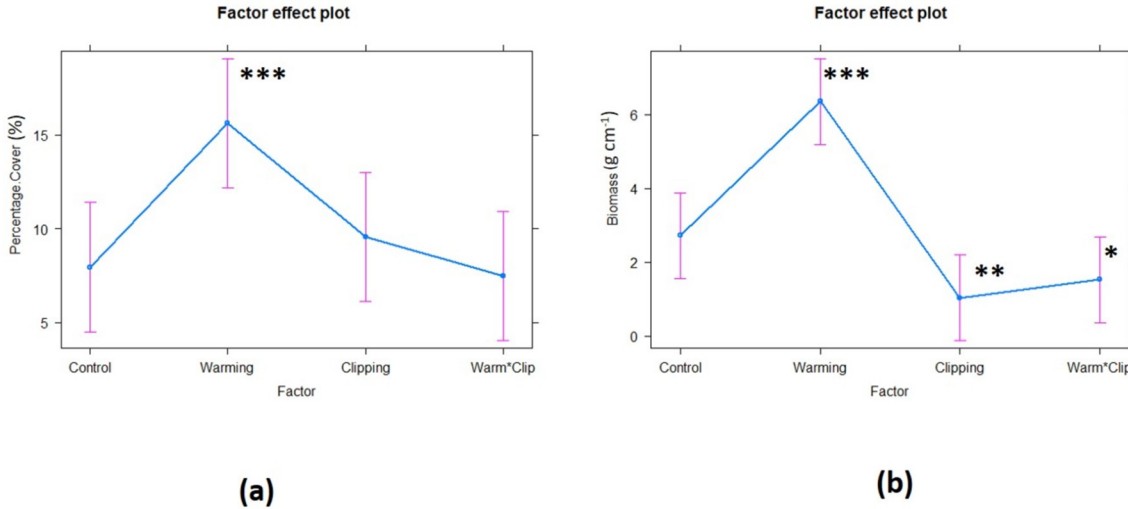

**Fig 2. Linear mixed effect model representation of significant effects of each treatment (fixed effect) on percentage cover and biomass. a)** Percentage cover was higher in warming plots in comparison to control but warming and clipping treatment significantly had less cover of species. **b)** greater biomass of all species in warming treatment in comparison to control and other treatments, suggested that under passive heating, plants tend to increase their vegetative growth. The significant codes '***' 0.001 '**' 0.01 '*' 0.05 '.' 0.1 ' ' 1. *red dot = 0.1.*

between control and treatments (S2 Fig, S3D Table). At other sites (M2,4059 m, M3,4022 m, M4,3990 m) the response of species between control and treatment plots was similar (S2 Fig, S3C and S3D Table). The response variability was high among the highest elevation site (M1) and the lowest (M5) suggesting that sites might be the determining factor for the prediction of certain species' significant responses at different sites. Changes in the species cover and biomass along the elevational gradient as random effect can be described as predictor about the variance in response between different sites.

The overall difference in response magnitude of all species percentage cover and biomass can be seen in the species effect plot where some plant species as *Poa alpina (PA)*, *Astragulus penduncularis (AS)*, *Artemisia rupestris (AR)*, and *Potentilla hololuca (PT)* have significantly higher percent cover in comparison to others while the effect of warming induced the significantly higher above ground biomass which determined its positive effect (S2 Fig, S3A Table).

The total increase in plant cover in response to warming was highly variable among taxa, ranging from $1 \pm 0.6\%$ for *Bistorta officinalis* to $18.7 \pm 4.2\%$ for *Poa alpina* (S3A Table, Fig 3). The plant community in warming plots responded significantly towards warming, while the response magnitude and intensity were highly variable among species. Important species as *Artemisia rupestris*, *Poa alpina*, *Carex divisa*, *Saxifraga*, *Taraxacum officinale*, etc. have higher percentage cover and biomass in warmed plots in comparison to control and other treatment plots. The negative effect of clipping and interactive treatment significantly decreased the aboveground biomass of species such as *Primula macrophylla (PC)*, *Oxytropis glabra*, *Hedinia tibetica*, and *Bistorta officinalis* (Fig 3, S4 Table). Compared to the control plots, clipping treatments had an overall negative effect on plant biomass and percentage cover (S2 Fig, Fig 2). Yet again, this general trend was not significant due to the high variability in plant responses among sites. Clipping has significantly reduced the biomass of some species as shown in Fig 3.

The mixed effect model showed that the interactive effects of warming and clipping were comparatively weaker than the main effects, both on aboveground biomass and percentage cover. Whereas clipping has a consistent negative effect in main effect plots, it decreased the aboveground biomass of the species in interactive plots too. Our finding showed that the

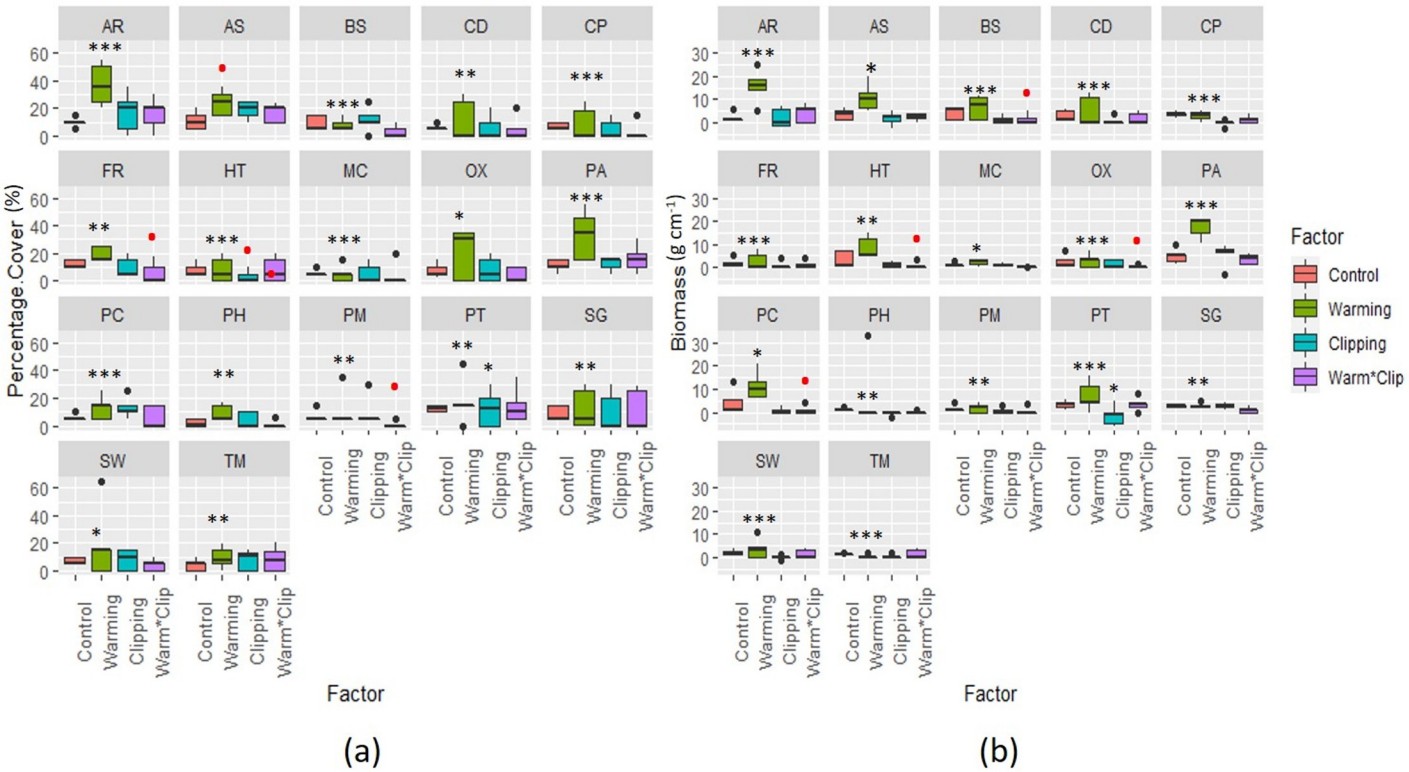

**Fig 3. Species-specific responses towards each treatment. a)** Data showing the percent increase in cover of each species in control, warmed, and clipped plots. **b)** in above ground biomass of each species in response to warming, clipping, and warming x clipping in comparison to control, warming has an overall positive effect on percentage cover and biomass of species while clipping has a negative effect. The * shows a significant response in particular species to the specific treatment while the dots are outliners. *The significant codes "***' 0.001 "**' 0.01 "*' 0.05 '.' 0.1 ' ' 1.*

response magnitude of percentage cover and aboveground biomass to clipping was stronger than the experimental warming in interaction (W x CL) plots (Fig 2, Table 2). It has decreased the aboveground biomass of species significantly (<0.01) as shown in Fig 2 and Table 2.

Overall, our NMDS analysis revealed that plant communities showed higher differences among sites than among warming treatments, reinforcing the idea that plant responses were different at different sites (S3 and S2 Figs).

We further assessed the response to experimental warming and clipping for the five most cited culturally important plant species; *Artemisia rupestris* (AR), *Astragulus penduncularis* (AS), *Poa alpina* (PA), *Potentilla hololeuca* (PT) and *Primlua macrophylla* (MC). These species responses were divided into four categories: combined positive effects of warming and clipping/ grazing (+W, +G), combined negative effects of warming and grazing/clipping (–W,–G), and antagonistic effects (–W, +G or +W,–G) (Fig 4, Table 1). The mean percentage cover of species was affected positively by warming and negatively by grazing (Fig 4, +W,–G zone). One species, *Artemisia rupestris*, was favored by both warming and grazing. However, *Poa alpina* and *Potentilla hololeuca*, had higher cover in warmed plots, while Primula *macrophylla* negatively affected by both treatments, did not show any significant response to warming (Fig 4).

## Discussion

### Experimental warming effects

Our short-term manipulative experimentation of temperature and clipping led to the distinct responses of vegetation cover and aboveground biomass. However, these responses were

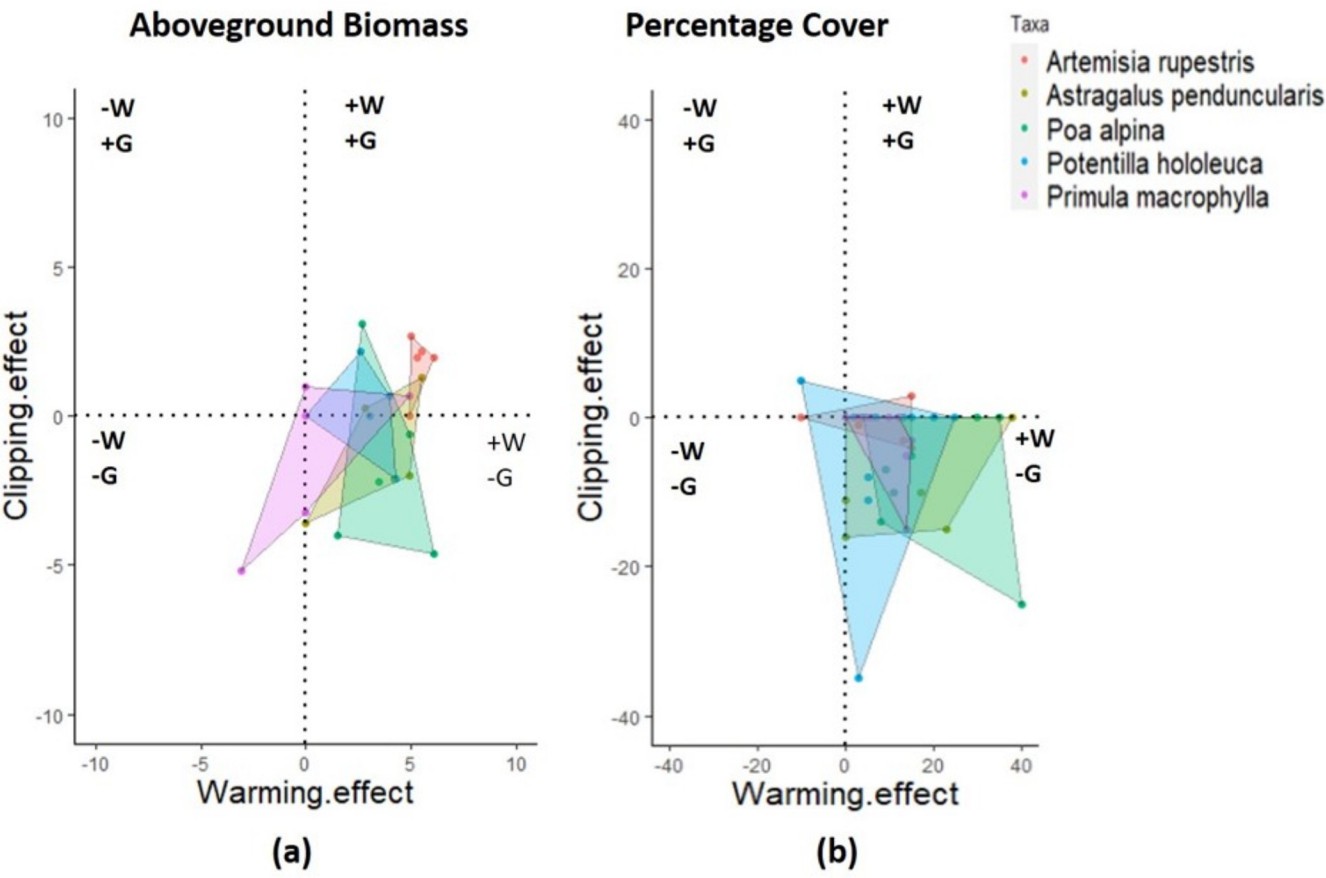

**Fig 4. Susceptibility zones of the five most frequent and culturally important plant species under combined impacts of warming (w) and clipping (g) treatments a) warming and clipping effect on aboveground biomass of plant species. b)** warming and clipping effect on percent cover of plant species. *The susceptibility zones were combined positive effects (+W, +G), combined negative effects (–W,–G), and antagonistic effects (–W, +G or +W,–G). Polygons join the dots of the presence of a species in the respective zone. * represent the significant positive impact of warming.*

individualistic for specific species and to the specific treatment. We combined three manipulative experimental treatments of warming, clipping, and warming x clipping as interaction plots (W x CL), to observe the overall effect and species-specific responses to each treatment. We hence concluded that experimental warming generally increased the ability of alpine plant species to increase their cover and biomass, but its effect depends greatly on the management of grazing practices. The fact that we observed was increased biomass and percentage cover at all sites, whereas clipping alone and combined interaction of clipping with warming treatment significantly reduced the biomass of species. This is in line with the previous warming experiments and studies carried out to assess the effects of increasing temperature on alpine ecosystems which also showed that warming enhanced the plant vegetation cover and height [23, 30, 52, 63], and this may have positive impacts on overall vegetation richness and productivity in cold areas [2, 64–66]. The significant positive effect of warming was in line with the information given by informants of the ethno-botanic survey, who emphasized the change to denser vegetation with projected climate warming. They also suggested that it may favor the competitive species to grow faster than others. This taxa-specific pattern of plant community responses to warming has been stressed in challenging environmental conditions such as the Arctic [13, 53, 63, 67, 68]. Likewise, experimental warming did not significantly increase the biomass of some species rather reduced the biomass of *P. hermala* and *S. gonospermum* in comparison to

control plots. These results are contradictory with De Boeck 2018 [69] who investigated the legacy effects of warming on biomass production in alpine meadows. This reinforces the claim that warming may generate a change to the environment in a diverse way which will be suitable for the adaptable medicinal plant to grow, thus being available for livelihood [70].

Warming may lead to enhanced plant growth and cover and it reduced the diversity of species as compared to the clipping treatment. However, the increase in specific plant species cover was observed along the elevation gradient but this effect was more profound for specific species, making it a random effect than fixed, in warmed plots as compared to control and other treatment plots (S2 Fig). This is attributable to increased competition; that when biomass increased, plants were usually taller, and few individual species could fit into plots.

In clipping treatment plots, the plant species did not grow well and had reduced their biomass and cover, but they maintain the diversity. This mechanism was elaborated by Borer et al, 2014 [71] who demonstrated the effect of herbivory on grassland production and revealed that where herbivory increased the ground light, it also maintains the biodiversity. Hence, where warming benefits some plant species in increasing their cover and biomass, it is likely to induce the dominancy of some species, whereas our experimental results suggested that clipping rescued the biodiversity. Conclusively, *Poa alpina*, *Artemisia rupestris*, *Primula macrophylla*, *Astragulus penduncularis*, *Potentilla hololeuca*, *Carex divisa*, *Comastoma pulmonarium* have increased their biomass in response to the elevated warming, and these species were observed by locals to be in abundance in the meadow's pastures. Experimental evidence was in line and supported the fact of these species being abundant to all pastures and important for the wellbeing of inhabitants (Table 1). Contrastingly, some conflicting exceptions opposed the indigenous knowledge about medicinal plants. *Oxytropis glabra*, *Taraxacum officinale*, *Saxifraga*, and *Pegunum hermala* have increased their biomass in warming plots, but local informants did not observe any evident effects (Table 1) in these plant species.

Bridging indigenous knowledge with the experimental manipulation of the alpine ecosystems considered an important approach for designing adaptation policies [47]. People living in high altitude areas appeared more sensitive to climate change. The impact of local climate change on traditional livelihood was strong both from the survey and experimental results when analyzed for positive as well as negative aspects. Several studies have shown the importance of traditional plant use for indigenous people in the Himalayas [17, 44, 50, 72]. Our study revealed that local people had extensive knowledge about plant identity, occurrence, population trends in recent years, and their medicinal and nutritional properties. Plant species from *Asteraceae* (e.g., *Artemisia rupestris*, *Artemisia santinofolia*, *Artemisa rutifolia*) were particularly important for the family medication (cited by 25% of respondents) and has the highest RFC value (0.9). In our survey, the promising reports of using medicinal plants to treat high blood pressure, gastrointestinal diseases, skin and eye allergies, the cardiovascular disease could be due to the prevalence of these diseases, but also the absence of effective and approachable pharmaceuticals. The importance of these plants has been highlighted in biodiversity and traditional medicine [41, 44] by other researchers.

### Clipping effect

The clipping effect marginally reduced the overall biomass and cover in all growing seasons (Table 2, Fig 2). The percent cover of the overall species was reduced non-significantly both in clipping and W x CL plots. The studies on alpine meadows in southern Norway also suggested that clipping reduced the aboveground biomass [13]. Similarly, the same effect was reported in Hiabei region, where heavy grazing resulted in a decrease in above-ground biomass and degradation of alpine pastures in a 4-year study [73]. However, clipping non-significantly increased

the biomass of *Artemisia rupestris (AR)* and *Carex divisa (CD)* in our studies (S4 Table, Fig 3) while the *Hedinia tibetica (HT)*, and *Oxytropis glabra (OX)*, and *Pedicularis kashmiriana (PC)*, responded significantly negatively towards W x CL experiment. These different responses of vegetative cover and biomass to clipping/simulated grazing can be attributed to species-specific traits which allow the plant species to regrow faster than the others. Also, where clipping helped certain plant species by increasing the ground level light, warming triggered the temperature and plants responded positively to the clipping (CL) and W x CL treatments. Some plants have evolved tolerance to herbivory by growing compensatory tissues so rapidly that help to cope with damage, prompting plants to develop a new form of resistance such as reduced apparency, chemical defenses (secondary metabolites), and indirect defense [74].

The interplay of species growth and grazing has received much attention in the ecological literature. Numerous studies suggested that grazing is one of the major drivers of rangeland degradation [20, 54, 73, 75, 76]. Our results supported this assumption, where clipping treatments had significant negative effects on plant biomass and non-significant negative effects on percentage cover. The general trend was a decline in cover and biomass and this decrease was highly variable among the taxa also the random effect of the sites can also be a predictor of variability in response [77].

## The combined effect of warming and clipping–interactive experimentation

To better place our results in this context we explained the interactive effects as stronger clipping effect being an important finding than warming. The relative effects of warming and clipping on vegetation growth are not fully understood. The warming positive effect was not consistent in the interaction plot explaining the plants' sensitivity to combined stress. The different responses of plant communities to biomass and cover can be attributed to clipping frequency and intensity which had overridden the warming positive impact. Moreover, clipping may induce compensatory growth, but it can decrease with clipping intensity. These effects were explained by Fu et al., 2017 [77] who showed that clipping reduced the leaf area thus decreasing plant photosynthetic ability by reducing the fractional radiations absorbed by plant species. The combined effect explained in Fig 4 is also supported by previous studies that the main effect of warming was more dominated in absence of grazing (+W,–G) that is significantly proven by the mixed effect model (Fig 2, S4 Table).

The magnitude of positive responses of species as increased percentage cover and aboveground biomass was in the zones of only warming treatment (+W,-G), in absence of clipping (Fig 4), thus predicting the availability of these species as preferred forage if the grazing is controlled [76]. The relatively small effects of experimental warming than clipping in interactive plots on vegetation biomass and cover can be related to experimental warming-induced soil drying and the frequency of clipping. The plant species are under the stress of two climatic drivers. Thus, warming has negligible positive effects on vegetation in presence of clipping. Grazing is reported to have strong effects on carbon storage than warming. It can significantly decrease leaf area and plant photosynthesis [76–78]. This mechanism explained the relatively large effects of clipping on vegetation biomass which overrides the warming effect in interaction plots. Again, the species-specific response of these taxa is shown in the biomass susceptibility zone where some species as *A. rupestris*, *and P. alpina* had non-significantly increased the biomass The natural selection may play a role for these plants to be more abundant, thus palatable for grazers. Our results support the clipping exclusion advantages on plant community as explained by Wang et al., 2019 where biomass was increased significantly in grazing exclusion treatments [73]. Thus, predicting future vegetation in the alpine ecosystem requires the knowledge of grazing interaction with climate variables which can influence the dynamics of the alpine ecosystem.

In our study, some of the species (*P. alpina* PA, *A. rupestris* AR) responded significantly towards climate warming and non-significantly towards clipping by not reducing their biomass (Fig 3B, S4 Table). This supports the assumption of natural selection where strong competitor species surpass weak competitors, thus lead to loss of biodiversity [33, 52]. This experimental evidence was also in line with the information gathered from the local informants, mentioning increased forage availability of these species throughout climate warming.

The interaction of plant community at all sites suggested that site 1 (M1, 4696 m) had maximum plant species diversity highly favored by warming treatment. Gentili et al., 2015 [79] showed that alpine life zones exhibit great biodiversity of cold-adapted biota which are sensitive to climate warming but some have adapted numerous strategies to cope with climate change. Henceforth, explaining the determents which shape the biodiversity of alpine species. The site effect on plant biomass and the cover was not profound at a low altitude site (M4, 3359m; M5, 3346m) compared to M1 (S2 Fig). We were informed during the survey about the culturally important plants that were migrating towards high altitudes such as *Hedinia tibetica*. Warming benefits the migration of the lowland species to high altitude tracking climate warming and this migration is likely to be important for the performance of alpine species [52].

The approaches studied in this paper, coupled with the people's perception and experimental shreds of evidence, may differ in their ability to measure the indirect impacts (human and environment interaction, health risks through rising air temperatures and heat waves) of climate warming but can provide a very rapid estimation of short-term direct impacts. The future prediction of alpine biodiversity of culturally important species depends highly on the optimal grazing practices in the current climate warming scenario in HKKH. The elevation gradient effect (site effect) would reflect the change in species community in long-term experimentations short-term interactions [80]. Here in our study, combining and integrating more than one approach of local knowledge about culturally important plants and providing the experimental confirmation of climate warming and grazing impacts on them, can be the best tool to examine the direct warming and herbivory impacts on the alpine ecosystem. Alpine ecosystems, having narrow geographic areas and constrict elevation range are more vulnerable to climate and vegetation changes [81]. Finally, in addition to the factors of global changes mentioned in current studies, the relationship between indigenous knowledge at mountain systems and the abiotic processes can be helpful to drive the future dynamics of ecosystems and designing policies [54, 76].

## Conclusion

Overall, our study is the first to provide experimental evidence, at Khunjerab National Park, of the combined effect of warming and grazing on culturally important plants. Our results provide valuable information for the evaluation and prediction of alpine sensitivity to future threats. Linking indigenous knowledge with experimentation and evaluating the combined and interactive manipulation experiments are likely the advanced tools to examine the responses of alpine community which has been proven by our studies. It can provide a jumping pad to policymakers for designing, mitigating, and adapting strategies for climate change in a region that is undergoing rapid change and for which scientific data are meagre.

## Supporting information

**S1 Fig. Percentage citation of each disease treated by medicinal plants.** Percent responses of informants for different diseases treated by medicinal plants.
(TIFF)

**S2 Fig. Treatments, site and species effects on culturally important plant species percentage cover and biomass. a)** the percentage cover of plant species increase in warming treatment and at multiple sites(M1, M2, M3) that is because of high abundance of these species on these sites while among species the response of individual species was not very different from each other but some species responded more positively by increasing their percent cover as AR and AS **b)** there is a decline in biomass of species in clipping treatment as compared to control but the significant positive effect of warming treatment. The side effect is not very different among all site similarly some species biomass increases while others received a decrease in the biomass.
(TIFF)

**S3 Fig. Non-metric multidimensional scaling (NMDS) plot of plant species biomass-with 95% confidence intervals using Bray and Curtis dissimilarity index over a 3–years experimental period along the elevation gradient of (3590–4696m).** The relative abundance of plant species biomass inside the OTCs on each elevation site was estimated for all study years and shown by paths of mean values in Non-metric multi-dimensional scaling (NMDS) using Bray and Curtis dissimilarity index in R The plot shows no significant difference between the relative abundance of there is no significant change in the relative abundance of species inside both OTCs. OTC = Open top chamber, OTC1(warmed), OTC2, (warm*clip). Site 1 4,696m, Site 2 = 4,059m Site 3 = 4022m Site4 = 3,990m Site 5 = 3,590mDifferent color represents each study year, red, 2016, green, 2017, blue 2018.
(TIFF)

**S1 Table. Demographic and socio-economic characteristics of informants.** Demographic features of informants interviewed for ethnobotanical information of culturally important plants. In the survey, their gender, age and socioeconomic details were recorded. A variety of plant species in experimental sites at different elevations. Each site was categorized according to the number of species present. majority of selected species were present at each site, but there was a representative specie of each elevation. *Bistorta officnalis* is present only at the highest altitude(4696m), similarly *Plantago major* (3690m) is present on site 5, lower altitude.
(DOCX)

**S2 Table. Plant species composition at experimental sites.** A variety of plant species in experimental sites at different elevations. Each site was categorized according to the number of species present. Majority of selected species were present at each site, but there was a representative specie of each elevation. X" represents if the specie is present at one or all five sites. "X*" are selected plant species.
(DOCX)

**S3 Table.** a: Effect of warming on species percentage cover and biomass within the treatment. One way-ANOVA summary representing the individual percent increase/decrease in cover and biomass and the significant difference between the response of each species within the treatment. Significance codes: 0 '***' 0.001 '**' 0.01 '*' 0.05 '.' 0.1 ' ' 1. (b) Linear effect model summary of treatment, site and species effect on percentage cover and biomass. Linear mixed-effect model summary for each factor considered as fixed to evaluate the overall effect on aboveground biomass and percentage cover. Significance codes: 0 '***' 0.001 '**' 0.01 '*' 0.05 '.' 0.1 '' 1. (c) Summary statistics of mixed effect model for the significant effect of treatment, site, and species on percentage cover. (d) Summary statistics of mixed effect model for the significant effect of treatment, site, and species on aboveground biomass.
(DOCX)

**S4 Table. Summary from fitting a linear mixed-effects model by restricted maximum likelihood, to predict species-specific response to each treatment.** Significance. codes: 0 '***' 0.001 '**' 0.01 '*' 0.05 '.' 0.1 ' ' 1. Marginal $R^2$ value represent variance by fixed effects and conditional $R^2$ explained variance by the entire model (fixed and random).
(DOCX)

**S5 Table. The true data points are attached as excel file in supplementary information.**
(XLSX)

## Acknowledgments

We are highly thankful to the Snow leopard foundation (SLF), local influential persons, hunters, and guides who facilitated and supported the survey team. The English language of the manuscript has been improved by Prof. Dr Richard Goodman which is also acknowledged.

## Author Contributions

**Conceptualization:** Zahid Ali.

**Data curation:** Saira Karimi, Olivier Dangles.

**Formal analysis:** Saira Karimi, Olivier Dangles.

**Funding acquisition:** Muhammad Ali Nawaz, Zahid Ali.

**Investigation:** Saadia Naseem, Hussain Ali.

**Methodology:** Saira Karimi, Muhammad Ali Nawaz, Hussain Ali.

**Project administration:** Zahid Ali.

**Resources:** Zahid Ali.

**Supervision:** Zahid Ali.

**Validation:** Olivier Dangles.

**Visualization:** Ahmed Akrem.

**Writing – original draft:** Saira Karimi, Zahid Ali.

**Writing – review & editing:** Saadia Naseem, Olivier Dangles.

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
