## [Decision Letter · Decision Letter 0]

11 Sep 2020

PONE-D-20-21262

The response of culturally important plants to experimental warming and grazing in Pakistan Himalayas

PLOS ONE

Dear Dr. Ali,

Thank you for submitting your manuscript to PLOS ONE. After careful consideration, we feel that it has merit but does not fully meet PLOS ONE’s publication criteria as it currently stands. Therefore, we invite you to submit a revised version of the manuscript that addresses the points raised during the review process.

Both of the reviewers highlighted that your interesting study covers an important but understudied topic, and they appreciated the combination of experimental-ecological and ethnobotanical approaches. However, the manuscript needs more detail and clarity, as well as some improvement of statistical analyses, and has to be extensively revised. At this place, I would like to refer to two major issues only:

First, the description of methods has to be improved. It remains unclear how the experiment was designed, thus more details of the experimental design have to be added. For instance, were all the (sub-)plots located within exclosures but clipped to simulate grazing? Was it a two-factorial experiment, with the factors warming and clipping crossed? Clarifying these details is essential to evaluate the appropriateness of statistical methods and the interpretation of results as well as to understand the discussion and the conclusion drawn.

Second, assuming that it was a two-factorial experiment with replicates along an environmental gradient, the statistical analyses (just comparing species within each treatment) would be inadequate. An appropriate analysis will be a two-factorial ANOVA including the main effects of warming and of clipping as well as the interaction between these two factors. This model would then, for instance, allow assessing if the effect of warming depends on whether vegetation was clipped or not (reflected by the interaction term), which is an important scientific question itself. Accordingly, results of the warming x grazing treatment have to be reported and discussed as well. Furthermore, if you consider the site effect to reflect some random variation, you should use a mixed model and include site as random effect. On the other hand, if you are interested in differences between sites, i.e. if you have a specific hypothesis how sites should differ or how the effects of clipping and warming should change along your environmental gradient, you may even include site and its interactions as fixed effects! If you conducted repeated measurements on the same plots, you could either average your data across census dates or should apply a repeated measure ANOVA. The two reviews provide many helpful suggestions how to improve your statistical analyses.

In general, the writing is a bit confusing and vague. The text has to be restructured in parts based on the comments by the two reviewers, and has to be carefully edited for grammar. The two reviewers provided many additional comments and detailed suggestions that have to be considered in a revised version of your manuscript.

We look forward to receiving your revised manuscript.

Kind regards,

Harald Auge

Academic Editor

PLOS ONE

Journal Requirements:

2. In your Methods section, please provide additional location information of the study sites, including geographic coordinates for the data set if available.

4. In your Methods section, please provide additional information about the participant recruitment method and the demographic details of your participants. Please ensure you have provided sufficient details to replicate the analyses such as: a) the recruitment date range (month and year), b) a description of any inclusion/exclusion criteria that were applied to participant recruitment, c) a table of relevant demographic details, d) a statement as to whether your sample can be considered representative of a larger population, e) a description of how participants were recruited, and f) descriptions of where participants were recruited and where the research took place.

Reviewers' comments:

Reviewer's Responses to Questions

**Comments to the Author**

1. Is the manuscript technically sound, and do the data support the conclusions?

Reviewer #1: Partly

Reviewer #2: Partly

2. Has the statistical analysis been performed appropriately and rigorously? 

Reviewer #1: No

Reviewer #2: No

3. Have the authors made all data underlying the findings in their manuscript fully available?

Reviewer #1: No

Reviewer #2: No

4. Is the manuscript presented in an intelligible fashion and written in standard English?

Reviewer #1: Yes

Reviewer #2: No

5. Review Comments to the Author

Reviewer #1: Note to the editor and authors: Our evaluation of the ms is from the perspectives of plant community ecologists, as such, we are not qualified to speak to the ethnobotany survey approach and hope another reviewer will cover that aspect.

Overall, this paper has potential to be an interesting study and a good contribution to the field of climate change responses, especially with the work in an important area for both biodiversity and human cultural, livestock, and medicinal use. Also importantly, the work covers a region of alpine tundra which is underrepresented in plant community climate change responses, which tend to be overly dominated by N. American and European datasets. As such, we see substantial value in the data the authors have collected.

However, lack of clarity on the experimental design and potential issues with the statistical approaches make it hard to interpret the results. We have focused our suggestions on areas in the methods where the experimental design can be clarified, and some suggestions on how better to approach the statistical analyses given the study design. We recognize that these may seem like intensive changes, but encourage the authors to tackle it because we see the great intent of their project.

Major comments:

1. The experimental design is really unclear. A diagram to include in or replace Figure 1 would help clarify this immensely. It was unclear whether all the measured plots were inside the grazing exclosures, or if any data was collected outside of the exclosures. It was also unclear whether there were four subplots in a factorial design (clipped, warmed, warmed + clipped, control) or some other nested design. Clarifying this is vital for readers to evaluate the statistical methods, interpret the results, and understand the discussion.

2. The statistical methods don't seem to be appropriate. It's hard to suggest an exact method that would be appropriate since the experimental design is unclear. However, a two-way model with grazing and warming as separate factors (and potentially an interaction term as well, depending on design) would be a good start. Additionally, we suggest using a mixed-effects model with a random effect for site. The authors repeatedly note that site effects were stronger than the treatment effects, and using a random effect for site will help separate these factors. Good packages in R for these types of simple mixed-effects models are 'lme4' using the lmer() function and 'nlme' using the nlme() function.

For example, if there is no data collected outside the fences, and the experimental design is fully crossed warming and clipping (clipped, warmed, warmed + clipped, control), a good simple model specification (using lmer syntax) would be:

biomass ~ warming * clipping + (1|site.id)

(which is the same as)

biomass ~ warming + clipping + warming:clipping + (1|site.id)

This would then give you an output for the effect of warming, the effect of clipping, the interaction between warming and clipping, and with a random effect (intercept) for site idenity (ie: site A, site B, ...) and you could run this model for each species individually. And similar separate models for cover.

There are more complex ways to approach this as well, including having all the species together in the model with either another random effect (intercept) for each species. However, if you think species respond differently to the different treatments, you would want a random slope, not just intercept. If you choose to go this route, or if there are also data collected outside the exclosure fences and you have a nested design, we would recommend consulting a statistician to help with the model design. Again, clarifying the experimental design would really help with this.

An alternate approach if you think the differences between site are based on the altitude gradient, with predictable effects at high sites versus low sites, would be to use ANOVA with interaction terms (using the aov() function), and site altitude as a fixed predictor rather than site id as a random effect. The model specification could be:

biomass~ warming + clipping + site.alt + warming:clipping + warming:site.alt

although you might also choose to incorporate an interaction between clipping and site altitude, or even a three-way interaction if you think that's appropriate.

3. There are findings referred to in the discussion that are not included in the methods or results. These need to either be clearly incorporated as main parts of the paper, or removed from the discussion. For example, flower production, analyses of community composition, and differences in response based on site location--see more detailed comments in the "discussion" section below.

The discussion in general is not clearly organized. It would help a lot if you identified your 3-4 main results, and then gave each of these a paragraph in the discussion (then connecting it to existing literature as you have done).

4. Please put the figure captions with the figures, it's very hard to evaluate it this way.

5. The data summaries are included but none of the underlying data (the actually collected data points for each plot etc) is available in the supplements or stored in repositories (ex Dryad, Figshare, etc) online.

Minor Comments:

Abstract

- No information about any interaction between grazing and warming. Unclear if the experiment is set up to test this and they didn't find it, or if there were just the two treatments compared to one control.

Introduction

line 47 – 48 can you state a reference for this?

line 56 this reference is not specifically about grazing, can you find more suitable ones?

line 56 unclear what is meant with “selective palatability issues”

line 64 you could start a new paragraph after “plant cover reduction”, this would facilitate reading the introduction and adds a clearer structure

line 69-71 meaning unclear

line 75 has previous research actually dealt with overgrazing or simply with the impact of grazing?

line 75 -76 add references to the first part of the sentence and specify that diversity in plants in general may be well covered by studies but not so information about medicinal plants. The way the sentence is phrased now does not convey the important message that studies on medicinal plants are still underrepresented

line 80 -81 sentence unclear, explain “benefit-relevant indicators”

line 83 – 84 you could delete this sentence, since it does not add important information

line 85 specify “resource” of what?, f.ex. income, remedies, etc.

line 86 "Second,"

line 91--would be useful to give a rough metric of historical livestock vs current, instead of just saying it's increasing

line 94--these two objectives seem pretty distinct, can you make the connection between them more clear?

I think in general the introduction could be improved by adding more structure and clarity by defining open questions/research gaps for each paragraph.

I also feel that references are partly missing or are not used in an accurate way

Materials and Methods

line 104 "July. "

line 105 warmest year? warm years?

line 106 only few °C below zero or much colder?

line 104-107 would be good to know temp and precipitation in the specific study years

line 110 "macrophylla, "

line 112 "cattle and yaks" (although I recognize that cattles is also used in many countries, but it's the less common form)

line 114 should be “Plantago lanceolata”

line 115 should be “Poa alpina”

line 117 greater than 5026 compared to what? And in what time frame and what is the unit (number of herbivores?)? Also 5026 is a very specific number, maybe just greater than 5000?

Line 118 “immense burden” in what respect?

line 119 varied rather than "the most versatile"

line 119 - 120 need language edits in general

line 128 "useful medicinal plant species are"

line 128 are these questionnaires accessible in the SI? If so, please refer to them.

line 119-126 this could be part of the introduction and would nicely support the aim of your study

line 129 could you add information on how these valleys were different? Given you study the impact of warming and grazing on specific medicinal plants it would be interesting to know the general species abundance of those plants in these valleys. We assume you chose these valleys because they combined people with medicinal plant knowledge and abundance of your studied species?

line 141 expression “collection of data the observations” unclear

line 142--would be good to refer back to the altitudinal zones mentioned earlier--seems like this falls primarily into the middle zone (Alpine meadows)? If so, it may not be necessary to give such detail on the other two zones in the study site description.

line 143 please also provide more details about the fence material you used i.e. height, mesh size, since this will inform the reader about the dimensions of the potential grazers in your system

line 148 your experimental set up seems to focus on summer warming only. Does this align with predications for your study area? Given that your fences allow smaller rodents to pass, how did the OTCs affect those and their impact on the vegetation?

line 149 does your observed temperature increase correspond with observations from other studies? Add references.

FIGURE 1 -- the photos are good but (a) should ideally show the grazing exclosures (if you have it) and it would be very useful to have a basic plot layout diagram to clarify the set up. It's unclear what photos d-g are supposed to show.

line 151 -- are the clipped subplots nested within the grazing exclosures? or within the open top chambers? Did you actually compare un-fenced areas, or is the clipping the only grazing treatment? That seems surprising since you have potential to do fenced and unfenced comparisons... would want better rationale for this decision. Herbivores can have many affects other than just clipping (dirt impaction, dung, nutrient cycling...)

line 152-- no clipping, or no warming? So there were 2 warmed, 1 clipped, and 1 control? Or 2 warmed (1 clipped, 1 unclipped) and 2 unwarmed (1 clipped, 1 unclipped)?

line 153 vegetative height?

line 154-156 unclear what this means, also how often did you clip the plants, does the clipping treatment mimic the natural grazing impact, did you mark the grazing level on individual plants? Please clarify what you have measured and if your measurements were done before or after the clipping and whether they were repeated during the peak season or several years.

line 169 the reference you state does not seem to focus on vegetation but on ground dwelling insects

line 171 what are the selected plant species?

line 174 briefly explain this method in a short sub sentence

line 176 did you sample all occurring species or only those that you previously selecting. If the latter, why were they sampled for classification?

line 178-179 Was this done in each year or just at the end of the study? Was it only done in the open top chambers or also the unwarmed controls? Was it for the whole 1 m x 1 m plot or just a portion of it? Was biomass sorted to species or just total biomass?

line 179 What was the timing of the clipping relative to the plot surveys? Did you survey before or after clipping?

line 180 why not a two-way ANOVA? are these treatments not interacting with each other?

line 181 If the treatments are interacting, it should be a two-way ANOVA. If they're not interacting, a Tukey test still doesn't seem like the best fit, since it's generally for different levels of the same type of treatments (ex-low intensity vs high intensity grazing). We have suggested a mixed-effects model as the best option (see major comment #2 above).

Given that you talk about “plant species” did you run separate models for each species?

line 188 "The 'vegan' R package"

Table 1 The inputs on what people have observed as climate change effects are really interesting! I think you could maybe highlight this more in your abstract/methods as a key component of the study. It would also be really interesting to directly compare the measured cover/biomass responses of species in the three categories: 1. observed by locals to have increased 2. observed by locals to have not changed in abundance 3. observed by locals to have decreased. Even if this is too small a sample to be statistically tested, it would be a great way to bridge the two components of the study. Also looking at the last two where the locals have observed range shifts--does this fit with the study's findings?

Is “no effect” and “not observed” the same? What does “forage availability” mean, i.e. due to climate change this species is now available for forage?

Just out of curiousity: all species occur at quite high elevations, presumably with snow cover during winter, still some of them are available all year round (f. ex. flowers and leaves of Primula macrophylla) is there simply no snow cover or do these species prevail under the snow?

line 227 Saxifraga -- particular species?

Fig 2. not really neccessary, can just go into the table

Fig 3. Looks like the warming and grazing treatments were crossed?

line 237 All the more important to include site as a random factor in the model, also did you expect to see differences among warming treatments?

Line 240 is the grazing treatment the actual grazing treatment or the clipping treatment?

line 259 This is where a two-way ANOVA with the interaction term would really show these results, not just having to visualize them graphically.

line 259-260 this is interesting and novel and should be emphasized more throughout the ms

line 264 are these results from a model across species?

line 265 -269 what are the statistics for these results? Are they mentioned in the data analysis part? If not, please add!

Figure 4 Are these the averages across sites? How do you define "susceptibility zones"? Please add more detail to the methods about this approach

S2 Fig are the different colored points the three years? I think this would be clearer without the lines. Are the non-warmed plots here to compare to?

Discussion

In general, try to line the discussion sections up with your main results, which will help with the organization of it. In the very beginning you can have a summary of your most important findings, which should not only focus on the ethnobotanical survey, but also on the warming/grazing results.

line 294 – 296 have you tested this relationship?

line 297-299 this is repeated

line 307 Did you analyze community composition overall? Or just visually with the NMDS? There is a function for this in the 'vegan' package, adonis() for PERMANOVA analysis that would be useful to support this part of your manuscript especially if it is one of your key findings.

line 309 You haven't mentioned flower production as part of your methods or results--either remove this from the discussion or include it as part of your paper (i.e. in Methods and Results).

line 311 Where in your results section do you discuss differences between species responses to warming at different elevations? This would be really interesting, but it isn't clear.

line 313-314 “new opportunities” this is quite a bold statement, maybe you can put it in context with medicinal plant use

line 314 to improve the structure of your discussion you could start a new paragraph after ending with “… for people living under cold conditions”. Remove “while” from the beginning of the sentence, otherwise the sentence does not make sense

line 318-319 as mentioned earlier it is unclear whether your data are from inside fences only or also from outside the fences. If you did not collect data from outside the fences than it does not make sense to argue with different cattle density. Please clarify this already in the Methods and adapt this sentence accordingly.

line 321 Here you mention cover collected outside cattle exclusion plots, but it's not clear from your methods or results that this data was collected and how it was collected/analyzed. Please either incorporate it fully or remove it from the discussion.

Conclusion

line 329 “grasslands” is mentioned here for the first time, so it feels a bit out of place, maybe you can add “alpine”?

line 330 this sentence is nice, though we think an evaluation of ethnobotanical survey methods is not the focus of your study. As plant ecologist we suggest to highlight here plant responses to your experimental treatments and put that into context with the medicinal plants.

Reviewer #2: This study addressed how culturally important alpine plants will respond to climate warming and increased grazing pressure. This is a relevant and understudied topic, and I appreciate the integration of experimental manipulations with data collected from interviews of local people. However, I was confused by the study design- were the grazing treatments simulated grazing (clipping) as stated in the methods (lines 151-152), or grazed by animals, as suggested in the discussion (lines 318-319)? There was mention of grazing exclosures, but it was not clear whether plots were established both within and outside of them. Also, how many plots were established at each site? A diagram of the different sites, plots, and subplots would be helpful in clarifying the study design. Also, were species responses averaged over all 5 sites? The authors state that there are large differences between the 5 sites along the elevational gradient, so it doesn’t seem appropriate to completely ignore this gradient. It seems that the 5 sites were treated as experimental replicates, but they are inherently different. Were there replicate plots at each site, as well?

In general, the methods and results could use much more detail (see line edits below). For instance, wasn’t there a warming x grazing treatment? I don’t see those results reported anywhere- just the separate effects of warming and grazing. Also, the discussion section could provide more synthesis about how the survey data and experimental results complement each other.

Overall, this is an interesting study (great work!), but the manuscript needs more detail and clarity. It also needs to be revised for grammar.

Line edits:

Lines 25-26: The description of the treatments is confusing, and lists 2 separate “controls”

Lines 30-31: Is the percentage cover or biomass reported here?

Lines 67-68: how do nutrients “counteract” the harsh environment- need more explanation

Lines 70-71: This is worded awkwardly

Line 80: Please explain what “benefit-relevant indicators” are

Lines 128-137: Need much more detail about the surveys. What types of questions were asked? Were they oral or written? How were people selected for the survey?

Lines 143-144: How do 2 plots fit inside a 5x5 m fenced area, when a single plot is 5x5 m (containing four 2.5x2.5 m subplots)?

Lines 151-152: The un-closed parentheses make this sentence confusing. What is considered a “control” in this experiment? It would be helpful to assign unique names to all your treatments so that you can remain consistent when referring to them.

Line 154: Does clipping (simulated grazing) one time each year effectively simulate the continuous season-long grazing this region usually experiences?

Line 169: I know you cite a source for this, but it would be helpful to briefly describe the “quadrat sampling method.”

Line 172: Size of the quadrat grids? Also, I thought the subplots were 2.5 x 2.5 m (not 1 x 1 m)?

Lines 174-175: Plant height is measured, but not reported in the results.

Line 177: What does it mean that plants were “sampled”? Weren’t they just identified to species?

Lines 178-179: Were the entire plots clipped? And does repeated clipping of the same location for multiple years affect results? Are plant species mostly annuals or perennials? Were the control and grazing plots also clipped (only the OTC plots are mentioned).

Lines 180-181: It looks like the ANOVAs were just done to find differences between species within a single treatment (warming or grazing). However, you might want to compare the responses across the different treatments to determine if there are differences between plant growth in the control, grazing, warming, and grazing x warming plots. I find that a much more compelling question than differences between individual plant species. Also, it looks like you compared relative increases of the treatment plots compared to control plots (% increase/decrease), but this is not stated anywhere in your methods. Finally, are responses averaged over all 3 treatments years? Since you sampled the same plots multiple years in a row, you probably need to use a repeated measures ANOVA.

Lines 190-192: This sentence is not needed.

Lines 192-197: This should be included in the “Ethnobotanical Surveys” section of the Methods. It is out of place here.

Lines 201-207: Need more details about the results of the surveys

Line 213: I think a part was omitted since it starts in the middle of a sentence.

Line 234: Since you calculated % increase in the treatment plots compared to the control plots, how did you account for pre-existing variation in plant cover of the different species between the 2 plots?

Lines 259-269: What data is used here? It looks like you are using separate plant responses to just the warming and just grazing plots. Where are the results for the warming x grazing plots?

Line 295-296: How does the RFC value influence plant availability in the warming treatments? I thought RFC was just an indicator of how often people mentioned the species in the interviews.

Lines 297-299: These sentences are repeated.

Lines 309-310: This is the first mention of flower production- it is not reported in the methods or results. You can’t introduce new results in the discussion section.

Lines 323-324: “A constant increase in species cover through time can occur under moderate grazing conditions” --- Are there sources to back up this claim? Your results do not necessarily support this.

Lines 334-338: Citations for this? Also, this would be good to mention earlier, maybe in the introduction.

Tables & Figures:

Table 1: You should specify which information is from the surveys and which is from another source (which should be cited). Also, how are the “most preferable” forage species determined?

Fig 1: These pictures are lovely, but I think they are mostly unnecessary. Maybe just panel b showing the OTC. A better figure would be a diagram (or annotated picture) of your plots and study design.

Fig 2: It might be nice to color the bars by plant functional group (shrub, forb, grass), so someone unfamiliar with the species can at least know the growth form.

Fig 3: The figures are very fuzzy, so it is hard to read, but I don’t see any letters showing statistical significance as stated in the legend. Also, what do the geometric shapes around each error bar show? You need units for the y-axis: Biomass (% change).

Fig 4: What does each point represent? A different site? A different year? Also, similar to my comment above, what data is shown? I don’t see how you can report these 4 zones if just showing the response to the combined warming x grazing treatment.

Fig S1: The colors in the pie chart (panel b) are impossible to distinguish- there are way too many categories. Consider an alternative format. Also, I’m unclear as to what is shown in that chart- is the “study area” across all 5 sites? And before or after the treatments? Doesn’t plant species coverage change significantly between the sites? Could be good to show separated by site.

Fig S2: Isn’t one of the OTC “grazed” for each site? I would be helpful to specify which one is grazed for each site. Also, I assume each color is a different year of the study? If so, you need to add a legend.

Table S1: In order to visualize which species are present across multiple sites, it would be better to list all possible species in a column on the left, and then put an “X” next to a species when it is present at a site. That way, a common species will have five “X’s” all the way across to indicate it’s present at all the sites.

Table S3: You don’t need both columns- just include the mean ± SE column. Also, what values are reported? It looks like relative differences between the treatment and control plots (and not actual biomass (g/m2)). If so, this needs to be stated in the table legend. Is there also a table for the warming x grazing treatment?

6. PLOS authors have the option to publish the peer review history of their article (what does this mean?). If published, this will include your full peer review and any attached files.

Reviewer #1: No

Reviewer #2: No

---

## [Author Response · Author response to Decision Letter 0]

3 Nov 2020

Dear Editor,

I am highly thankful for the valuable inputs given by you and the reviewers for the improvement of our manuscript titled “The Response of Culturally Important Plants to Experimental Warming and Clipping in Pakistan Himalayas”

We have responded to all comments raised by the honourable reviewers. We gratefully acknowledge the time taken to provide valuable feedbacks. The helpful comments of the reviewers have contributed commendably for the improvement of the manuscript. We have made all the suggested revisions and a comprehensive yet coherent response is formatted (see below the Comments bold, black and responses to the Editor and Reviewer’s comments in normal italic font, color blue). The revised manuscript is resubmitted here please.

It is pertinent to mention that all key review remarks in 1) the experimental design, 2) statistical analysis and 3) language and grammar; have been incorporated with utmost consideration.

Once again, on behalf of all authors, I thank you very much for your valuable inputs, support and look forward for kind consideration to publish this manuscript.

Editors comment

First, the description of methods has to be improved. It remains unclear how the experiment was designed, thus more details of the experimental design have to be added. For instance, were all the (sub-)plots located within enclosures but clipped to simulate grazing? Was it a two-factorial experiment, with the factors warming and clipping crossed? Clarifying these details is essential to evaluate the appropriateness of statistical methods and the interpretation of results as well as to understand the discussion and the conclusion drawn.

Second, assuming that it was a two-factorial experiment with replicates along an environmental gradient, the statistical analyses (just comparing species within each treatment) would be inadequate. An appropriate analysis will be a two-factorial ANOVA including the main effects of warming and of clipping as well as the interaction between these two factors. This model would then, for instance, allow assessing if the effect of warming depends on whether vegetation was clipped or not (reflected by the interaction term), which is an important scientific question itself. Accordingly, results of the warming x grazing treatment have to be reported and discussed as well. Furthermore, if you consider the site effect to reflect some random variation, you should use a mixed model and include site as random effect. On the other hand, if you are interested in differences between sites, i.e. if you have a specific hypothesis how sites should differ or how the effects of clipping and warming should change along your environmental gradient, you may even include site and its interactions as fixed effects! If you conducted repeated measurements on the same plots, you could either average your data across census dates or should apply a repeated measure ANOVA. The two reviews provide many helpful suggestions how to improve your statistical analyses.

In general, the writing is a bit confusing and vague. The text has to be restructured in parts based on the comments by the two reviewers, and has to be carefully edited for grammar. The two reviewers provided many additional comments and detailed suggestions that have to be considered in a revised version of your manuscript.

Response to Editor’s 1st comment

The treatments include warming (OTC1), clipping (simulated grazing), warming x clipping (OTC2) and control. All the measured plots were inside the enclosure, (also the subplot that was clipped). 

Please see: Section Material and Method, page 9, line 192

Further details are described below as well

Warming treatment was employed using two six-sided open-top chambers (OTC) at each site.1.5 x 1.5m was divided into a central square region of 1 x1 m dimension and six surrounding sloping regions. The central region was used for plant species composition sampling and the area under sloping sides of OTCs for destructive sampling (biomass and plant functional trait estimation). For each data point (OTC, clipped, warmed*clipped, control) the quadrat of 1m2 was placed and plant species cover, and biomass were assessed. The simulated grazing experiment consists of three comparisons: annual clipping plot inside the enclosure where the manual clipping of vegetation was done to the average height of vegetation outside the enclosure in an area that is grazed by herbivores and its control (un-warmed un-grazed) and OTC.

The diagrammatic experimental model of randomized block design (RBD) is given below. 

At each site the field experiment unit (5x5m) comprised of a complete two by two factorial design( warming and clipping) with four treatments: warmed plots(OTC) clipped plots(CL), control plots and warmed plus clipped plot(W x CL). Each experimental unit of 5x5m is divided into subplot of 2.5 x 2.5 m. Open top chambers (OTCs bottom diameter 1.5m a top diameter of 1m were used to increase surface and air temperature since 2015. A 1 x 1m quadrat is placed randomly within each plot. This location was fixed for all sampling years and permanently marked. Percent cover of each species rooted within each quadrat was to be estimated once yearly, preferably at peak biomass. Second year of experiment onwards (after one full season of climate manipulation) above ground biomass was harvested and sorted out on species level, dried and weighed. Biomass was harvested within a plot but outside the 1m x 1m quadrat. In OTC, it was under one of a six sides on rotational basis.

Editor’s comment 2

Second, assuming that it was a two-factorial experiment with replicates along an environmental gradient, the statistical analyses (just comparing species within each treatment) would be inadequate. An appropriate analysis will be a two-factorial ANOVA including the main effects of warming and of clipping as well as the interaction between these two factors. This model would then, for instance, allow assessing if the effect of warming depends on whether vegetation was clipped or not (reflected by the interaction term), which is an important scientific question itself. Accordingly, results of the warming x grazing treatment must be reported and discussed as well. Furthermore, if you consider the site effect to reflect some random variation, you should use a mixed model and include site as random effect. On the other hand, if you are interested in differences between sites, i.e. if you have a specific hypothesis how sites should differ or how the effects of clipping and warming should change along your environmental gradient, you may even include site and its interactions as fixed effects! If you conducted repeated measurements on the same plots, you could either average your data across census dates or should apply a repeated measure ANOVA. The two reviews provide many helpful suggestions how to improve your statistical analyses.In general, the writing is a bit confusing and vague. The text has to be restructured in parts based on the comments by the two reviewers, and has to be carefully edited for grammar. The two reviewers provided many additional comments and detailed suggestions that have to be considered in a revised version of your manuscript.

Response to Editor 2nd comment

Statistical analysis has been improved keeping all the suggestions. We tried to improve our results and presentation to best of our data available in the guidance of editor and reviewers’ suggestions.

Please see: Section Material and Method, page 12, line 249

Further details are described below as well

The species-specific responses within each treatment and their significance was estimated through One-way ANOVA (S3 Table a&b). For each response factor (biomass, percentage cover), Two way ANOVA test was used to estimate the main and interactive effects of experimental warming and clipping. We constructed the linear mixed effect models (fit by maximum likelihood, package lmerMod) was used to determine the parsimonious relationships between the response factors and various biodiversity measures like treatments, plant diversity and elevational gradient (sites as random effect). We assumed the experimental treatments (warming (W), clipping (CL) and warming x clipping (W x CL)) and biodiversity measures (AGB, % cover) as fixed factors whereas sites and plant taxa firstly chosen as random factors and then only site as random factor to check the effect of each treatment per specie. Then susceptibility zones (Fig 4) were determined according to significant species responses depending on the interactive effects of both treatments (W x CL). To assess the change in the relative distribution of plant species within the plant communities, we used non-metric multidimensional scaling (NMDS) ordination analyses [58] on plant species that were present across the plot at each site (3590 m–4696 m) so that we could characterize plant community changes for each warming and grazing treatment groups. The Bray–Curtis dissimilarity measure was used to deal with relative abundance data as it allowed using both presence/absence and abundance data. All data analyses were carried out using the ‘vegan’ R package version 3.5.3 (2019).

Reviewer 1

Major comments

1. General comment

 Comment: The experimental design is unclear. A diagram to include in or replace Figure 1 would help clarify this immensely. It was unclear whether all the measured plots were inside the grazing enclosures, or if any data was collected outside of the enclosures. It was also unclear whether there were four subplots in a factorial design (clipped, warmed, warmed + clipped, control) or some other nested design. Clarifying this is vital for readers to evaluate the statistical methods, interpret the results, and understand the discussion.

Response: 

The diagrammatic experimental model of randomized block design (RBD) is given below.

Please see: Section Material and Method, page 9, line 192

Further details are described below as well

Each experimental unit of 5x5m is divided into subplot of 2.5 x 2.5 m. Where treatments were employed. The treatments were warmed (OTC1), clipped (simulated grazing), warmed*clipped (OTC2), and control. All the measured plots were inside the enclosure, (also the subplot that was clipped). Warming treatment was employed using two six-sided open-top chamber (OTC) at each site.1.5 x 1.5m was divided into a central square region of 1x1 m dimension and six surrounding regions. The central region was used for plant species composition sampling (1m2) and the area under sloping sides of OTCs for destructive sampling (biomass and plant functional trait estimation). For each data point (OTC, clipped, OTC*clipped, control) the quadrat of 1m2 were placed and plant species cover, and biomass were assessed. The simulated grazing experiment consists of two comparisons: annual clipping plot inside the enclosure where the manual clipping of vegetation was done to the average height of vegetation outside the enclosure in an area that is grazed by herbivores and its control (un-warmed un-grazed).

Comment: 

The statistical methods do not seem to be appropriate. It is hard to suggest an exact method that would be appropriate since the experimental design is unclear. However, a two-way model with grazing and warming as separate factors (and potentially an interaction term as well, depending on design) would be a good start. Additionally, we suggest using a mixed-effects model with a random effect for site. The authors repeatedly note that site effects were stronger than the treatment effects and using a random effect for site will help separate these factors. Good packages in R for these types of simple mixed-effects models are 'lme4' using the lmer() function and 'nlme' using the nlme() function.

 For example, if there is no data collected outside the fences, and the experimental design is fully crossed warming and clipping (clipped, warmed, warmed + clipped, control), a good simple model specification (using lmer syntax) would be:

biomass ~ warming * clipping + (1|site.id)

(which is the same as)

biomass ~ warming + clipping + warming: clipping + (1|site.id)

This would then give you an output for the effect of warming, the effect of clipping, the interaction b etween warming and clipping, and with a random effect (intercept) for site identity (i.e.: site A, site B, ...) and you could run this model for each species individually. And similar separate models for cover.

There are more complex ways to approach this as well, including having all the species together in the model with either another random effect (intercept) for each species. However, if you think species respond differently to the different treatments, you will want a random slope, not just intercept. If you choose to go this route, or if there are also data collected outside the enclosure fences and you have a nested design, we recommend consulting a statistician to help with the model design. Again, clarifying the experimental design would really help with this.

An alternate approach if you think the differences between site are based on the altitude gradient, with predictable effects at high sites versus low sites, would be to use ANOVA with interaction terms (using the aov() function), and site altitude as a fixed predictor rather than site id as a random effect. The model specification could be:

biomass~ warming + clipping + site.alt + warming:clipping + warming:site.alt

although you might also choose to incorporate interaction between clipping and site altitude, or even a three-way interaction if you think that's appropriate.

Response:

I would like to appreciate the input given to do the analysis. We have followed each and every suggestion and re-analyzed our results using the Two-way ANOVA, and linear mixed effect models using fixed and random effects.

Please see: Section Material and Method, page 12, line 249,

Further details are described below as well

 First the main and interactive effects of warming clipping and Warm x Clip were analyzed to have the evidence of significance of each factor and most importantly interaction effect on biomass and percent cover.

Please see section supplementary information (data summaries Table S3 Table a &b)

The data summaries of the model are added and described below as well

S3_Table(a) The individual and interactive effect of warming, clipping and interaction between each treatment

Model Aboveground Biomass (AGB) Percentage Cover (PC)

Warming (W) 0.0000(< 2e-16 ***) 0.000(5.18e-14 ***) 

Clipping (CL) 0.000315 *** 0.583

WxCl. 0.014844 * 0.787 

Signif. codes: 0 ‘***’ 0.001 ‘**’ 0.01 ‘*’ 0.05 ‘.’ 0.1 ‘ ’ 1

 Two-way Analysis of variance(2-ANOVA) was used to estimate the main and interactive effects of experimental Warming(W) and Clipping(CL) on the Above ground Biomass(AGB) and Percentage Cover(PC) across all experimental sites during the course of three experimental years(2016-2018). It summarizes individual effect of experimental warming exerted significant response in percentage cover and biomass while clipping’s negative response was not significant.

S3_Table(b) Factor site and species effect on Percentage cover and Biomass

Variables Perc Cover Significance AGB_Significance

Factors(W&CL) 

1.200e-06 ***

P<0.01 

<2e-16 ***

P<0.01

Site 4.933e-05 ***

P<0.01 0.5331

Species .264e-10 ***

P<0.01 <2e-16 ***

P<0.01

Linear mixed effect model summary for each factor considered as fixed to evaluate the overall effect on biomass and cover

Then linear mixed effect model were generated first to evaluate each factor response, determining the fixed (treatments) and random (site + species) and then fixed as (factors and species) keeping the site as random effect.(S3_Table a&b S5_Table c) The factor: species interaction was also analyzed to check species specific response to each treatment and has been graphically represented in the main figure 2B and supplementary figure S3_Figure A and B. The summary statistics of each model is provided in supplementary files(S3-S5Table)

S5 Table(a): Summary statistics of linear mixed-effects model fit by maximum likelihood to determine the fixed and random effects of treatments, and site & species 

Fixed Effects Per-cover Significance AGB Significance

Warming(W) 0.000540*** 4.05e-08 ***

Clipping (CL) -0.000462 *** -0.00625 **

Warming and clipping(W*CL) -4.09e-05 *** -0.02992 *

Random Effects Marginal/Conditional R2 value Marginal/Conditional R2 value

Site and species 0.151 / 0.318 0.205 / NA

Signif. codes: 0 ‘***’ 0.001 ‘**’ 0.01 ‘*’ 0.05 ‘.’ 0.1 ‘ ’ 1. Marginal R2 value represent variance by fixed effects and conditional R2 explained variance by entire model (fixed and random)

The following figures are added in main manuscript as Fig 2 A&B

Fig 2A: Linear mixed effect model representation of significant effects of each treatment (Fixed effect) on(A) percentage cover and (B) biomass. Warming significantly increased the Perc_Cover(A), while the combine effect of Warm and clip significantly decreased it suggesting that clipping has a negative effect on overall percent cover and biomass as compared to control. Random effect is not significant supporting the evidence that each experimental treatment effect is potentially affecting the biomass and cover and have strong evidences as compared to the altitudes (sites)

Fig 2B: Species specific responses towards each treatment by linear mixed effect model 

A) Data showing the percent increase in cover of each species in control, warmed, and clipped plots B) significant change in aboveground biomass of each species in response to control, warmed and clipped plots

Other figures are added in supplementary section as S1 Fig-S4 Fig please refer to the supplementary section

Comment:

There are findings referred to in the discussion that are not included in the methods or results. These need to either be clearly incorporated as main parts of the paper or removed from the discussion. For example, flower production, analyses of community composition, and differences in response based on site location--see more detailed comments in the "discussion" section below.

Response:

The findings which are not included as main results have been removed

Please see: Section Discussion, page21, line 410,

Comment: 

The discussion in general is not clearly organized. It would help a lot if you identified your 3-4 main results, and then gave each of these a paragraph in the discussion (then connecting it to existing literature as you have done).

Response: 

Discussion is re-written included main results and structures keeping in view the valuable suggestions of reviewers 

Please see: Section Discussion, page21, line 410,

Comment:

Please put the figure captions with the figures, it's very hard to evaluate it this way.

Response: Needful done

Comment: 

The data summaries are included but none of the underlying data (the actually collected data points for each plot etc) is available in the supplements or stored in repositories (ex Dryad, Figshare, etc) online

Response:

The data files are submitted for online availability

Minor Comments:

Comment: Abstract: No information about any interaction between grazing and warming. Unclear if the experiment is set up to test this and they didn't find it, or if there were just the two treatments compared to one control.

Response:

Please refer to the abstract section page 1 line 16

which is re-phrased as

The relative effects of climate warming with grazing on medicinally important plants are not fully understood in HKH region. Therefore we combined the indigenous knowledge about culturally important therapeutic plants and climate change with experimental warming(OTC) and manual clipping (simulated grazing effect) and compare the relative difference on aboveground biomass and percent cover of plant species at five alpine meadow site on elevation gradient (4696m-3346m) from 2016-2018. There were significant responses towards main effect of experimental warming on biomass and percent cover (P<0.001) which increased throughout the plant community keeping the site as random. However, the interactive treatment effects (WxCL) were significant on biomass but not on percent cover. These responses were taxa specific. Warming induced an increase of 1±0.6% in Bistorta officinalis while for Poa alpina it was 18.7 ± 4.9% due to its abundance at the site. Contrastingly clipping has marginally significant effect in reducing the biomass and cover of all plant species. Simulated grazing treatment reduced vegetation cover & biomass by 2.3% and 6.26%, respectively, but that was not significant due to the high variability among taxa response at different sites. Thus, warming may increase the availability of therapeutic plants for indigenous people while overgrazing would have deteriorating effects locally. This research illustrates that vegetation sensitivity to warming and overgrazing is likely to affect man–environment relationships, and traditional knowledge on a regional scale.

Introduction

Comment: line 47 – 48 can you state a reference for this?

Response:

Please see section introduction page 3 line 49-51

Added the reference and sentence is re-structured

In the Himalayas, an increase of 3.7 °C is expected in global mean surface temperatures between 2018–2100 (relative to 1986–2005). That is expected to cause substantial effects on the water cycle, biodiversity, and livelihood of the local population [8].

 Kraaijenbrink PDA, Bierkens MFP, Lutz AF, Immerzeel WW. Impact of a global temperature rise of 1.5 degrees Celsius on Asia’s glaciers. Nature. 2017;549: 257–260. doi:10.1038/nature23878

Comment

line 56 this reference is not specifically about grazing; can you find more suitable ones?

line 56 unclear what is meant with “selective palatability issues

Response

Added two suitable references as below.We have added this paragraph that better describes the term palatability, preference, and overgrazing relatedness.

Please see section introduction page 3 line 57-63

Further details are described below as well

The plant species selection by animals as feed is its preference. Grazing animals have sensory impulses stimulation that make them choose their preferred feed, either a plant or its parts which refers to the term palatability. This palatability is affected by animal as well as plant factors such as preference of forage species and seasonal availability of plant. Thus, some plants are selected repeatedly as forage which lead to exploitation by overgrazing. Overgrazing may lead to significant reduction in the palatable species and attribute to selective grazing of highly palatable species that are intolerant to severe grazing and flattening.

Kochare T, Tamir B, Kechero Y. Palatability and Animal Preferences of Plants in Small and Fragmented Land Holdings: The Case of Wolayta Zone, Southern Ethiopia. Agri Res Tech Open Access J. 2018;14.

1Al-Rowaily SL, El-Bana MI, Al-Bakre DA, Assaeed AM, Hegazy AK, Ali MB. Effects of open grazing and livestock exclusion on floristic composition and diversity in natural ecosystem of Western Saudi Arabia. Saudi J Biol Sci. 2015;22: 430–437. doi:10.1016/j.sjbs.2015.04.012

Comment

line 64 you could start a new paragraph after “plant cover reduction”, this would facilitate reading the introduction and adds a clearer structure

Response

Needful done

Comment: 

line 69-71 meaning unclear, For instance, as a result of extensive surveys to evaluate the nutritional status of higher altitude plants, it was proposed that higher altitude plants always had higher N content per unit leaf area when compare with contrasting altitudes a this increased with altitude in herbaceous plants [18]

Response: 

Please see section introduction page 4 line 72-75

Further details are described below as well

Warming and grazing both have substantial impacts on the natural elements of high-altitude ecosystems, with critical consequences for local livelihoods [18]. In cold habitats such as the high Himalayas, leaves are generally rich in nutrients (high N content) to circumvent the detrimental effects of adverse environment on overall physiology of plants [19] These attributes and specific characteristics are making them more desirable and palatable for grazers [20].

Comment

line 75 has previous research dealt with overgrazing or simply with the impact of grazing?

Response:

yes, with warming and moderate grazing also with overgrazing the exploitation of natural biodiversity is studied 

Comment

line 75 -76 add references to the first part of the sentence and specify that diversity in plants in general may be well covered by studies but not so information about medicinal plants. The way the sentence is phrased now does not convey the important message that studies on medicinal plants are still underrepresented

Response:

Please see section introduction page 4 line 81-84

Further details are described below as well

The paragraph is now structured as:

While a great deal of research has been conducted to predict warming and grazing effects on alpine biodiversity [20] , studies on medicinal plants are still lacking and indigenous knowledge is underrepresented where some indigenous communities rely, mostly on traditional system of medicines.

Comment: line 80-81 sentence unclear, explain “benefit-relevant indicators”

Response:

Please see section introduction page 5 line 86-90

Further details are described below as well

benefit relevant indicators are those measures that link any ecological and social outcomes in response to a change in the ecosystem. The paragraph has changed to explain the term more understandably

Comment line 83 – 84 you could delete this sentence, since it does not add important information

Response:

deleted 

Comment: line 85 specify “resource” of what?, f.ex. income, remedies, etc.

Please see section introduction page 5 line 93- 95

Response: the invaluable resources are herbal remedies prepared by local herbalists to earn their bread from also indigenous knowledge that transfers from generation to generations

Comment line: 86 "Second,"

Response: improved the sentence structure

Comment: line 91--would be useful to give a rough metric of historical livestock vs current, instead of just saying it's increasing

Response: 

no such data available though the report of pastures of KNP has rough estimation of >5000 potential grazers. The details are given in Material and methods 138-142

Comment: line 94--these two objectives seem pretty distinct, can you make the connection between them more clear?

Response: 

Please see section introduction page 6 line 110-116

Further details are described below as well

The specific objectives of this study were to 1) indicate culturally and medicinally important plant communities responses to climate warming and grazing factors by manipulative experiments and 2) link these ecological factors to socioeconomic wellbeing of local inhabitants by assessing the dependence and utilization of these resources through ethnobotanical surveys. First to document the indigenous knowledge of culturally important medicinal plants through ethnobotanical surveys then to quantify the effects of climate change (warming) and grazing.

Comment: 

I think in general the introduction could be improved by adding more structure and clarity by defining open questions/research gaps for each paragraph.

I also feel that references are partly missing or are not used in an accurate way

Response

the overall structure of introduction is updated as per your comments and suggestions. All the references are double checked for any missing one and style has been improved

Comment : Materials and Methods

line 104 "July. "

Response: corrected

Comment: line 105 warmest year? warm years?

Response: In a warm year, the summer temp in May has been reported up to 25

Comment: line 106 only few °C below zero or much colder?

Response: 

Please see section Materials and Methods page 6 line 120-125

It was established in alpine zone where climate is defined by warm summers starting from May till August at some lower elevations (3340m) while at higher altitudes it ends in late early August. The maximum temperature in May in recorded the warm year goes up to 25ºC [41] while in winter it drops down much degrees below 0ºC (up to -10 ºC) from October (3590m) [15,42,43]. 

Comment:line 104-107 would be good to know temp and precipitation in the specific study years

Response: It is explained further with more references about the trends of temperature increase in the regions and following paragraph is added

Please see section Materials and Methods page 6 line 126-130. Following references have been added

The current state of knowledge is based on the analyzed climatic trends in HKKH (Hindukush-Karakarm-Himalaya) region from data available since 1980 to 2009 [46]. indicate the rapid declines in the Great Himalaya which attributes to the global warming [47]. The reports of the weather station at Gilgit conclude during last 35 years there has been significant increase in mean annual temperature (average 0.65 ºC) which is higher than global average (0.17 ºC) [48].

35. Bocchiola D, Diolaiuti G. Recent (1980-2009) evidence of climate change in the upper Karakoram, Pakistan. Theor Appl Climatol. 2013;113: 611–641. doi:10.1007/s00704-012-0803-y

36. IPCC 2019. 2019 — Ipcc. 2019 [cited 1 Jun 2020]. Available: https://www.ipcc.ch/2019/

37. Spies M. Mixed manifestations of climate change in high mountains: insights from a farming community in northern Pakistan. Clim Dev. 2019;0: 1–12. doi:10.1080/17565529.2019.1701974

Comment: line 110 "macrophylla, "

Response: Corrected

Comment: line 112 "cattle and yaks" (although I recognize that cattles is also used in many countries, but it's the less common form)

Response: Corrected

Comment: line 114 should be “Plantago lanceolata”

line 115 should be “Poa alpina”

Response: Corrected

Comment: 

line 117 greater than 5026 compared to what? And in what time frame and what is the unit (number of herbivores?)? Also 5026 is a very specific number, maybe just greater than 

5000?

Response: 

Please see section Materials and Methods page 7 line 141-145

The summer pastures of KNP are subjected to grazing management system by indigenous communities because they are rich in herbaceous biomass [5]. In 2013, an analysis conducted to document the changing aspects of above ground biomass in high altitude alpine regions of Pakistan, proclaimed the number and types of livestock’s in KNP pastures featuring more than 5000 species including(sheep, goat cattle yak and others(horse &donkeys)[40].

Comment: Line 118 “immense burden” in what respect?

Response: 

Please see section Materials and Methods page 7 line 146

The structure of sentence is changed 

Comment: line 119 varied rather than "the most versatile"

Response: corrected

Comment: line 119 - 120 need language edits in general

Response: Needful done

Comment: line 128 "useful medicinal plant species are"

Response: corrected

Comment: line 128 are these questionnaires accessible in the SI? If so, please refer to them.

Response: yes, needful done. These questionnaires are accessible in the SI.

Comment: line 119-126 this could be part of the introduction and would nicely support the aim of your study

Response: added in introduction

Comment: line 129 could you add information on how these valleys were different? Given you study the impact of warming and grazing on specific medicinal plants it would be interesting to know the general species abundance of those plants in these valleys. We assume you chose these valleys because they combined people with medicinal plant knowledge and abundance of your studied species?

Response

yes, it’s what our hypothesis is based on. The purpose of conducting ethnobotanical survey was to gather the information about key medicinal plant species. How native people depend on them culturally and medicinally and to find about their abundance in the area. Then selection was based on the most cited plant species by informants and their occurrence on the experimental sites

For species abundance in these valleys’ study area has details of different vegetation zones in KNP and the representative species illustrating their abundance. Following paragraph is added

Please see section Materials and Methods; Ethnobotanical survey; page 7 line 152. Further details are below

Valleys of KNP are characterized by stony beds, water channels (Nullah), vegetation with grasses and large plants like bushes along slopes. Ghunjerab is covered with grasses and had pastures like appearance. Genus Artemisia exist here predominantly. In Shimshal valley, there are some agricultural zones as well and an abundance of sea buck thorn (Hippophae rhamnoides) crop. Plant species particular to family Asteraceae, Brassicaceae, and Eleagnaceae were cited considerably. Based on this initial screening survey, sites for manipulative experiments were designated comprised of the predominant species as excerpted by informants.

Comment: line 141 expression “collection of data the observations” unclear

Response: Corrected

Comment: line 142--would be good to refer back to the altitudinal zones mentioned earlier--seems like this falls primarily into the middle zone (Alpine meadows)? If so, it may not be necessary to give such detail on the other two zones in the study site description.

Response: Improved and un-necessary details are removed as below. The project included the installation of a manipulative experimental set up at five different sites along an altitudinal gradient ranging from 3590 to 4696 m which lies in alpine meadow of the KNP vegetation zones.

Comment: line 143 please also provide more details about the fence material you used i.e. height, mesh size, since this will inform the reader about the dimensions of the potential grazers in your system

Response: The more details about mesh size and height of it is given. 

Please see section Materials and Methods page 8 line 173

Each site consisted of a fenced area (un-grazed) of 5 x 5 m, which was designed as a randomized block design (RBD) and laid out four subplots of 2.5 x 2.5 m (Fig1).

We used standard cattle 1-2 wire fencing (1.7m height with 15x20cm mesh size).

Comment: line 148 your experimental set up seems to focus on summer warming only. Does this align with predications for your study area? Given that your fences allow smaller rodents to pass, how did the OTCs affect those and their impact on the vegetation?

Response:

 Yes, it focused on summer warming only because winters are very harsh, and these sites are covered in snow all over the winter thus access to the area is restricted. Also, in winter, the effect of manipulative warming would have not counted because of harsh weather, snow and wind

We included the investigation regarding summer temperature because all the understudy’ plants were perennial, also these pastures are accessible in summers only because KNP pastures at these elevations are summer pastures. Hence the effect of warming and clipping could be best studied during spring and summer. 

Please see section Materials and Methods page 9 line 191-196

 In general, the fenced area with OTCs acted as enclosure and grazing barrier for animals, however it did not eliminate small rodents and birds. But their effect on the plant destruction was not that significant because not only OTCs but as well as control and clipping plots were also subjected to their infringement. OTCs infact served as additional grazing barrier and they were not removed all the summer season. Grazing effect was induced by clipping only

Comment: line 149 does your observed temperature increase correspond with observations from other studies? Add references.

Response:

Please see section Materials and Methods page 10 line 197-199

It does correspond with the other studies as well as in line with the temperature projections by IPCC 2013. The warming effect of OTC at 10 cm above the soil surface was between 1.7–1.9°C on average[53,54].This temperature enhancement is within the range of expected degrees of warming in scenario of future climate change in the region[55].

Comment:: FIGURE 1 -- the photos are good but (a) should ideally show the grazing exclosures (if you have it) and it would be very useful to have a basic plot layout diagram to clarify the set up. It's unclear what photos d-g are supposed to show.

Response:

Figure 1 is removed and replaced by experimental design showing enclosures

Comment: 

line 151 -- are the clipped subplots nested within the grazing enclosures? or within the open top chambers? Did you actually compare un-fenced areas, or is the clipping the only grazing treatment? That seems surprising since you have potential to do fenced and unfenced comparisons... would want better rationale for this decision. Herbivores can have many affects other than just clipping (dirt impaction, dung, nutrient cycling...)

Response: 

Clipped plots were inside the fenced area. One clipped plot was warmed (OTC*clipping) as one of the two OTCs were clipped. The clipping was done to average height of vegetation in grazing plot outside fenced area which was used as reference to mimic the grazing impact. As there were management strategies about the grazing animals to use the KNP pastures our project funding did not included this factor. The unfenced comparisons were used only to mark the height of plants which were grazed upon. Comparison was done with control (fenced, no clipping, no herbivory) and outside the fenced area plot. Other nutrient addition treatment plot is not part of this study. Because it was not initiated in the first experimental year.

Please see section Materials and Methods page 10 line 200-214

Comment: 

line 152-- no clipping, or no warming? So there were 2 warmed, 1 clipped, and 1 control? Or 2 warmed (1 clipped, 1 unclipped) and 2 unwarmed (1 clipped, 1 unclipped)?

Response: 

Please see section Materials and Methods page 9 line 188

The treatments were warmed, clipped, warmed*clipped, and unclipped (control). All the measured plots were inside the grazing enclosure, (that was clipped).we can say 2 warmed( 1 clipped, 1 unclipped) and 2 unwarmed( 1 clipped and 1 unclipped)

Comment: line 153 vegetative height?

Response: sapling height

Comment: line 154-156 unclear what this means, also how often did you clip the plants, does the clipping treatment mimic the natural grazing impact, did you mark the grazing level on individual plants? Please clarify what you have measured and if your measurements were done before or after the clipping and whether they were repeated during the peak season or several years.

Response:

The clipping treatment was done taking the reference of the grazing level of individual plants. Each year, we measured the height of individual plant and then we clipped and marked it according to the grazing level outside. Measurement were done before (length) and after the clipping and fresh weight was measured in field. After the samples were dried and measure for dry weight for biomass estimation of each species.

Please see section Materials and Methods page 10 line 201-214.

The experiment consisted of three treatments , a 2.5m plot outside fence for herbivory termed as Grazing, one similar dimensioned plot inside fence subjected to simulated clipping treatment in order to mimic the grazing treatment outside and their control(fenced).The clipping was done one time each year to effectively simulate growth. The simulated grazing experiment consists of two comparisons: annual clipping plot inside the enclosure where the manual clipping of vegetation was done to the average height of vegetation outside, the enclosure in an area that is grazed by herbivores and its control (un-warmed un-grazed). Plant height was measured, and clipping was done during the peak season each year (May–Aug). The grazing levels were permanently marked at the starting year and each plant species growth was recorded individually from the clipped point. In the clipped plots the aboveground part of all selected plant species were clipped to about 2-4cm on average above the plot. There was no disruption in the unclipped plots (OTC, control). Most of the selected plant species are perennials they grow well in growing season and shed off their seeds when it ends and sprouting begin at the start of spring each year.

Comment: line 169 the reference you state does not seem to focus on vegetation but on ground dwelling insects

Response: corrected

Comment: line 171 what are the selected plant species?

Response: 

Please see section Materials and Methods page 11 line 227. Further details are added below as well.

The 17 plant species were selected among them are Artemisia rupestris (AR), Poa alpine (PA), Oxytropis glabra (OX), Plantago major (PM), Tamaricaria officinalis (TM), Comastoma pulmonarium (CP), Potentilla hololeucaI (PT), Carex divisa (CD), Astragulus penduncularis (AS), Silene gonospermum (SG), Smelowskia calycina (SW), Primula macrophylla (MC), Hedinia tibetica (HD), Saxifraga (FR), Pedicularis kashmiriana (PC), Bistorta officinalis (BS), Peganum hermala (PH).

Comment: line 174 briefly explain this method in a short sub sentence

Response: 

Please see section Materials and Methods page 11 line 233. Further details are added below as well.

Implementing this method, we commonly measure the cover; defined as vertical projections of vegetation that include the area of a quadrat. In our study we used 1m2 quadrat for species sampling and cover estimation. The percentage cover of the selected plant species was recorded individually at the start of the experiment (2016) by visualizing the numbers of grids of quadrat covered by a plant species.

Comment: line 176 did you sample all occurring species or only those that you previously selecting. If the latter, why were they sampled for classification?

Response: 

Please see section Materials and Methods; Data sampling; page 10 line 236. Further details are added below as well.

Clipping and sampling was done each year, from both OTCs and Control plots. Inside OTC the 1m x 1m plot was designated for species cover and composition. the area along the slopping side was selected for destructive sampling(plant height + biomass).Aboveground biomass was sorted to species level and recorded carefully

Comment: line 179 What was the timing of the clipping relative to the plot surveys? Did you survey before or after clipping?

Response: we surveyed in the initial year of experimentation. That was the first objective of the study. The clipping was done after the surveys of plots

Comment: line 180 why not a two-way ANOVA? are these treatments not interacting with each other?

Response: 

Please see section Materials and Methods page 12 line 249. Further details are added below as well.

We calculated the individual effect of warming (OTCs) and simulated grazing (clipping) on overall plant community and each selected plant species occurring within the experimental site by calculating the biomass and percentage cover of individual sapling.

Comment: line 181 If the treatments are interacting, it should be a two-way ANOVA. If they're not interacting, a Tukey test still doesn't seem like the best fit, since it's generally for different levels of the same type of treatments (ex-low intensity vs high intensity grazing). We have suggested a mixed-effects model as the best option (see major comment #2 above).

Given that you talk about “plant species” did you run separate models for each species?

Response: Our observation consisted of a whole warming effect on plant community that includes these 17 species so aboveground biomass and percentage cover was estimated and model was run for each species where overall and individual response of each plant was estimated (using multiple comparison.Moreover, according to the suggestion mix effect model was run to evaluate the effect of fixed factors(treatments+Percentage cover and Biomass. 

Please see section Materials and Methods page 12 line 252. 

Comment: line 188 "The 'vegan' R package"

Response: corrected

Comment: Table 1 The inputs on what people have observed as climate change effects are really interesting! I think you could maybe highlight this more in your abstract/methods as a key component of the study. It would also be really interesting to directly compare the measured cover/biomass responses of species in the three categories: 1. observed by locals to have increased 2. observed by locals to have not changed in abundance 3. observed by locals to have decreased. Even if this is too small a sample to be statistically tested, it would be a great way to bridge the two components of the study. Also looking at the last two where the locals have observed range shifts--does this fit with the study's findings?

Response: 

There were reports by informants that many low elevation plants are now moving towards the high elevation and they mentioned the Tibet (Hedinia tibetica) quite astonishingly also Pedicularis kashmiriana is high elevation plant but its species observed at low elevation. It was an interesting fact that can link to current scenario of climate change. This supports the general change in specie phenology which is not directly related to this study but could be a supporting information of indigenous knowledge about local plants and future scenario. In discussion the relationship between our findings and survey observation is discussed.

Please see section Discussion page 21 line 406. 

Comment: Is “no effect” and “not observed” the same? What does “forage availability” mean, i.e. due to climate change this species is now available for forage?

Response: 

Please see Table 1

The informants were asked questions about the overall vegetation and then about the plant species they use as medicine or is a representative of the area. Based on this information, then they were inquired about general changes in the availability of those species. They were well aware of the climate warming fact and confident enough to relate it to particular species which enhances the acceptance of the fact that these species are of high importance. Thus, their responses were categorized as no change (no effect), or any change that they might not have seen or not noticed about some species. Due to climate warming more pastures of KNP are now accessible, thus the locals perceived it as forage availability increased. Also, summer extension as main climate change impact, was mentioned and this observation included in a parallel study. Thus ultimately, summer extension may have increased the life cycle of certain plants thus they are present for long period. 

Comment: Just out of curiousity: all species occur at quite high elevations, presumably with snow cover during winter, still some of them are available all year round (f. ex. flowers and leaves of Primula macrophylla) is there simply no snow cover or do these species prevail under the snow?

Response: They were perennial the corrected

Comment: line 227 Saxifraga -- particular species?

Response: This genus was not identified on species level

Comment: Fig 2. not really necessary, can just go into the table

Good suggestion. Added a column in the Table 1 with the name citation frequency (FC).

Fig 3. Looks like the warming and grazing treatments were crossed?

Response: One OTC was clipped and interaction between treatment is included in methods, results and discussion

Comment: line 237 All the more important to include site as a random factor in the model, also did you expect to see differences among warming treatments?

Response: Site as random factor in model is included in statistical analysis

As there were two OTCs per site, we wanted to evaluate the distribution and relative abundance of each specie during the experimental years. 

Comment: Line 240 is the grazing treatment the actual grazing treatment or the clipping treatment?

Response: It is the clipping treatment referring to the simulated herbivory effect in the surroundings of fenced area.

Comment: line 259 This is where a two-way ANOVA with the interaction term would really show these results, not just having to visualize them graphically.

Response: The mixed effect model suggested that the profound effect of warming has significantly increased the percentage cover of species. The analysis has shown the same effect and significance is added to Fig4 

Please see section Results; Fig 4; page 21 line 400.

Comment: line 259-260 this is interesting and novel and should be emphasized more throughout the ms

Response: We further assessed the response to experimental warming and grazing for the five plant species that were most cited by surveyed people (A. rupestris, Astragulus penduncularis, P.alpina, P.hololeuca and P macrophylla). These responses were divided into four categories: combined positive effects of warming and grazing (+W, +G), combined negative effects of warming and grazing (–W, –G) and antagonistic effects (–W, +G or +W, –G) (Fig 4, Table 1).

Comment: line 264 are these results from a model across species?

Response: no, this is just graphical representation of susceptibility, but a cross model would make it statistically correct. This effect has already been shown in mixed effect model

As the results of individual treatment of warming and clipping, we selected most vulnerable species which were highly responsive to the treatment and being culturally important as mentioned in the survey. Based on occurrence of these species on each experimental site and to calculate the combined impact of warming and grazing, the model was run to check their response towards both warming and to put it more effectively we made four susceptibility zones to check the positive and negative interacting impacts of both treatments. 

Comment: line 265 -269 what are the statistics for these results? Are they mentioned in the data analysis part? If not, please add!

Response: Statistics were the same as cover and biomass of each species on each experimental site.

Please see section Materials and Methods page 12 line 255-260. 

Comment: Figure 4 Are these the averages across sites? How do you define "susceptibility zones"? Please add more detail to the methods about this approach

Response: 

Please see section Results page 21 line 400. Also for methods please refer to section Materials and Methods line 258.

Yes, these were average across sites.

Comment: S2 Fig are the different colored points the three years? I think this would be clearer without the lines. Are the non-warmed plots here to compare to?

Response: yes, each color point is experimental year (2016-2018). non-warmed plots are not here. As we wanted to show the relative abundance of each species across the experimental years. The stats are included in the supplementary information (relative abundance).

Discussion

Comment: In general, try to line the discussion sections up with your main results, which will help with the organization of it. In the very beginning you can have a summary of your most important findings, which should not only focus on the ethnobotanical survey, but also on the warming/grazing results.

Response: The overall discussion has been improved and structured as per valuable comments and suggestion. The results are discussed in sections

Comment: line 294 – 296 have you tested this relationship?

Response: No, it is removed

Comment: line 297-299 this is repeated

Response: It is deleted

Comment: line 307 Did you analyze community composition overall? Or just visually with the NMDS? There is a function for this in the 'vegan' package, adonis() for PERMANOVA analysis that would be useful to support this part of your manuscript especially if it is one of your key findings.

Response: We evaluated the relative abundance and tried to visualize with NMDS. It is not one of my key findings. We wanted to show the relative abundance by the collected point inside in response to warming in each year.

Comment: line 309 You haven't mentioned flower production as part of your methods or results--either remove this from the discussion or include it as part of your paper (i.e. in Methods and Results)

Response: removed.

Comment: line 311 Where in your results section do you discuss differences between species responses to warming at different elevations? This would be really interesting, but it isn't clear.

Response: Here, the results of warming are discussed as these species show positive response towards experimental warming by significantly increasing their cover it could be suggested that these species can survive at high altitudes in the onset of climate warming as our experimental sites are along the elevation gradient , and occurrence of these species at altitude and their positive response to the clipping and warming strongly supports the fact that they can adapt to the changing environment provided the controlled grazing.

Please see section Discussion page 25 line 491; Fig S2.

Comment: line 313-314 “new opportunities” this is quite a bold statement, maybe you can put it in context with medicinal plant use

Response: Good comment. The whole structure of discussion has been changed accordingly. Please see Discussion part, page 21 onward

Comment: line 314 to improve the structure of your discussion you could start a new paragraph after ending with “… for people living under cold conditions”. Remove “while” from the beginning of the sentence, otherwise the sentence does not make sense

Response: Removed

Comment: line 318-319 as mentioned earlier it is unclear whether your data are from inside fences only or also from outside the fences. If you did not collect data from outside the fences than it does not make sense to argue with different cattle density. Please clarify this already in the Methods and adapt this sentence accordingly.

Response: we used the unfenced area as a reference which could be used as a model to mimic the grazing effect. Also, the estimation of local herbivory suggested that KNP is rich in herbaceous biomass and an attractive place for local community as compared to the neighboring pastures (Ishaq et al., 2013). Knowing this fact, we proposed that if it continued to be the same situation, soon the flora of KNP will be under the grazing pressure. And will lose its biodiversity, as some species will adapt to this pressure (as shown in this study) other will not (negative impacts of grazing)

Comment: line 321 Here you mention cover collected outside cattle exclusion plots, but it's not clear from your methods or results that this data was collected and how it was collected/analyzed. Please either incorporate it fully or remove it from the discussion.

Response: corrected as suggested

The inside plot (clipping) the impact of grazing did not influence the percentage cover significantly

Conclusion

Comment: line 329 “grasslands” is mentioned here for the first time, so it feels a bit out of place, maybe you can add “alpine”?

Response: added

Comment: line 330 this sentence is nice, though we think an evaluation of ethnobotanical survey methods is not the focus of your study. As plant ecologist we suggest highlighting here plant responses to your experimental treatments and put that into context with the medicinal plants.

Response: Conclusion is improved as per suggestion. Please see page 25-26

Please see section Conclusion page 25 line 507 onward. 

Overall, our study is the first to provide experimental evidence at Khunjerab National Park, of the combined effect of warming and grazing on culturally important plants. Our results provide valuable information for the evaluation and prediction of alpine sensitivity to future threats. Linking indigenous knowledge with experimentation and evaluating the combined and interactive manipulation experiments are likely the advanced tools to examine the responses of alpine community which has proven by our studies it can provide a jumping pad to policymakers for designing mitigation and adaptation strategies for climate change in a region that is undergoing rapid change and for which scientific data are meager. 

Reviewer #2: 

Comment: This study addressed how culturally important alpine plants will respond to climate warming and increased grazing pressure. This is a relevant and understudied topic, and I appreciate the integration of experimental manipulations with data collected from interviews of local people. However, I was confused by the study design- were the grazing treatments simulated grazing (clipping) as stated in the methods (lines 151-152), or grazed by animals, as suggested in the discussion (lines 318-319)? There was mention of grazing enclosures, but it was not clear whether plots were established both within and outside of them. Also, how many plots were established at each site? A diagram of the different sites, plots, and subplots would be helpful in clarifying the study design. Also, were species responses averaged over all 5 sites? The authors state that there are large differences between the 5 sites along the elevational gradient, so it doesn’t seem appropriate to completely ignore this gradient. It seems that the 5 sites were treated as experimental replicates, but they are inherently different. Were there replicate plots at each site, as well?

In general, the methods and results could use much more detail (see line edits below). For instance, wasn’t there a warming x grazing treatment? I don’t see those results reported anywhere- just the separate effects of warming and grazing. Also, the discussion section could provide more synthesis about how the survey data and experimental results complement each other.

Overall, this is an interesting study (great work!), but the manuscript needs more detail and clarity. It also needs to be revised for grammar.

This study addressed how culturally important alpine plants will respond to climate warming and increased grazing pressure. This is a relevant and understudied topic, and I appreciate the integration of experimental manipulations with data collected from interviews of local people. However, I was confused by the study design- were the grazing treatments simulated grazing (clipping) as stated in the methods (lines 151-152), or grazed by animals, as suggested in the discussion (lines 318-319)? There was mention of grazing enclosures, but it was not clear whether plots were established both within and outside of them. Also, how many plots were established at each site? A diagram of the different sites, plots, and subplots would help clarify the study design. Also, were species responses averaged over all 5 sites? The authors state that there are large differences between the 5 sites along the elevational gradient, so it does not seem appropriate to completely ignore this gradient. It seems that the 5 sites were treated as experimental replicates, but they are inherently different. Were there replicate plots at each site, as well?

Response:

Please see section Material and Methods page 9 line 192. Further details are described below as well. 

The treatments were warmed (OTC1), clipped (simulated grazing), warmed*clipped(OTC2), and control. All the measured plots were inside the enclosure, (also the subplot that was clipped). 

Warming treatment was employed using two six-sided open-top chambers (OTC) at each site.1.5 x 1.5m was divided into a central square region of 1x1 m dimension and six surrounding sloping regions. The central region was used for plant species composition sampling and the area under sloping sides of OTCs for destructive sampling (biomass and plant functional trait estimation). For each data point (OTC, clipped, OTC*clipped, control) the quadrat of 1m2 was placed and plant species cover, and biomass were assessed. The simulated grazing experiment consists of three comparisons: annual clipping plot inside the enclosure where the manual clipping of vegetation was done to the average height of vegetation outside the enclosure in an area that is grazed by herbivores and its control (un-warmed un-grazed) and OTC.

The diagrammatic experimental model of randomized block design (RBD) is given below.

The grazing treatments are the simulated grazing (clipping) subplots at each site and herbivory effect was induced by manual clipping of vegetation to the average height of each plant species outside the fenced area serving as grazing treatment. Aboveground biomass was estimated by calculating the plant length before and after clipping and taking the fresh and dry weight of the clipped vegetation. At each site there was one clipping treatment with its control (no clipping-no warming) and a subplot outside the fenced area. The five sites are also experimental replicates. 

Line edits:

Comment: Lines 25-26: The description of the treatments is confusing, and lists 2 separate “controls”

Response: 

Please see section Abstract page 2 line 18-22. 

Improved, there was only one control plot

Comment: Lines 30-31: Is the percentage cover or biomass reported here?

Response: Both percentage cover or biomass reported here.

Please see section Abstract page 2 line 26-28. 

Comment: Lines 67-68: how do nutrients “counteract” the harsh environment- need more explanation

Response: 

Please see section Introduction page 4 line 76-77. Further details are described below as well. 

In cold habitats such as the high Himalayas, leaves are generally rich in nutrients (high N content) to circumvent the detrimental effects of adverse environment on overall physiology of plants [17] These attributes and specific characteristics are making them more desirable and palatable for grazers [18].

Comment: Lines 70-71: This is worded awkwardly

Response: improved

Please see section Introduction page 4 line 76-79. 

Comment: Line 80: Please explain what “benefit-relevant indicators” are

Response: 

Please see section Introduction page 5 line 90. Further details are described below as well. 

Benefit relevant indicators are those measures that link any ecological and social outcomes in response to a change in the ecosystem. The paragraph has changed to explain the term more understandably

Comment: Lines 128-137: Need much more detail about the surveys. What types of questions were asked? Were they oral or written? How were people selected for the survey?

Response: The questionnaire was added in the supplementary data.

The designated team of snow leopard foundation had already communicated with the local community coordinators to inform and enlist people interested in the survey. We focused on elderly people because of their experience and indigenous knowledge. Demographic features are enlisted in supplementary information. Elderly informants’ Formal consent were taken from informants about data collection and publication. The questions were formulated with the purpose of procuring invaluable wealth to knowledge with special emphasis on medicinal and culturally important plants. Questions about important plants their local name, availability, part(s) used, disease treated, mode of preparation of drug, and other cultural used were asked. The questions were asked in local language taking the services of local interpreter

Please see section Materials and Methods page 7 line 151-168. 

Comment: Lines 143-144: How do 2 plots fit inside a 5x5 m fenced area, when a single plot is 5x5 m (containing four 2.5x2.5 m subplots)?

Response: Each site has a fenced area of 5 x 5m which is divided into 2.5 x 2.5m subplots designated to each treatment

Comment: Lines 151-152: The un-closed parentheses make this sentence confusing. What is considered a “control” in this experiment? It would be helpful to assign unique names to all your treatments so that you can remain consistent when referring to them.

Response: 

Please see section Materials and Methods page 9 line 196-204. Further details are described below as well. 

Each experimental site has a warming treatment (OTC) and its control (un-warmed, unclipped subplot placed at 2m distance, clipping treatment (simulated grazing) and a warming x clipping treatment (OTC x Clipping) and a control.

Comment: Line 154: Does clipping (simulated grazing) one time each year effectively simulate the continuous season-long grazing this region usually experiences?

Response: 

Please see section Materials and Methods page 9 line 188-195. Further details are described below as well. 

The experimental sites included summer pastures of alpine meadows. After fencing, the vegetation was observed in both fenced sites and surrounding open sites subjected to continuous grazing during the peak season. The herbaceous summer pastures of KNP are attributed to a grazing management system adapted by local community that allows them to bring their livestock for grazing during peak season which usually starts from late March and lasts till August. It is carefully observed and managed to control the repetitive grazing on same pastures. The grazing levels were permanently marked at the starting year and each plant species growth was recorded individually from the clipped point. All the selected plant species are perennials they grow well in growing season and shed off their seeds when it ends and sprouting begin at the start of spring each year. A summary of number and type of grazing animals in selected pastures of KNP was reported earlier making the sheep, yok, goats and cattle the potential grazers. [51].Thus controlled herds grazing and plant species being perennial effectively simulate the grazing effect.

Comment: Line 169: I know you cite a source for this, but it would be helpful to briefly describe the “quadrat sampling method.”

Response: 

Please see section Materials and Methods; Data sampling page 11 line 231. Further details are described below as well. 

Implementing this method, we commonly measure the cover; defined as vertical projections of vegetation that include the area of a quadrat. In our study we used 1m2 quadrat for species sampling and cover estimation. he percentage cover of the selected plant species was recorded individually at the start of the experiment (2016) by visualizing the numbers of grids of quadrat covered by a plant species.

Comemnt: Line 172: Size of the quadrat grids? Also, I thought the subplots were 2.5 x 2.5 m (not 1 x 1 m)?

Response: Yes the plots were 2.5m the quadrat grid used were of 1m2

Comment: Lines 174-175: Plant height is measured, but not reported in the results.

Response: 

Please see section Materials and Methods page 12 line 244. Further details are described below as well. 

The plant actual height, (length) its clipped length and clipped biomass were recorded each year and growth was measured as g/cm 

Comment: Line 177: What does it mean that plants were “sampled”? Weren’t they just identified to species?

Response: They were clipped for biomass estimation as well from clipped and OTC2 plot.

Comment: Lines 178-179: Were the entire plots clipped? And does repeated clipping of the same location for multiple years affect results? Are plant species mostly annuals or perennials? Were the control and grazing plots also clipped (only the OTC plots are mentioned).

Response:

Please see section Materials and Methods; Manipulative experimental design of warming and clipping; page 9 line 196-222. Further details are described below as well. 

In OTCs, the central 1m x 1m portion was designated for plant species composition analysis, and cover estimation. the area around the sloping sides of OTC was consigned for destructive sampling. The entire plots were not clipped but used the quadrat sampling method, and after clipping the individual species were sorted for identification. Plant species were perennial, and their life cycle usually starts from April to August. The grazing plot inside the fenced area was clipped only. The control served as no warming and no grazing fenced subplot 

The percentage of aboveground biomass and cover doesn’t seem to be affected by repeated the clipping treatment because they were sampled at peak of the growing seasons, each year, and the plant had the sufficient time to survive in presence of enough resources. Also the clipping was done inside the fence imposing a simulated grazing effect as in naturally grazed plot. There was no disruption in the unclipped plots (OTC, control).

In the clipped plots the aboveground parts of all selected plant species were clipped to about 2-4cm on average above the entire plot. Most of the selected plant species are perennials they grow well in the growing season and shed off their seeds when it ends and sprouting begins at the start of spring 

Comment: Lines 180-181: It looks like the ANOVAs were just done to find differences between species within a single treatment (warming or grazing). However, you might want to compare the responses across the different treatments to determine if there are differences between plant growth in the control, grazing, warming, and grazing x warming plots. I find that a much more compelling question than differences between individual plant species. Also, it looks like you compared relative increases of the treatment plots compared to control plots (% increase/decrease), but this is not stated anywhere in your methods. Finally, are responses averaged over all 3 treatments years? Since you sampled the same plots multiple years in a row, you probably need to use a repeated measures ANOVA.

Response: The responses were averaged over all 3 treatment years, the data used in Figure 3 was to evaluate the witn in treatment responses of each species towards main effect of warming and clipping and %increase and decrease calculated by the difference in control plot(2015) in first year and experimental plot in last experimental year(2018).The sampling was done annually each year during blooming season and control plot difference were also measured each year. The statistical analysis now has done using Two way ANOVA to estimate the main and interactive effects of treatment and linear mixed effect model to investigate the fixed and random effects of site, species and treatment.

Please see section Materials and Methods; Statistical analysis; page 12 line 253. 

Comment: Lines 190-192: This sentence is not needed.

Response: corrected

Comment: Lines 192-197: This should be included in the “Ethnobotanical Surveys” section of the Methods. It is out of place here.

Response: moved to Material and Method

Comment: Lines 201-207: Need more details about the results of the surveys

Response: Added the results of survey

Please see section Materials and Methods page 16 line 285-299. Further details are described below as well. 

We interviewed a total of 80 informants who identified approximately 50 medicinal plant species widely spread in the area (see supplementary S1_ Table). Based on ethnobotanic survey results, some highly cited species were selected; those species are palatable, essential for therapeutic remedies and abundant on experimental site thus prone to the change in environment. Predominant botanical families comprised of Asteraceae, Fabaceae 8%, Solanaceae 6%, Tamaricaceae 4%, and others count for 2% or less (see Table 1). We have selected those characterized species that were cited by three or more informants, which is general criterion for reliability of medicinal plants uses[57]. Those interviews also allowed us to collect detailed information about the therapeutic properties of plant species along their altitudinal range of occurrence, time of blooming, part used for remedies, and potential climate change effect on their availability (Table 1).Based on the collected information, we screened out 17 highly important and common plant species, which we categorized based on their percentage of use Artemisia. rupestris (AR) was the most cited plant due to its availability and medicinal potential. Other widely cited plant genera include Poa alpina (PA) and Oxytropis glabra (OX), Taraxacum officinalis (TM), and Plantago major (PM)

In KNP therapeutic treatment is intended as an essential mode of primary health care to heal minor illness. The frequently reported and treated ailments were high blood pressure, diarrhea, skin allergies (40%), followed by eye irritation, fever, gastrointestinal diseases (25-30%) including stomachache, worms’ treatment, acidity (<30%) etc. Problems related to heart, diabetes, joint pains, obesity and metabolism were also reported.(see supplementary S1Figure).In our survey the promising reports of using medicinal plants to treat high blood pressure, gastrointestinal diseases, skin and eye allergies, cardio vascular disease could be due to prevalence of these disease but also the absence of effective and approachable pharmaceuticals. The relative citation frequency indicates the informant’s agreement was high for disease also the uniformity in usage of plant species. 

The information related to other cultural uses of plant species one fundamental use as a feed and forage to livestock and wild animals As KNP is rich in herbaceous biomass, local communities rely on livestock to generate income. The summer pastures are important, and the mentioned species are frequently available there. Due to biodiversity, animals have preference for some species which is determined when livestock selectively nibbling off the new leaves of species. they do not usually feed upon the mature plant. A variety of methods were reported by informants to prepare remedies i.e. decoction, infusion, tincture, juice or ground to power etc. The dominant parts used in preparation of medicine were leaves, flowers or whole plant.

Comment:Line 213: I think a part was omitted since it starts in the middle of a sentence.

Response: Corrected

Comment: Line 234: Since you calculated % increase in the treatment plots compared to the control plots, how did you account for pre-existing variation in plant cover of the different species between the 2 plots?

Response: 

Each site selected for the experiment was relatively homogenous (in terms of

topography, substrate type, vegetation type,), dominated by herbaceous vegetation and representative of the ecosystem. Each site was thoroughly examined to account for

any variation in vegetation types, topographic or orographic factors. In the first experimental year, the species cover, count, abundance, the frequency was recorded before the estimation of the response.

Please see section Materials and Methods page 11 line 231. Further details are described below as well. 

Comment: Lines 259-269: What data is used here? It looks like you are using separate plant responses to just the warming and just grazing plots. Where are the results for the warming x grazing plots?

Response: The interaction treatment results are included in linear mixed effect model. These results were to evaluate the intensity if response magnitude of each species and the difference between them

Please see section Materials and Methods page 12 line 260. 

Comment: Line 295-296: How does the RFC value influence plant availability in the warming treatments? I thought RFC was just an indicator of how often people mentioned the species in the interviews.

Response: 

Please see section Discussion page 23 line 450. Further details are described below as well. 

Here the sentence isn’t clearly structured (improved now). We wanted to build a relation between experimental evidences and survey information by quoting the similar results. This point is highlighted in main manuscript now so it is removed from here

Comment: Lines 297-299: These sentences are repeated.

Response: Removed the repeated sentences

Lines 309-310: This is the first mention of flower production- it is not reported in the methods or results. You can’t introduce new results in the discussion section.

Response: Removed the un-necessary details

Lines 323-324: “A constant increase in species cover through time can occur under moderate grazing conditions” --- Are there sources to back up this claim? Your results do not necessarily support this.

Response: 

Conclusion is rephrased

Please see section Conclusion page 26 line 500. 

Comment: Lines 334-338: Citations for this? Also, this would be good to mention earlier, maybe in the introduction.

Response: Removed from here and mentioned in introduction

Tables & Figures:

Comment: Table 1: You should specify which information is from the surveys and which is from another source (which should be cited). Also, how are the “most preferable” forage species determined?

Response: The information related to other cultural uses of plant species one fundamental use as a feed and forage to livestock and wild animals As KNP is rich in herbaceous biomass, local communities rely on livestock to generate income. The summer pastures are important, and the mentioned species are frequently available there. Due to rich biodiversity, animals have preferences for some species which is determined when livestock selectively nibbling off the new leaves of that species. They do not usually feed upon the mature plant.

Please see section Results; Table 1 page 15 line 277 and 309-314. Further details are described below as well. 

Comment: Fig 1: These pictures are lovely, but I think they are mostly unnecessary. Maybe just panel b showing the OTC. A better figure would be a diagram (or annotated picture) of your plots and study design.

Response: corrected as per suggestion. Please see response 1

Comment: Fig 2: It might be nice to color the bars by plant functional group (shrub, forb, grass), so someone unfamiliar with the species can at least know the growth form.

Response: Included the information in the Table to support the RFC value as suggested by the reviewer 

Comment: Fig 3: The figures are very fuzzy, so it is hard to read, but I don’t see any letters showing statistical significance as stated in the legend. Also, what do the geometric shapes around each error bar show? You need units for the y-axis: Biomass (% change).

Response: 

Please see section Results; Fig 3 page 20 line 383. Further details are described below as well. 

The figure quality is improved and statistical significance is added. As only warmed plot showed the significant result of increase. These are pirate plots which has following components. Points show raw data, the geometric shape called bean show the smoothed density full data distribution. The biomass was calculated as growth g/cm the axis is labeled(plant height was measured before and after clipping and clipped biomass was weighed (fresh/dry)then growth was calculated as g/cm in initial year and g/cm in last experimental year)

Comment: Fig 4: What does each point represent? A different site? A different year? Also, similar to my comment above, what data is shown? I don’t see how you can report these 4 zones if just showing the response to the combined warming x grazing treatment.

Response:

Please see section Results; Fig 4 page 21 line 408 and 309-314. Further details are described below as well. 

 Each point represents the %increased/decrease in species specific response to warming and clipping at different site. +W¬-G means main effect of warming in absences of grazing.+W +G is the combined effect which represent the interaction between treatment and response of these species.

Comment: Fig S1: The colors in the pie chart (panel b) are impossible to distinguish- there are way too many categories. Consider an alternative format. Also, I’m unclear as to what is shown in that chart- is the “study area” across all 5 sites? And before or after the treatments? Doesn’t plant species coverage change significantly between the sites? Could be good to show separated by site.

Response: The idea was to just illustrate the cited plant families, now this chart is removed and main families are reported in to main text

Please see section Results; page 16 line 289. 

Comment: Fig S2: Isn’t one of the OTC “grazed” for each site? I would be helpful to specify which one is grazed for each site. Also, I assume each color is a different year of the study? If so, you need to add a legend.

Response: yes, OTC2 was clipped, each color is the year of experiment. the legend is added

Please see section Supplementary information; Now as Fig S

S1 Table: In order to visualize which species are present across multiple sites, it would be better to list all possible species in a column on the left, and then put an “X” next to a species when it is present at a site. That way, a common species will have five “X’s” all the way across to indicate it’s present at all the sites.

Response: Its done and described below as well. 

Please see section Supplementary information; Now as Table S2

Comment: S3Table: You don’t need both columns- just include the mean ± SE column. Also, what values are reported? It looks like relative differences between the treatment and control plots (and not actual biomass (g/m2)). If so, this needs to be stated in the table legend. Is there also a table for the warming x grazing treatment?

Response: Only one column of mean± SE is added. The %increase in cover and biomass , compared to control, in response to elevated warming, is recorded here. The objective was to demonstrate the within treatment species-wise variation to predict the occurrence. The interactive treatment effects are included in the main mixed effect model data summaries.

Please see section Supplementary information; Now as S4 Table

---

## [Decision Letter · Decision Letter 1]

10 Dec 2020

PONE-D-20-21262R1

The Response of Culturally Important Plants to Experimental Warming and Clipping in Pakistan Himalayas

PLOS ONE

Dear Dr. Ali,

Thank you for submitting your manuscript to PLOS ONE. After careful consideration, we feel that it has merit but does not fully meet PLOS ONE’s publication criteria as it currently stands. Therefore, we invite you to submit a revised version of the manuscript that addresses the points raised during the review process.

First of all, we acknowledge the substantial improvements of the manuscript in response to previous reviewers' comments, however there are still various issues that have to be considered in a thoroughly revised version.

For instance, you applied a variety of different statistical approaches for the same research questions, which makes the interpretation of results difficult and confusing. You may wish to focus on the approach that is most appropriate for a respective research question or hypothesis, and show only tables and figures related to that particular approach. The reviewers provide various further recommendations how to improve the clarity of figures and tables.

In addition, the storyline, or direction, of your manuscript is still a bit unclear. For instance, you could have stronger focus on how your experimental results provide information about the future abundance of culturally important plants in your region. Furthermore, I recommend considering the detailed recommendations and suggestions by the reviewers how to improve the introduction, the methods section, the results and also the discussion.

Finally, the whole text would benefit from a thorough edit for grammar (including typos) as well as more clarity.

We look forward to receiving your revised manuscript.

Kind regards,

Harald Auge

Academic Editor

PLOS ONE

Reviewers' comments:

Reviewer's Responses to Questions

**Comments to the Author**

1. If the authors have adequately addressed your comments raised in a previous round of review and you feel that this manuscript is now acceptable for publication, you may indicate that here to bypass the “Comments to the Author” section, enter your conflict of interest statement in the “Confidential to Editor” section, and submit your "Accept" recommendation.

Reviewer #1: (No Response)

Reviewer #2: (No Response)

2. Is the manuscript technically sound, and do the data support the conclusions?

Reviewer #1: Yes

Reviewer #2: Partly

3. Has the statistical analysis been performed appropriately and rigorously? 

Reviewer #1: No

Reviewer #2: No

4. Have the authors made all data underlying the findings in their manuscript fully available?

Reviewer #1: No

Reviewer #2: Yes

5. Is the manuscript presented in an intelligible fashion and written in standard English?

Reviewer #1: Yes

Reviewer #2: No

6. Review Comments to the Author

Reviewer #1: The authors' revisions have improved this manuscript substantially, especially with the clarification of the experimental set up and much clearer flow in the introduction and discussion. We appreciate all their work. In its current form, this manuscript still needs substantial edits to be ready for publication. Most importantly, the statistical approaches need to be focused and clarified--we suggest ways to do that.

Our major suggestion is that you focus on one statistical approach, and highlight the model output and related figures from that approach. At the moment you have added the approaches suggested by the reviewers, but as a result have ended up with multiple overlapping modeling approaches. This makes the results difficult to interpret, and also an overwhelming number of supplemental tables and figures. The most appropriate one seems to us to be the linear mixed model with the random effects for site and species, which is discussed in the methods and highlighted in Figure 2 and Supplemental table S5a. Having both the ANOVA and the mixed model for the same response variables is redundant and causes confusion. In our following comments we highlight ways that you can focus on and further clarify this statistical analysis and the results, as well as other more minor areas in the manuscript that need some revision.

Other general comments:

Plant height is mentioned in the methods as data you collected, and again in the discussion. However, it is not included in the statistical analysis methods or in the results. You either need to clearly explain how you analyzed height and what the results were or remove it from the manuscript.

Check for typos in the main manuscript (species needs the 's' at the end for both one species and many species, authors names should be capitalized, other small typos) and the supplementary information

Figures

Figure 1 is very helpful but is still somewhat unclear. For a start, please add the more detailed information you gave in your responses to the reviewers on the plot layout to the figure caption--the current caption is insufficient. Also, please make the various rectangles areas roughly to scale (the 1 x 1 m sampling plot inside of the 2.5 m x 2.5 m area is much bigger than it should be). Additionally, it's unclear what the smaller extra "control plot" areas adjacent to the OTC diagrams are supposed to be, or what they were used for--are those part of your study and sampling design? It would be really helpful if you could mark where one year's biomass sampling would have been conducted, since this is still somewhat unclear in the methods--for example by shading the relevant area in grey. Finally, it's not clear what the "Species composition" labels are supposed to indicate.

Figure 2 seems to us to be the most useful results figure, and just needs some small edits. The subplots should be labeled A, B, C, D not A.A, A.B etc for clarity. Specify in the figure caption if the error bars show standard error, standard deviation, or something else. Also, the *** significance marks are not clear--do these show differences from the control? from each other?

Figure 3 this is mostly overlapping information with the second half of figure 2, and should be removed or moved to the supplement.

Figure 4 is good as it is

Supplemental Figure S1 good as-is, although it would be clearer to write out "percentage" rather than "%age"

Supplemental Figure S2 the order of the treatments in A and B is different, which is very confusing when looking quickly at the figure. Please change the order for S2B to match the order in Figure 2 and S2A. Please add more information to the figure caption with what tests etc were used to produce the figure.

Supplemental Figure S3 is redundant with Figure 2 and can be removed

Supplemental Figure S4 needs information in the figure caption as to what the different colors represent. Also, the caption says "The relative abundance of plant species (presence/absence)" Which is it? Abundance measures are generally biomass or maybe cover, while presence/absence is a binary.

Tables

Supplemental table S2 This would be better to have one column for each site, and mark which species occur where rather than just tallying the number of sites they occur at. "Understudy" (used here and elsewhere in the ms) is unclear--do you mean understudied, that they have not been paid attention to in the literature?

Supplemental table S3 we recommend focusing on the linear mixed effects model and dropping the ANOVA approach in the methods, results, and here. For that reason, we suggest deleting these tables.

Supplemental table S4 is good as it is

Supplemental table S5. We suggest moving Table a to the main manuscript, and including F-values rather than just p-values. These can be gotten by using the summary() function on the model fit. Table b should be dropped -- it's better to have the interaction between the treatments than combining them in this way. Table c can be kept in the supplement.

Introduction

Line 56 – 61 These information are all good, but go beyond scope of the introduction. From my point of view, the essential information would be contained in a sentence like “ Repeated grazing can lead to exploitation and reduction of highly palatable species which are less resistant to severe grazing”. I still think that “selective palatability issue” is little unclear. Maybe instead of using this term you can rephrase it into the issues you are talking about. For example, do you mean that grazing leads to a change in forage species availability grazing animals can chose from? You could say “This leads to changes in plant availability that grazing animals can feed on and also to changes in nutrient availability”. Nutrient availability of what? Leaves?

Line 85-86 Maybe you can add a reference to this sentence

Line 88-89 Even though you nicely improved this sentence, I still don’t have any idea about what these indicators are, maybe you can provide an example of such an indicator or explicitly write what are the indicators in your study. This information is essential for readers who have not been working with these kind of indicators before.

Line 94 unique richness of what? Plants?

Line 108 you may mean “investigate” instead of “indicate”?

Line 110 “factors” is not necessary

Line 112 -114 This could be deleted since the information are already given in the objectives before. If you want to keep it, change the order so it matches the order of the objectives.

Methods:

Line 119 km2, lowercase k

Line 120 replace “it” with “the study”

Line 121 replace “it” with “summer”

Line 120-121 simplify this to just "from May till August" since it's also August at higher elevations

Line 126 declines of what?

Line 129 "The alpine KNP region includes the following..."

Line 141 space missing between “biomass” and “(“

Line 148 -151 The exact same sentences appear earlier in the introduction, you might want to delete them here to avoid unnecessary redundancy?

Line 153 Needs to refer to a specific table or part of the supplementary information (currently Supplement S6?). It may also be helpful to have one supplementary document with figures and tables and another one for the questionnaires only. This you could for example name supplementary information II, so the interested reader directly knows where to find the respective information.

Line 161 -163 This is phrased little awkward, I assume you mean that those species were abundant in your experimental sites? I suggest to rephrase this sentence.

Line 163 This would be a good place to add more information on what main questions you asked about each plant, since most of your readers won't want to be searching through your supplement. Something like "For each species, we collected asked informants about... [climate change responses, etc]"

Line 206 -209 I suggest that this entire part could be integrated into the data sampling part

Line 216 Why to simulate growth, isn't the intention to simulate herbivory? Or do you mean you want to observe growth under herbivory?

Line 219 -220 For consistency it would be good to move the sentence about the height measurements to the data sampling paragraph

Line 257- 260 In the first part you write that you analyzed species-specific responses, but the tables show model results across species. I suggest you either adjust this in the text or refer to a table/graph with species-specific results. Furthermore, table b in S3 does report results of linear mixed effect models (according to the subscript of the table), this is confusing, because you don’t mention linear mixed effect models when you refer to the table in the supplementary information. Also related to table b in S3, it is not clear to me whether site and species were used as random or fixed factors in the model. In the text in your ms you report them as random factors, but in the table they seem to be reported as fixed factors. Also, you only report the interaction of W x CL but no main effects. It would be good to include these in the table and also specify which test you used to assess the significance of the treatments like F- or ChiSq test. You can access these tests after you have run your mixed effect model via the Anova command from the “car”-package: Anova(mymodel, test.statistic = “F”) for F-test. Instead of using the summary statistics as you now did in table S5 you can instead report F-test results. Related to table S5 do you have an explanation for why the conditional R2results in NA when percentage cover is the response variable?

Line 261 Please add a reference to the R package you used. The most common packages to run linear mixed effect models are either lme4 (Bates et al. 2015) or nlme.

Line 264 -267 This is a key section that needs to be clarified. First, I think your response variables were the same as in your ANOVA models e.g. biomass and percentage cover, so these should not be termed as fixed factors but as response variables. Second, neither biomass nor percentage cover are metrics of diversity! It can be helpful to actually include the model formulation here, which I think what you're trying to say is:

cover ~ warming * clipping + (1|site) + (1|species). The best way to describe this model would be something like "Response variables of percentage cover and biomass were modeled using [xx package] with fixed effect predictors of the warming treatment, the clipping treatment, and their two-way interaction, and random effects for site and species (Table xx). We separately ran models with species as fixed effect to check the effect of treatment for each species (Table xx)"

Line 267 I suggest starting a new paragraph for each different analysis you conducted--one for the linear mixed model, one for the susceptibility zones, one for the NMDS. I think you also need more information about this "susceptibility zones" figure 4--why did you do it, how did you do it, what is it intended to show?

Line 275 You should also report which version of R or RStudio you used (latest version in R is 4.0.3). You can f.ex. write: We used R (RStudio) to run all analyses (R Core Team 2020 version XXX). For the vegan package you have to cite (Oksanen et al. 2017). There is a neat function in R called citation() which shows you how to cite the packages you used

Results

Line 321 - 328 This section with the ANOVA can be removed if you focus on the results from the linear mixed effect model. Also the numbers reported here don't match the p-values in S3 Table B, shouldn't they be consistent with that?

Line 331 - 343 The language here makes it sound like you are testing the effect of the site "impact" rather than just variation. Also words like "increased" and "changes" make it sound like you're looking at the change through time, but your methods indicate you're just looking at the differences between treatments. Instead you should use statements like "has higher cover than the control" to match the analyses you're doing.

Line 337 It is advisable to avoid “seem” if you have statistical evidence for something having an effect.

Line 340 Negative? I thought it was positive--higher in the warmed plots?

Line 341 How can you state this, if you have pooled the data across years? Did you run separate models for separate years or did you include year as a fixed factor in your mixed effect model? You would need to either explain these models in your statistical analysis section and include them as ex supplemental tables, or remove this part.

Line 344 You should have 1-2 more sentences here on the species-specific results that are presented in Figure 2B

Line 347-351 This sentence is not clear to me. Do you intent to say that sites did not significantly increase the variation in your data? Assuming you had site as a random effect in your model.

Line 355 You can compare the responses in your treatments plots to the control, but usually control is not considered a treatment. It would thus sound better if you wrote something like "...in above ground biomass of each species in response to warming, clipping and warming x clipping compared to control.

Line 378 Should there be a new paragraph here from the NMDS results?

Line 380 Isn't this supplemental figure S4?

Line 382 This supports the choice to incorporate site as a random effect in your mixed effects model

Line 403 Delete “for”

Discussion

In your discussion it would be really nice if you could further emphasize the connection between your experimental findings and the ethnobotanical survey.

Also, instead of AGB just say biomass, it doesn't take much more space and is much easier to read.

Line 418 You did not really measure community data, but rather species-specific warming and clipping responses.

Line 419 - 420 The commas before and after "was" should be removed, and there should be a comma added after "sites"

Line 425-427 In your methods section (the statistical analysis) you don't report that you have used height as a response variable, nor do you report it in any of your results. If you did, you should add it, otherwise you cannot discuss it here.

Line 429 I think you are combining two plausible mechanisms here: competition for resources and limitation in space. You might want to elaborate on both of them. Check ex. Borer et al. 2014 in Nature

Line 463 "Other studies on alpine meadows [where?] also suggested..."

Line 469 Since intensity and frequency of clipping should have been the same for all of your studied species I think it is rather something else that drives this species-specific response. You could think about whether this species has specific traits allowing it to regrow fast after clipping/grazing which the other species might not have.

Line 473 "clipping" not grazing

Line 478 "stronger" not "sturdy"?

Line 490-492 This sentence is not clear to me. Do you want to state that the species you studied are more susceptible to warming than to grazing?

Line 493 Wang et al. add the year

Line 494-499 The first part of this part is about the impact of clipping, but you then refer to results of your survey in relation to warming. Please, clarify this. Also, can you state references for the loss of biodiversity. There are plenty of good ones available!

Line 501 You haven't reported any measures of diversity related variables like species richness, so speaking about biodiversity isn't appropriate here.

Reviewer #2: The authors have made substantial improvements, especially with describing the experimental design and survey methods. However, while there have been clear attempts to improve the statistical methods and results, I’m unclear about which approaches were used and why. It seems the authors used several different models (one-way ANOVA, two-way ANOVA, mixed models with different random factors), but I don’t think they are all necessary. The model interpretations in the results section are confusing, and do not clearly align with specific experimental questions. Instead, the authors should decide which questions are most interesting and choose 1-2 statistical methods based on that. Also, I’m unclear on how the % increase/decrease values were calculated (lines 250-252) and whether they represent the entire 3-year experimental duration. I still think it could be best to use repeated measures with year as a variable in the statistical model to see how the warming and clipping effects built up (or not) over multiple treatment years. Also, there is some discussion about differences between elevational plots (i.e., lines 333-335, 500-505) but it’s unclear how that was formally assessed.

The survey and experimental data are better integrated in this revision, but the paper still lacks a clear storyline. There needs to be more discussion around how the experimental results provide information about the future survival/abundance of culturally important plants in the region. The figures also need refinement (see comments below), and the entire manuscript needs to be edited for grammar and clarity.

Lastly, this might be the journal format, but I found it extremely difficult to review the figures and supplemental figures when they were all separated from their legends (figure legends embedded in text, figures at end, supplementary legends at end, supplemental figures had to be individually downloaded).

Again, I believe this is an interesting study, but the manuscript still needs improvement.

Lines edits:

Lines 44-46: This is unclear. Land and water degradation does not “cause” climate change.

Line 49: Not “global” mean surface temperature if just referring to the Himalayan region.

Lines 85-86: I appreciate that “ecosystem change assessment” is defined, but the definition is still unclear. Also, the rest of the paragraph doesn’t mention these “ecosystem change assessments,” so why is it brought up here?

Line 106: “primarily runs the financial circle” is unclear

Lines 112-114: This last sentence is unnecessary

Line 123: Incomplete sentence

Line 133: Is “yoks” supposed to be “yaks”?

Lines 144-151: This was all stated in introduction already, so is not needed here.

Lines 156-160: Put this description of the valleys into the previous section (Study Area).

Lines 161-162: Put this sentence into the next section (Warming and Grazing Experiments)

Lines 167-170: The RFC description should be included in the “statistical analyses” section instead.

Lines 171: What does “we involved plant material” mean?

Lines 182-183: What does “which lies in alpine meadow of the vegetation zones” mean?

Line 184: Since there was only 1 replicate for each elevation, the entire experiment is an RBD, but not each site.

Lines 186-188: Were the data collected outside of this fenced area used for any analyses? I don’t see it used.

Line 191: Where was this reported earlier? Or do you mean it was reported by another prior study?

Line 197: How can a hexagonal shape have the dimensions of 1.5 x 1.5 m or 1 x 1 m? This reporting style is reserved for squares/rectangles. Do each of the six sides have that length? If so, that needs to be stated.

Line 198: Since your fences prevented grazing, they should be called “exclosures” instead of enclosures throughout the paper.

Lines 212-214: This is confusing.

Line 214: Define the abbreviation “W x CL” the first time it’s used.

Line 216: Simulate “grazing” (instead of “growth”)?

Lines 218-219: What is described here? I don’t understand the 2 grazing comparisons. The analyses only discuss the manually clipped plots.

Line 220-221: This is confusing.

Line 224-225: Remove this sentence.

Line 132: How large were the quadrats and the quadrat grids?

Line 234: How were the “selected plant species” determined?

Lines 236-241: Put all the species into a table and reference it instead.

Lines 243-244: What does “vertical projections of vegetation” mean?

Line 248: How large were the clipping areas? Were the same areas clipped multiple years in a row?

Line 250-252: I’m unclear about what you used to calculate % increase/decrease for the data. If the values were averaged over all 3 years, how was % increase/decrease calculated as the “difference between control and last year data points”?

Lines 155-156: This sentence not needed here.

Line 290: Missing % for Asteraceae

Lines 291-293: Unclear

Lines 301-302: This sentence not needed.

Lines 306-308: Save for discussion section

Lines 308-309: This is confusing.

Lines 313-315: This sentence not needed here.

Line 321: What does “across all three growing seasons” mean in this context? Average over 3 years? The value at the end of year 3? Each year assessed independently (then year would need to be included in the statistical model)?

Line 425: Plant height was not statistically analyzed or discussed in results section, so should not be mentioned here.

Lines 425-428: Unclear

Line 456: State author’s names (instead of reference number) if specifically referring to a study.

Lines 490-494: Unclear

Lines 496-497: How does the data support this assumption?

Lines 500-505: I like the biodiversity discussion. Expand on this more.

Figures:

Figure 1: The figure legend needs to include much more detail – it needs to fully describe each component of the plot design. For example, why are there control plots in the warming and warming x clipping larger plots? What do the letters on the OTC mean? Unclear what the 1 m and 1.5 m arrow are pointing to.

Figure 2A: First, these should be bar charts or box and whisker plots instead of line graphs. Line graphs are usually used to represent changes through time. What do the stars signify? Panel A needs a better y-axis label (no shortened words, underscores).

Legend line 348-349: Is there statistical evidence for this claim? Line 349-351: This is not needed here.

Figure 2B: First, why is this labeled “2B”? It appears to be an entirely separate figure, so should be “Fig 3.” What do the stars and dots represent? The axis labels should be written without underscores. Also, why does it appear that most of the control plots increased? How is it that none happened to decrease over the 3-yr period?

Figure 3: You explain in your review response what the “beans” represent, but this description needs to be included in the figure legend. It would be helpful to label each panel as “warming” or “clipping” depending on what it shows. It might also be nice to identify (maybe by color) which species are the most culturally important- it ties this data to the survey responses that way. Again, there are still no letters indicating statistical significance (as stated in the legend). The stars show statistical significance of each bar compared to what- each other?

Figure 4: What do the stars mean?

Figure S1: Improve y-axis label

Figure S2: Similar to figure 2A, these should all be bar chars or box plots. Improve axis labels. For the right panels, re-order species to group by cultural importance or functional group. Much more detailed legend needed.

Figure S3: Omit or re-structure this figure- it’s unreadable.

Figure S4: What do the different colored dots represent? It’s hard to tell which labels go with which points. I’m not familiar with these types of plots, but this is confusing to me. Maybe put more description in the legend?

Table S2: Thanks for editing this table. One suggestion – make separate columns for each elevation, and then put an X in each column where a species is present at that elevation. Right now, you can’t tell which species are present at which elevation.

Table S3 & S5: Include the full ANOVA tables/statistical output. More than just the p-values should be included.

Data points: Need metadata- full description of what each dataset it and the meaning of each abbreviation.

7. PLOS authors have the option to publish the peer review history of their article (what does this mean?). If published, this will include your full peer review and any attached files.

Reviewer #1: No

Reviewer #2: No

---

## [Author Response · Author response to Decision Letter 1]

2 Feb 2021

Authors response to the Editor and Reviewers

 Title: The Response of Culturally Important Plants to Experimental Warming and Clipping in Pakistan Himalayas

 Authors: Saira Karimi, Muhammad Ali Nawaz, Saadia Naseem, Ahmed Akrem, Olivier Dangles, Zahid Ali1*

Article Type: Original Study

 Dear Editor,

Please find enclosed the responses to all comments raised by the reviewers for our manuscript “The Response of Culturally Important Plants to Experimental Warming and Clipping in Pakistan Himalayas” to be considered for publication in PLOS-ONE. We gratefully acknowledge the time taken by the honorable reviewers to provide valuable feedback. The helpful comments of the reviewers have contributed commendably for the improvement of our manuscript. We have made all the suggested revisions and comprehensive, yet coherent responses are formatted (see below the responses to the Editor, and Reviewer’s comments in italics). It is pertinent to mention that all key review remarks in: 1) the statistical approaches have been narrowed down to recommended approach suitably, 2) The direction and storyline of the manuscript is defined further, and the connection of survey results to experimental pieces of evidence has been built-up and explained in a clearer fashion 3) the language and grammar have been improved further with the utmost consideration. Figures are also updated as suggested by both the reviewers. Here I would like to mention that about Figure 2, reviewer #1 accepted it as it is, in line graph and recommended minor edits. We have done all minor edits recommended by reviewer #1 for this Fig, however reviewer #2 suggested to change it into box and whiskers plot with suggestion that line graphs represent changes over time. It is humbly mentioned here that this statement supports to our study data too, we also studied climate warming impacts over a period of 3-years. Thus, we request to consider the Fig 2 in line graphical form. Figure 2 as box and whisker plot is also submitted as a separate file and with the rebuttal letter just for your perusal.

We have fulfilled all necessary requirements and I on behalf of all authors thank you very much, and request to kindly consider the manuscript for publication.

Sincerely,

Zahid Ali (Ph.D.)

Dr. rer. nat.

Department of Biosciences

COMSATS University Islamabad, Islamabad campus

Pakistan

E.mail: zahidali@comsats.edu.pk

Editors Comment 1: For instance, you applied a variety of different statistical approaches for the same research questions, which interprets results difficult and confusing. You may wish to focus on the approach that is most appropriate for a respective research question or hypothesis and show only tables and figures related to that particular approach. The reviewers provide various further recommendations on how to improve the clarity of figures and tables.

Response to Editor

We agree to this point about the statistical approaches that have been addressed in this revised version of the manuscript. Among all the analysis that were suggested by the respected editor and reviewers, the most appropriate analysis which describe our result most effectively is Linear effect model. The tables and figures related to only this model are shown and all other overlapping data is removed. 

Please see statistical analysis in section Material and Methods page 11 line 228

result section see page 16 line 288

Comment 2: In addition, the storyline, or direction, of your manuscript is still a bit unclear. For instance, you could have stronger focus on how your experimental results provide information about the future abundance of culturally important plants in your

region. Furthermore, I recommend considering the detailed recommendations and suggestions by the reviewers how to improve the introduction, the methods section, the results, and the discussion. Finally, the whole text would benefit from a thorough edit for grammar (including typos) as well as more clarity.

Response to Editor

We have clarified the focus of the manuscript and improved it in the direction which can relate our experimental results to the future abundance of medicinally important plants in the current scenario of climate change at Gilgit Baltistan. Moreover, the valuable comments and suggestions of both the reviewers have been considered and addressed carefully.

Please see section Discussion page 25 line 490-491

Reviewer 1 

General Comment 

Our major suggestion is that you focus on one statistical approach and highlight the model output and related figures from that approach. At the moment you have added the approaches suggested by the reviewers, but as a result, have ended up with multiple overlapping modelling approaches. This makes the results difficult to interpret, and also an overwhelming number of supplemental tables and figures. The most appropriate one seems to us to be the linear mixed model with the random effects for site and species, which is discussed in the methods and highlighted in Figure 2 and Supplemental table S5a. Having both the ANOVA and the mixed model for the same response variables is redundant and confuses. In our following comments, we highlight ways that you can focus on and further clarify this statistical analysis and the results, as well as other more minor areas in the manuscript that need some revision.

Response to General comment

This useful suggestion has been adapted and results of linear effect model are shown in Material and Method as well as results and discussion section All other overlapping information and statistical approaches (one-way ANOVA, two-way ANOVA results) are removed

Please see statistical analysis in section Material and Methods page 11 line 228

result section see page 16 line 288

Other general comments:

Plant height is mentioned in the methods as data you collected, and again in the discussion. However, it is not included in the statistical analysis methods or in the results. You either need to clearly explain how you analyzed height and what the results were or remove it from the manuscript. Check for typos in the main manuscript (species needs the 's' at the end for both one species and many species, authors names should be capitalized, other small typos) and the supplementary information

 Response:

The plant height was used to measure the growth/cm along with biomass (see supplementary information data points) However, it is not the main result so has been removed. The typos and grammar mistakes have been carefully omitted.

Following key changes have been made in figures and Tables

Figures 

Figure 1: More detail is added in the caption to best illustrate the experimental design and explain the sampling sites for species composition and biomass estimation. The schematic diagram has been improved with more clarification for plots and subplots.

 Please see Section Results page 10 line 202

Comment: Figure 2 seems to us to be the most useful results figure, and just needs some small edits. The subplots should be labeled A, B, C, D not A.A, A.B etc for clarity. Specify in the figure caption if the error bars show standard error, standard deviation, or something else. Also, the *** significance marks are not clear--do these show differences from the control? from each other?

Response:

 The subplots labels have been improved and added more details to it. 

Please see Figure 2-page 18 line 317

The error bars show standard error. The significance marks represent the treatment’s significant effect over response variables as compared to control. The caption is edited as 

Linear mixed effect model(lme) representation of significant effects of each treatment (fixed effect) on percentage cover and biomass. Warming significantly increased the percentage cover, while the combined effect of Warm and clip significantly decreased it suggesting that clipping has a negative effect on overall percent cover and biomass as compared to control. a) Percentage cover was higher in warming plots in comparison to control but Warm and clip treatment significantly had less cover of species suggesting that clipping has a negative effect on overall per cent cover. b) greater biomass of all species in warming treatment as compared to control and other treatments, suggested that under passive heating plants tend to increase their vegetative growth. The significant codes are 0 ‘***’ 0.001 ‘**’ 0.01 ‘*’ 0.05 ‘.’ 0.1 ‘ ’ 1.

Comment: Figure 3 this is mostly overlapping information with the second half of figure 2, and should be removed or moved to the supplement.

Response 

It has been removed

Supplemental Figure S1 good as-is, although it would be clearer to write out "percentage" rather than "%age"

Response 

The axis label is changed to percentage cover please see supplementary S1Figure.

Supplemental Figure S2 the order of the treatments in A and B is different, which is very confusing when looking quickly at the figure. Please change the order for S2B to match the order in Figure 2 and S2A. Please add more information to the figure caption with what tests etc were used to produce the figure.

Response

The order of the treatments is the same for both percentage cover and biomass, the caption has been improved as below. Please see supplementary information S2Fig

S2 Fig: Species, Site and Treatment effects on percentage cover and biomass: a) the percentage cover of plant species increases in warming treatment and at multiple sites (M1, M2, M3). This is because of high abundance of these species on these sites while among species the response of individual species was not very different from each other, but some species responded more positively by increasing their percent cover as AR and AS. b) in comparison to control, above-ground biomass declined in clipping treatment but significant positive effect of warming treatment increased it for all species. 

Reviewer comment: Supplemental Figure S3 is redundant with Figure 2 and can be removed

Response: it is removed

Supplemental Figure S4 needs information in the figure caption as to what the different colors represent. Also, the caption says, "The relative abundance of plant species (presence/absence)" Which is it? Abundance measures are generally, biomass or maybe cover, while presence/absence is a binary.

Response

We acknowledge this point. S4figure updated as S3 figure also more details in caption are added. The relative abundance measure was biomass. The caption is improved as

S3Figure: Non-metric multidimensional scaling (NMDS) plot of plant species biomass with 95% confidence intervals using Bray and Curtis dissimilarity index over a 3 years experimental period along the elevation gradient of (3590–4696m). The relative abundance of plant species biomass inside the OTCs on each elevation site was estimated for all study years and shown by paths of mean values in Non-metric multi-dimensional scaling (NMDS) using Bray and Curtis dissimilarity index in R The plot shows no significant difference between the relative abundance of there is no significant change in the relative abundance of species inside both OTCs. OTC = Open top chamber, OTC1(warmed), OTC2, (warm*clip). Site 1 4,696m, Site 2= 4,059m Site 3= 4022m Site4=3,990m Site 5= 3,590mDifferent color represents each study year, red, 2016, green, 2017, blue 2018 

Reviewer’s comment: Tables

Supplemental table S2 This would be better to have one column for each site, and mark which species occur where rather than just tallying the number of sites they occur at. "Understudy" (used here and elsewhere in the ms) is unclear— do you mean understudied, that they have not been paid attention to in the literature?

Response 

It has improved as per suggestion. Please see the section Supplementary information S2 Table. The caption is improved, and “understudy” is changed to selected plant species as follows. 

A variety of plant species in experimental sites at different elevations. Each site was categorized according to the number of species present. Majority of selected species were present at each site, but there was a representative specie of each elevation.

Reviewer’s comment: 

Supplemental Table S3 we recommend focusing on the linear mixed-effects model and dropping the ANOVA approach in the methods, results, and here. For that reason, we suggest deleting these tables.

Response: 

These tables have been removed

Reviewer’s comment: Supplemental table S4 is good as it is

Response: Done

Reviewer’s comment: Supplemental table S5. We suggest moving Table a to the main manuscript and including F-values rather than just p values. These can be gotten by using the summary () function on the model fit. Table b should be dropped -- it's better to

have the interaction between the treatments than combining them in this way. Table c can be kept in the supplement.

Response:

S5Table a is moved to the main manuscript, which is now Table 2. Our model used t-test to assess the significance of the treatment, t-value are included in the main table, also the confidence level and estimates of the fixed effect to represent the best average effects. 

Please see page 17- line 302

Introduction

Reviewer’s comment

Line 56 – 61 These information are all good, but go beyond scope of the introduction. From my point of view, the essential information would be contained in a sentence like “ Repeated grazing can lead to exploitation and reduction of highly palatable species which are less resistant to severe grazing”. I still think that “selective palatability issue” is little unclear. Maybe instead of using this term you can rephrase it into the issues you are talking about. For example, do you mean that grazing leads to a change in forage species availability grazing animals can chose from? You could say “This leads to changes in plant availability that grazing animals can feed on and also to changes in nutrient availability”.

Nutrient availability of what? Leaves?

Response 

Needful done. Please see Section Introduction page 3 line 57-62

Repeated grazing can initiate exploitation and reduction of highly palatable species which are less resistant to severe grazing. This leads to changes in plant availability and nutrient content to grazing animals feed and forage in natural ecosystems. Thus, some plants are selected repeatedly as forage an d causes the exploitation by overgrazing. It may cause a significant reduction in the palatable species and attribute to selective grazing of highly palatable species that are intolerant to severe grazing and flattening [11,12]

Line 85-86 Maybe you can add a reference to this sentence

Response:

 Needful done. Following reference is added.

Olander LP, Johnston RJ, Tallis H, Kagan J, Maguire LA, Polasky S, et al. Benefit relevant indicators: Ecosystem services measures that link ecological and social outcomes. Ecol Indic. 2018;85: 1262–1272. doi:10.1016/j.ecolind.2017.12.001

Line 88-89 Even though you nicely improved this sentence, I still don’t have any idea about what these indicators are, maybe you can provide an example of such an indicator or explicitly write what are the indicators in your study. This information is essential for readers who have not been working with these kind of indicators before.

Response:

 Needful done. The sentence has been simplified, please see the section Introduction page 5 line 85-88

To address these shortcomings, here, we provide an example of integrating plant community responses to grazing and warming with ecological and socioeconomics characteristics to assess how people's approach to culturally important plant species may be affected in the future. 

Reviewer’s comment: Line 94 unique richness of what? Plants?

Response: Unique plant species page 5 line 92

Reviewer’s comment: Line 108 you may mean “investigate” instead of “indicate”?

Response: corrected page 5 line 105

Reviewer’s comment: Line 110 “factors” is not necessary.

Response: Corrected

Reviewer’s comment: Line 112 -114 This could be deleted since the information are already given in the objectives before. If you want to keep it, change the order so it matches the order of the objectives.

 Response: Needful done,The repetitive information is deleted, and sentence structure is updated for more clarity. Please see the section Introduction page 5 line 104

Therefore, the specific objectives of this study were to 1) investigate culturally and medicinally important plant communities’ responses to climate warming and clipping (simulated grazing) by manipulative experiments and 2) to link these ecological factors to socioeconomic wellbeing of local inhabitants by assessing the dependence and utilization of these resources through ethnobotanical surveys.

Methods:

Reviewer’s comment: Line 119 km2, lowercase k

Response: Corrected page 6 line 113

Reviewer’s comment: Line 120 replace “it” with “the study”

Response: Corrected page 6 line 114

Reviewer’s comment: Line 121 replace “it” with “summer”

Response: Corrected line 115

Reviewer’s comment: Line 120-121 simplify this to just "from May till August" since it's also August at higher elevations.

Response: Corrected line 115-117

Reviewer’s comment: Line 126 declines of what?

Response: decline of glacier line 121

Reviewer’s comment: Line 129 "The alpine KNP region includes the following..."

Response: Corrected line 125

Reviewer’s comment: Line 141 space missing between “biomass” and “(“

Response: Corrected

Reviewer’s comment: Line 148 -151 The exact same sentences appear earlier in the introduction; you might want to delete them here to avoid unnecessary redundancy?

Response: Thanks for pointing it. It is deleted 

Reviewer’s comment: Line 153 Needs to refer to a specific table or part of the supplementary information (currently Supplement S6?). It may also be helpful to have one supplementary document with figures and tables and another one for the questionnaires only. This you could for example name supplementary information II, so the interested reader directly knows where to find the respective information.

Response: The questionnaire is added as supplementary information S5. Please see supplementary information

Reviewer’s comment: Line 161 -163 This is phrased little awkward, I assume you mean that those species were abundant in your experimentalities? I suggest to rephrase this sentence.

Response: the sentence is now rephrased Please see section Material and Methods page 7 line 146

Based on this initial screening survey, sites for manipulative experiments were selected where these species were in abundance as excerpted by the informants. 

Reviewer’s comment: Line 163 This would be a good place to add more information on what main questions you asked about each plant, since most of your readers will not want to be searching through your supplement. Something like "For each species, we collected asked informants about... [climate change responses etc.]"

Response: Needful done. Please see section material and methods page 7 line 148-150 

We asked questions about local medicinal plants, their traditional uses, methods of crude drug preparation, climate change responses etc.

Reviewers comment: Line 206 -209 I suggest that this entire part could be integrated the data sampling part

Response

Needful done: please see section material and methods page 10 line 209

Reviewer’s comment: Line 216 Why to stimulate growth, isn't the intention to simulate herbivory? Or do you mean you want to observe growth under herbivory?

Response: Yes, it means growth under herbivory

Line 219 -220 For consistency it would be good to move the sentence about the height measurements to the data sampling paragraph

Response: As plant height was measured before clipping and taking the saplings for biomass measurement, these results haven’t been included as main results, so we have removed this particular line about plant height measurement.

Line 257- 260 In the first part you write that you analyzed species-specific responses, but the tables show model results across species. I suggest you either adjust this in the text or refer to a table/graph with species-specific results. Furthermore, table b in S3 does report results of linear mixed effect models (according to the subscript of the table), this is confusing, because you don’t mention linear mixed effect models when you refer to the table in the supplementary information. Also related to table b in S3, it is not clear to me whether site and species were used as random or fixed factors in the model. In the text in your ms you report them as random factors, but in the table, they seem to be reported as fixed factors. Also, you only report the interaction of W x CL but no main effects. It would be good to include these in the table and also specify which test you used to assess the significance of the treatments like F- or ChiSq test. You can access these tests after you have run your mixed effect model via the Anova command from the “car”-package: Anova (mymodel, test.statistic = “F”) for F-test. Instead of using the summary statistics as you now did in table S5 you can instead report F-test results. Related to table S5 do you have an explanation for why the conditional R2results in NA when percentage cover is the response variable?

Response: 

The statistical analysis section has now been improved Please see section material and methods page 11 line 228. The further response to the comment is as follows: 

The species-specific response within each treatment and the variation of each species response towards the warming and clipping was estimated through one way ANOVA to investigate how significant was the response of a particular species (S3Table, Figure 3). Though the linear effect model also shows that concerning each treatment it is good to know the variance among species.

S3tables are removed as per reviewer’s comment. Results of the main and interactive treatment are included in Table 2 and S4Table. The model statistical summary where each treatment is considered as fixed with random effects of species and site is now added in the main text as Table2. The test we used for lme was t test and then to check the variance we used ANOVA chi sq test. The detailed statistics are also added in the table 2. The summary statistics of S5 table has been improved to remove the error of NA.

Line 261 Please add a reference to the R package you used. The most common packages to run linear mixed effect models are either lme4 (Bates et al. 2015) or nlme.

Response: The reference is added. Please see page 11 line 234 [59]

59.Bates D, Mächler M, Bolker BM, Walker SC. Fitting linear mixed-effects models using lme4. J Stat Softw. 2015;67. doi:10.18637/jss.v067.i01

Reviewer’s comment: Line 264 -267 This is a key section that needs to be clarified. First, I think your response variables were the same as in your ANOVA models e.g. biomass and percentage cover, so these should not be termed as fixed factors but as response

variables. Second, neither biomass nor percentage cover is metrics of diversity! It can be helpful to include the model formulation here, which I think what you're trying to say is: cover ~ warming * clipping + (1|site) + (1|species). The best way to describe this model would be something like "Response variables of percentage cover and biomass were modelled using [xx package] with fixed effect predictors of the warming treatment, the clipping treatment, and their two-way interaction, and random effects for site and species (Table xx). We separately ran models with species as a fixed effect to check the effect of treatment for each species (Table xx)"

Response:

 We agree to the reviewer’s viewpoint of keeping only linear mixed effect model results. The correction is made by removing the fixed factors but as response variables. Also, the model formulation is described, and a suitable reference is added by describing the details of the package used. The reference to the table showing the summary statistics is also mentioned as suggested by reviewers as follows also see material and methods page 11 line 231-240

The species-specific responses within each treatment and their significance were estimated through One-way ANOVA (S3 Table, S2Fig). Response variables of percentage cover and biomass were modelled using lme4 package [59] with fixed-effect predictors of the warming treatment, the clipping treatment and their two-way interaction(warming x clipping, W x CL ), and random effects for site and species(Table 2) We separately ran models with species as a fixed effect to check the effect of treatment for each species (S4Table) The following model formulation was used

Perc_cover ~ warming + Clipping+warming* clipping + (1|site) + (1|species) (main and interactive effect as fixed effect while site and species as random. 

Perc_cover ~ Factor+species+Factor:species(1|site) 

Reviewer’s comment: Line 267 I suggest starting a new paragraph for each different analysis you conducted--one for the linear mixed model, one for the susceptibility zones, one for the NMDS. I think you also need more information about this "susceptibility

zones" figure 4--why did you do it, how did you do it, what is it intended to show?

Response:A new paragraph is sectioned also more details are added to susceptibility zones 

Please see section material and methods page 12 line 241

From the results of surveys (Table1) and linear effect model some highly medicinal and cultural important species (Artemisia rupetris, Poa alpina, Primula macrophylla, Potentilla hololeuca, Astragulus penduncularis) and illustrated their susceptibility to combine effect of warming and grazing (W x CL). The approach adopted was polygons-based susceptibility zones to show the positive and negative effects of both the treatments and future prediction about the occurrence of constitutive species under the stress of climate warming and herbivory. 

Line 275 You should also report which version of R or RStudio you used (latest version in R is 4.0.3). You can f.ex. write: We used R (RStudio) to run all analyses (R Core Team 2020 version XXX). For the vegan package you have to cite (Oksanen et al. 2017). There is a neat function in R called citation() which shows you how to cite the packages you used

Response:

We highly appreciate reviewer’s helpful remarks and suggestion, the details of the version of RStudio with suitable reference is added. Sufficient information about susceptibility zones made under combine effects of both the treatments is added. Please see section M&M page 12 line 252

All the analyses were performed in RStudio 2.4-4. (R Core Team 2017) using the vegan packages [61]

Results

Reviewer’s comment: Line 321 - 328 This section with the ANOVA can be removed if you focus on the results from the linear mixed effect model. Also the numbers reported here don't match the p-values in S3 Table B, shouldn't they be consistent with that?

Response: Needful done

Reviewers comment: Line 331 - 343 The language here makes it sound like you are testing the effect of the site "impact" rather than just variation. Also words like "increased" and "changes" make it sound like you're looking at the change through time, but your methods indicate you're just looking at the differences between treatments. Instead you should use statements like"has higher cover than the control" to match the analyses you're doing.

Response: 

The result section has been updated accordingly. Please see section Results page 16 line 291-339

Further details are added as followed

The individual factor impact on the percentage cover and biomass is shown in the Fig 2 Table 2, illustrating the profound impact of warming(W), clipping (CL) and interaction (W x CL) between the treatments keeping the site as a random effect. Also, the overall differences in percentage cover of control and treatment plots at site MERGE1(M1, 4696 m) and site MERGE5(M5,3346 m), is found to be significantly different (M1 has higher cover and biomass than control while at M5 there is no significant variation between control and treatments(S2Fig). At other sites (M2,4059 m, M3,4022 m, M4,3990 m) the response of species between control and treatment plots was similar (S2 Fig). While species-specific response can be seen in species effect plot where some plant species as Poa alpina (PA), Astragulus penduncularis (AS), Artemisia rupestris (AR) Potentilla hololuca (PT) have significantly higher per cent cover in comparison to others while the above-ground biomass changes were significantly positive in the warmed plot of these species (S3Table,S2Fig). However, inter-annual variations among the species percentages cover were significant. Total increase in plant cover in response to warming was highly variable among taxa, ranging from 1 ± 0.6% for Bistorta officinalis to 18.7 ± 4.2% for Poa alpina (S3Table). 

Reviewer’s comment: Line 337 It is advisable to avoid “seem” if you have statistical evidence for something affecting.

Response: corrected

Reviewer’s comment: Line 340 Negative? I thought it was positive--higher in the warmed plots?

Response: Corrected

Line 341 How can you state this, if you have pooled the data across years? Did you run separate models for separate years, or did you include the year as a fixed factor in your mixed effect model? You would need to either explain these models in your statistical analysis section and include them as ex supplemental tables or remove this part.

Response: The confusing sentence is removed

We pooled the data across the years and did not check the effect of year on growth instead considered the final experimental year as a result.

Line 344 You should have 1-2 more sentences here on the species-specific results that are presented in Figure 2

Response: Needful done Please see section results page 19 line 339-346

 The plant community in warming plots responded significantly positive towards warming, while the response magnitude and intensity was highly variable among species. Important species - Artemisia rupestris Poa alpina, Carex divisa, Saxifraga, Taraxacum officinale etc have higher percentage cover and biomass in warmed plots in comparison to control and other treatment plots. Above ground biomass was also higher in warmed plots for some species. The negative effect of clipping and interactive treatment was evident for species such as Taraxacum officinale (TM) Peganum hermala (PH) and Primula macrophylla (PC) (Fig 3)

Line 347-351 This sentence is not clear to me. Do you intent to say that sites did not significantly increase the variation in your data? Assuming you had site as a random effect in your model.

Response: While keeping the site as random, it was intended to state that site might be the determining factor for certain species significant response variables as considering the fixed factors the variability was high among site species responded differently at different sites. 

Please see section results page 18 line 330-333

Commnent: Line 355 You can compare the responses in your treatment’s plots to the control, but usually control is not considered a treatment. It would thus sound better if you wrote something like "...in above ground biomass of each species in response to warming, clipping and warming x clipping compared to control.

Response: Needful done

Please see section results page 19 line 348 - 350. In above ground biomass of each species in response to warming, clipping, and warming x clipping compared to control, warming has overall positive effect on percentage cover and biomass of species while warming x clipping has negative effect. The * shows significant change in particular species with respect to the specific treatment as compared to control while the dots are outliners

Line 378 Should there be a new paragraph here from the NMDS results?

Response: Corrected see page 20 line 365

Reviewer’s comment: Line 380 Isn't this supplemental figure S4?

Response: Corrected

Reviewer’s comment: Line 382 This supports the choice to incorporate site as a random effect in your mixed effects model

Reviewer’s comment: Line 403 Delete “for”

Response: Corrected

Discussion

In your discussion, it would be nice if you could further emphasize the connection between your experimental findings and the ethnobotanical survey. Also, instead of AGB just say biomass, it doesn't take much more space and is much easier to read.

Response: Needful done. We have further elaborated the connection between our experimental results and survey information and justified the correlation. Please see section Discussion page 25 line 490-491

Reviewer’s comment: Line 418 You did not measure community data, but rather species-specific warming and clipping responses.

Response: Needful done. Please see section “Discussion” page 21 line 389-390

Line 419 - 420 The commas before and after "was" should be removed, and there should be a comma added after "sites"

Response: Corrected

Line 425-427 In your methods section (the statistical analysis) you don't report that you have used height as a response variable, nor do you report it in any of your results. If you did, you should add it, otherwise, you cannot discuss it here.

Response: Needful done paragraph from line 393-396

Line 429 I think you are combining two plausible mechanisms here: competition for resources and limitation in space. You might want to elaborate on both of them. Check ex. Borer et al. 2014 in Nature

Response: The mechanism details are given, and sentence structure is rephrased as follows also Please see page 21 line 402-412

This is attributable to increased competition, as biomass increased, plants were usually taller, and few individual species could fit into plots. Warming may lead to enhance plant growth and cover, it reduced the diversity of species as compared to the clipping treatment. In clipping treatment plots, the plant species did not grow well and had reduced their biomass and cover, but they maintain the diversity. This mechanism was elaborated by Borer et al, 2014[67] who demonstrated the effect of herbivory on the grassland production and showed that where herbivory increased the ground light it also maintains the biodiversity. Hence, where warming benefits some plant species in increasing their cover and biomass, it is likely to induce the dominancy of some species, whereas our experimental results suggested that clipping rescued the biodiversity.

Line 463 "Other studies on alpine meadows [where?] also suggested..."

Response: Norway, corrected and added reference Please see Discussion page 23 line 447

Line 469 Since intensity and frequency of clipping should have been the same for all of your studied species, I think it is rather something else that drives this species-specific response. You could think about whether this species has specific traits allowing it to regrow fast after clipping/grazing which the other species might not have.

Response: 

Please see page 24 line 453-460

These different responses of vegetative biomass to clipping/simulated grazing can be attributed to specie- specific traits which allow the plant species to regrow faster than the others. Also, where clipping helped the certain plant species by increasing the ground level light, warming triggered the temperature and plant responded positively to the W x CL treatment. Some plants have evolved tolerance to herbivory by growing compensatory tissues so rapidly that help to cope with damage, thus allowing natural selection to favor herbivores and prompting plants to develop a new form of resistance, such as reduced apparency, chemical defenses (secondary metabolites) and indirect defense [73].

Line 473 "clipping" not grazing

Response: Corrected 

Line 478 "stronger" not "sturdy"?

Response: Corrected 

Line 490-492 This sentence is not clear to me. Do you want to state that the species you studied are more susceptible to warming than to grazing?

Response: It is explained as follows: Please see page 25 line 480-482 

They are vulnerable to clipping but warming has favored some species over others, so those selected species outcompete the other slow growers. The natural selection may play a role for these plants to be more abundant, thus palatable for grazers.

Line 493 Wang et al. add the year

Response: Wang et al., 2019 line 483

Line 494-499 The first part of this part is about the impact of clipping, but you then refer to results of your survey in relation to warming. Please, clarify this. Also, can you state references for the loss of biodiversity. There are plenty of good ones available!

Response: new reference has been added and rephrased the paragraph as follow 

Please refer to page 25 line 484-490

 In our study, some of the species [P. alpina (PA), A. rupestris (AR)] responded significantly towards climate warming and clipping but not reducing their biomass in comparison to others (Fig 2). This supports the assumption of natural selection where strong competitor species surpass weak competitors, thus lead to loss of biodiversity[34,53]. This experimental evidence was also in line with the information gathered from the local informants, mentioning increased forage availability of these species throughout climate warming.

Line 501 You have not reported any measures of diversity related variables like species richness, so speaking about biodiversity isn't appropriate here.

Response: Our maximum plant species biodiversity was at site 1 and they responded positively towards warming. Here the purpose of explaining the dete/rments of biodiversity is to predict the species diversity over here in current scenario of climate warming. The species composition and diversity on experimental site is further elaborated.

Please see page 25 line 491-500 

Reviewer 2

General comment

The authors have made substantial improvements, especially with describing the experimental design and survey methods. However, while there have been clear attempts to improve the statistical methods and results, I’m unclear about which approaches were used and why. It seems the authors used several different models (one-way ANOVA, two-way ANOVA, mixed models with different random factors), but I don’t think they are all necessary. The model interpretations in the results section are confusing, and do not clearly align with specific experimental questions. Instead, the authors should decide which questions are most interesting and choose 1-2 statistical methods based on that. Also, I’m unclear on how the % increase/decrease values were calculated (lines 250-252) and whether they represent the entire 3-year experimental duration. I still think it could be best to use repeated measures with year as a variable in the statistical model to see how the warming and clipping effects built up (or not) over multiple treatment years. Also, there is some discussion about differences between elevational plots (i.e., lines 333-335, 500-505) but it’s unclear how that was formally assessed.

Response: 

The valuable comments and suggestions of reviewers have been adapted carefully and linear mixed effect models’ approach is used to answer the main and interactive effects of treatments while keeping the site and species as random effects. As we want to show the treatment effects on response variables of biomass and percentage cover of selected species, species-specific response to check the variation within treatment is also shown and interpreted.

During three-year course study (2015-2018) we have pooled the data across years to show main and interactive effects and did not run separate models for separate years as it was not the objective. We have not included a year as a fixed factor in our mixed effect model. The per cent increase and decrease value were calculated as considering first experimental year as control and last experimental year as the effect of treatment thus the values are then compared with control to determine the %increase or decrease in experimental plots. The random effect of the site is further explained through NMDS to check the variation in species relative abundance of biomass in the OTCs plots. 

Please see statistical analysis in section Material and Methods page 11 line 228

result section see page 16 line 288

Reviewer’s Comment: Lines 44-46: This is unclear. Land and water degradation do not “cause” climate change.

Response: Please see section introduction page 3 line 44-46

The sentence is structured for more clarity. Importantly, high altitude regions are already under the stress of land and water degradation, climate change may exert extra pressure and makes them highly exposed

Line 49: Not “global” mean surface temperature if just referring to the Himalayan region.

Response: Corrected

Lines 85-86: I appreciate that “ecosystem change assessment” is defined, but the definition is still unclear. Also, the rest of the paragraph does not mention these “ecosystem change assessments,” so why is it brought up here?

Response: The lines have moved to start of the paragraph to give it a more clarity please see page 4 line 80 

Reviewer’s comment: Line 106: “primarily runs the financial circle” is unclear

Response: Removed from the text as it is irrelevant here

Reviewer’s comment: Lines 112-114: This last sentence is unnecessary

Response: Removed

Line 123: Incomplete sentence

Response: please see section M&M page 6 line 115-117

Corrected as : The maximum temperature in May recorded goes up to 25ºC [44] while in winter it drops down below 0ºC (up to -10 ºC) [17,45,46].

Line 133: Is “yoks” supposed to be “yaks”?

Response: Corrected

Lines 144-151: This was all stated in introduction already, so is not needed here.

Response: Corrected

Lines 156-160: Put this description of the valleys into the previous section (Study Area).

Response: Corrected and added to study area Please see section M&M page 7 line 133-136

Lines 161-162: Put this sentence into the next section (Warming and Grazing Experiments)

Response: Needful done Added to the relevant section. see line 162

Lines 167-170: The RFC description should be included in the “statistical analyses” section instead.

Response: Done Added to statistical analysis section please see section M&M page 11 line 229-231

Lines 171: What does “we involved plant material” mean?

Response: The specific question asked were about native plants and their medical and cultural properties. The sentence is now restructured, please see line 148-149

Lines 182-183: What does “which lies in alpine meadow of the vegetation zones” mean?

Response: There are four vegetation zones in which KNP is divided already described in introduction section. The study sites are in the alpine meadows of those zones Please see section introduction page 6 line 125-134.

Line 184: Since there was only 1 replicate for each elevation, the entire experiment is an RBD, but not each site.

Response: Corrected, please see line 164-169

Lines 186-188: Were the data collected outside of this fenced area used for any analyses? I don’t see it used.

Response: we did not use the data outside of fenced area but the plots in surrounding were considered as reference to mimic the herbivory level on plant species for clipping treatment. 

Line 191: Where was this reported earlier? Or do you mean it was reported by another prior study?

Response: Yes, prior study the reference is given, line 176

Line 197: How can a hexagonal shape have the dimensions of 1.5 x 1.5 m or 1 x 1 m? This reporting style is reserved for squares/rectangles. Do each of the six sides have that length? If so, that needs to be stated.

Response: The area covered by OTCs was inside the 2.5m2 subplot. The area it covered was 1.5m2and 1m2 quadrat was used to sample the species biomass and cover inside OTC that corresponded from the open top of chamber. Please refer to Figure 1 line 202

Line 198: Since your fences prevented grazing, they should be called “exclosures” instead of enclosures throughout the

paper.

Response: Needful done 

Lines 212-214: This is confusing.

The experiment consisted of two treatments, one 2.5m2 subplot inside exclosure subjected to clipping treatment, one similar dimensioned plot outside the fenced area as reference to estimate the herbivory level on plant species in order to mimic the grazing. Line 190

Line 214: Define the abbreviation “W x CL” the first time it’s used.

Response: Needful done. please see line 193

Line 216: Simulate “grazing” (instead of “growth”)?

Response: Corrected

Lines 218-219: What is described here? I don’t understand the 2 grazing comparisons. The analyses only discuss the manually clipped plots.

Response: The grazing plots outside the fenced area is used as reference to check the level of herbivory outside and then employing it to clipping treatment. the intention is not the comparison.

Line 220-221: This is confusing.

Response: Corrected as follows Please see section M&M page 10 line 197-201

At the start of experiment, the grazing level of herbivores was estimated and clipped inside the exclosure in clipping treatment, then each year plant growth was recorded from that clipped point. In the clipped plots the aboveground part of all selected plant species were clipped to about 2–4 cm on an average above the entire plot. There was no disruption in the unclipped plots.

Line 224-225: Remove this sentence.

Response: Needful done

Line 132: How large were the quadrats and the quadrat grids?

Response: 1m x 1m quadrat plot with 10cm quadrat grids. Please see line 217-219

Line 234: How were the “selected plant species” determined?

Response: The plant species were selected based on their medicinal and cultural potential mentioned by informants, citation frequency, occurrence in alpine meadows and also the palatability as described by local inhabitants Please see section M&M page 10 line 215-217

Lines 236-241: Put all the species into a table and reference it instead.

Response: Yes, it would be wonderful to add them into a table with reference, but that would be a repetition as these are already in the tabulated form in Table 1 and S3Table. The reference to Table 1 is given

Lines 243-244: What does “vertical projections of vegetation” mean?

Response: That is how cover is defined, as to estimate the area covered by a plant 

Daubenmire, R. 1959. A canopy-coverage method of vegetational analysis. Northwest Sci. 33:43-64

Line 248: How large were the clipping areas? Were the same areas clipped multiple years in a row?

The clipping treatment area was 2.5 x 2.5m at each site, Yes, the same area was clipped in a row

Line 250-252: I’m unclear about what you used to calculate % increase/decrease for the data. If the values were averaged over all 3 years, how was % increase/decrease calculated as the “difference between control and last year data

points”?

Response: yes, it was calculated as difference between control and last year data. % increase= last year cover/cover in control x100. If the number is positive, then its increase and if negative, it’s percent decrease. Please see line 224-226

Lines 155-156: This sentence not needed here.

Response: It is removed

Line 290: Missing % for Asteraceae

Added

Lines 291-293: Unclear

According to the reference given, if a plant species is mentioned or cited for a typical trait by more than three informants it is considered as reliable medicinal property [62], please see 15 ;line 270

Lines 301-302: This sentence not needed.

Response: removed 

Lines 306-308: Save for discussion section

Response: Added in discussion section, line 395.

Lines 308-309: This is confusing.

Response: The sentence is removed as it has already explained at the start of paragraph

Lines 313-315: This sentence not needed here.

Response: Needful done 

Line 321: What does “across all three growing seasons” mean in this context? Average over 3 years? The value at the end of year 3? Each year assessed independently (then year would need to be included in the statistical model)?

Response: this section is removed from the manuscript and only mixed effect model results are shown. Please see results

Line 425: Plant height was not statistically analyzed or discussed in results section, so should not be mentioned here.

Response: It is removed

Lines 425-428: Unclear

Response: It is clarified now. The increase in plant species abundance and cover was observed along the elevation gradient but this effect was more profound for specific species which increased their vegetation cover and biomass in warmer plots, but reduced biodiversity. Please see page 21 line 403

Line 456: State author’s names (instead of reference number) if specifically referring to a study.

Response: Needful done. Please see line 439

Lines 490-494: Unclear

Response: It is clarified. Please see page 25 line 478-480

The intensity of positive response was higher in the zone of warming as higher percentage cover and biomass (Fig 4), thus predicted the vulnerability of these species towards herbivores as more availability more grazing [78]

Lines 496-497: How does the data support this assumption?

Response: Our data supports the natural selection as some species has higher biomass and cover in warming plot than others. please see page 25 line 486-490

This supports the assumption of natural selection where strong competitor species surpass weak competitors thus lead to loss of biodiversity[34,53]. These experimental evidences were also in line with the information gathered from the local informants, mentioning increased forage availability of these species over the period of climate warming.

Lines 500-505: I like the biodiversity discussion. Expand on this more.

Response: Needful done. please see page 25 line 491-513

Figures:

Figure 1: The figure legend needs to include much more detail – it needs to fully describe each component of the plot design. For example, why are there control plots in the warming and warming x clipping larger plots? What do the letters on the OTC mean? Unclear what the 1 m and 1.5 m arrow are pointing to.

Response: Figure 1: the details are added in the caption to best illustrate the experimental design and explain the sampling sites for species composition and biomass estimation. The schematic diagram has been improved with more clarification for plots and subplots. Please see Figure 1 page 10 line 202

Figure 2A: First, these should be bar charts or box and whisker plots instead of line graphs. Line graphs are usually used to represent changes through time. What do the stars signify? Panel A needs a better y-axis label (no shortened words, underscores).

Response: It is suggested to change it into box and whiskers plot with suggestion that line graphs represent changes over time. It is humbly mentioned here that this statement supports to our study data too, we also studied climate warming impacts over a period of 3-years. Thus, we request to consider the Fig 2 in line graphical form. For Panel A, needful done as suggested. 

Figure 2 as box and whisker plot is also submitted as a separate file and with the rebuttal letter just for your kind perusal.

Linear mixed effect model representation of significant effects of each treatment (Fixed effect) on percentage cover and biomass. a) As compared to control percentage cover was higher in warming plots(p<0.01) but Warm and clip treatment significantly had more cover of species but less than warming plots suggesting that clipping affects the overall percent cover. b) greater biomass of all species in warming treatment as compared to control and other treatments, suggesting that under passive heating plants tend to increase their vegetative growth. The stars represent the significant level. The significant codes are 0 ‘***’ 0.001 ‘**’ 0.01 ‘*’ 0.05 ‘.’ 0.1 ‘ ’ 1

Legend line 348-349: Is there statistical evidence for this claim?

Response: This line is improved as above in the legend of figure 2

Line 349-351: This is not needed here.

Response: removed

Figure 2B: First, why is this labelled “2B”? It appears to be an entirely separate figure, so should be “Fig 3.” What do the stars and dots represent? The axis labels should be written without underscores. Also, why does it appear that most of the control plots increased? How is it that none happened to decrease over the 3-yr period?

Response: This figure is updated as Fig 3. Here is the improved complete legend of it 

Fig 3: Species-specific responses towards each treatment. a) data showing the percent increase in cover of each species in control, warmed, and clipped plots b) in above ground biomass of each species in response to warming, clipping, and warming x clipping compared to control, warming has overall positive effect on percentage cover and biomass of species while clipping has negative effect. The * shows a significant change in particular species for the specific treatment while the dots are outliners.

Figure 3: You explain in your review response to what the “beans” represent, but this description needs to be included in the figure legend. It would be helpful to label each panel as “warming” or “clipping” depending on what it shows. It might also be nice to identify (maybe by color) which species are the most culturally important- it ties this data to the survey responses that way. Again, there are still no letters indicating statistical significance (as stated in the legend). The stars show statistical significance of each bar compared to what- each other.

Response: As this Figure was in repetition with figure 3 so it is removed now

Figure 4: What do the stars mean?

The significant positive effect of warming on species percent cover and biomass in absence of grazing

Figure S1: Improve the y-axis label

Response: Needful done. Improved and full form of the y-axis is added.

Figure S2: Similar to figure 2A, these should all be bar chars or box plots. Improve axis labels. For the right panels, reorder species to group by cultural importance or functional group. Much more detailed legend needed.

Response: As a supplementary figure, this has been kept as it is, because it is not the main result but as a supporting information, and the information given in it has already been explained in figure 2 and figure 3, but for sites as a fixed factor this can be important information. The order of treatment has been changed.

Figure S3: Omit or re-structure this figure- it’s unreadable.

Response: Needful done. 

Figure S4: What do the different colored dots represent? It’s hard to tell which labels go with which points. I’m not familiar with these types of plots, but this is confusing to me. Maybe put more description in the legend?

Response: NMDS (Non-metric multi-dimensional plots) used to determine the relative difference between the species in a community over the period of a particular time

Non-metric multidimensional scaling (NMDS) plot of plant species biomass-with 95% confidence intervals using Bray and Curtis dissimilarity index over a 3–years experimental period along the elevation gradient of (3590–4696m). The relative abundance of plant species biomass inside the OTCs on each elevation site was estimated for all study years and shown by paths of mean values in Non-metric multi-dimensional scaling (NMDS) using Bray and Curtis dissimilarity index in R The plot shows no significant difference between the relative abundance of there is no significant change in the relative abundance of species inside both OTCs.

OTC = Open top chamber, OTC1(warmed), OTC2, (warm*clip). Site 1 4,696m, Site 2= 4,059m Site 3= 4022m Site4=3,990m Site 5= 3,590mDifferent colour represents each study year, red, 2016, green, 2017, blue 2018. Please see Figure S3 

Table S2: Thanks for editing this table. One suggestion – make separate columns for each elevation, and then put an X in

each column where a species is present at that elevation. Right now, you can’t tell which species are present at which

elevation.

Response: corrected please see S2 table

Table S3 & S5: Include the full ANOVA tables/statistical output. More than just the p-values should be included.

Response: As the analysis were repeated with the linear effect model so we removed the un-necessary results and figures and only kept the results of mixed effect model keeping the effect of site and species as random. Thus S3_tables are removed, only the species-specific responses are added and the values are represented as mean ± SE and more information as added to Table 2 

Data points: Need metadata- full description of what each dataset it and the meaning of each abbreviation.

Response: The data points excel sheet is now edited with full form and description of abbreviations

---

## [Decision Letter · Decision Letter 2]

5 Mar 2021

PONE-D-20-21262R2

The Response of Culturally Important Plants to Experimental Warming and Clipping in Pakistan Himalayas

PLOS ONE

Dear Dr. Ali,

Thank you for submitting your manuscript to PLOS ONE. After careful consideration, we feel that it has merit but does not fully meet PLOS ONE’s publication criteria as it currently stands. Therefore, we invite you to submit a revised version of the manuscript that addresses the points raised during the review process.

The manuscript has improved significantly after tow rounds of revisions. However, there are still some minor issues, particularly related to statistical tests and language, which need further attention.

We look forward to receiving your revised manuscript.

Kind regards,

Abid Hussain

Academic Editor

PLOS ONE

Journal Requirements:

Additional Editor Comments (if provided):

Dear Authors,

There are still some minor issues in statistical testing and language. Please address reviewer's all comment adequately and resubmit.

Reviewers' comments:

Reviewer's Responses to Questions

**Comments to the Author**

1. If the authors have adequately addressed your comments raised in a previous round of review and you feel that this manuscript is now acceptable for publication, you may indicate that here to bypass the “Comments to the Author” section, enter your conflict of interest statement in the “Confidential to Editor” section, and submit your "Accept" recommendation.

Reviewer #1: (No Response)

2. Is the manuscript technically sound, and do the data support the conclusions?

Reviewer #1: Partly

3. Has the statistical analysis been performed appropriately and rigorously? 

Reviewer #1: No

4. Have the authors made all data underlying the findings in their manuscript fully available?

Reviewer #1: No

5. Is the manuscript presented in an intelligible fashion and written in standard English?

Reviewer #1: No

6. Review Comments to the Author

Reviewer #1: General comment

The manuscript has hugely improved and is much easier to read and understand. There are still some issues of confusion with the statistics, including the "Factor x species" model on line 240, use of t-test to assess the significances of the treatments, and understanding/reporting/interpretation of the interactive effect. Furthermore, the authors present results for the different sites for which the description of the analysis and model output tables are missing. Specific comments marked with * are particularly important.

There is still some language that is confusing or unclear in wording, especially in the results and discussion.

Also, please check for upper- and lower-case letters throughout your figures and tables and chose a consistent style. Consistency is also needed concerning the terminology of the clipping/grazing treatment and aboveground biomass.

The authors state that "Yes - all data are fully available without restriction" and "All relevant data are within the manuscript and its Supporting Information files." but we see NO raw data in the supporting information file, just the supplemental tables with output. We hope this is an accidental error but it needs to be corrected to meet the journal data requirements.

Running title and abstract

Line 13: Lowercase “H” in herbivory

Line 22-23: This sentence may be easier to understand and read if you wrote: “Experimental warming increased biomass and percent cover throughout the experiment”. It is not necessary to report the p-value or that you used sites as a random factor. The abstract should highlight your main findings in a concise form, so avoid any unnecessary detail.

Line 27: Unclear what “due to it’s abundance at the site” means.

Line 28: Had instead of “has”.

Line 29: For consistency it would be good to give the same names to your treatments. Clipping and simulated grazing are the same, right? Can you change this?

Line 31: Stronger effects, what does this mean? Please, report whether this refers to an increase or decrease.

Introduction

Line 58-59: Sentence unclear, maybe the “to” needs to be replaced with “for”?

Line 60: Delete the “s” at the end of “causes”.

Line 83: underrepresented?

Line 84-85: Sentence unclear.

Line 101: This is the first time your mention KNP, please provide an explanation of the abbreviation.

Methods:

Line 117: No brackets needed when you mention the altitude.

Line 126: Parking area??

Line 132: Comma missing between lanceolata and Saxifraga.

Line 139: I think you mean individuals instead of species here.

Line 190: Here you write that your clipping plot inside the fence is 2.5 m2, but from Figure 1 it looks like it is 1m2. Please clarify this.

Line 193: Could you add the clipped area inside the OTC to Figure 1?

Line 209: Lowercase “a” in “all”.

Line 212: As pointed out in an earlier review, this reference seems not appropriate for plant metrics, but focusses on ground dwelling insects in riverbeds. You could cite for example Shaver and Chapin 1991: Production: biomass relationships and element cycling in contrasting arctic vegetation types, Ecological Monographs.

Line 223-224: What do you mean with “We clipped the vegetation from OTC”? This sentence needs clarification.

* Line 240: The model formulation for this second, species-specific model isn't clear. What is "Factor" in this model? I think what you're intending to write is Factor * species + (1|site)?

Line 241 -244: Sentence unclear.

*Line 244: “polygons- based susceptibility zones” unclear. If this is an established method you should provide references.

* Line 246: You should add here which kind if test you used to assess the significances of the treatments, and what function or package you used for that test.

Line 250: Point missing between “groups” and “the”.

Results

Line 267: Percentage for Asteraceae is missing.

Line 295: You state that clipping is not significant, but in Figure 2a there are three asterisks to indicate high significance. Same for the interaction which is not significant according to Table2. Also based on Table 2 there should be two asterisks for the interaction the Figure 2b and only one for the clipping treatment.

Line 307: Why two models? What you show in the table can be derived from one model which contains both the main effects and the interaction.

Line 314 – 316: Species is missing an “s”, the sentence is unclear.

Line 318 -320: This sentence could be moved to the results, but is not needed to describe the figure. You already do that in a) and b).

Line 319: Write the full name of the treatment, not just “warm” and “clip”. It is always good to be consistent in the use of treatment names/terminology!

Line 319 – 320: If clipping had a consistent negative effect you would see this as a main effect. Here you need to think about why the combination of warming and clipping decreases percentage cover.

Line 322: See the above comment. In general, you don’t need to add interpretations of the results to the figure caption, reserve this for the discussion.

* Line 326: You did not mention in your statistical analysis that you assessed differences in percentage cover and biomass between the different sites. Please add this information. Furthermore, you should also provide a table with the model output. Referring to the figure only is not sufficient because the reader cannot judge based on a figure whether something is significant or not.

Line 330 – 333: Sentence unclear and confusing.

Line 336 – 337: Higher biomass is a result of warming, but warming is not determined by higher biomass. Please, rephrase this.

Line 337: Figure S2 species-specific plots, are these plots in response to warming or in response to clipping or both? What does this figure add in comparison to figure 3?

Line 343: Comma missing between ruprestis and Poa.

Line 353: Using “suggest” here seems not fitting, rather use “show”.

Line 356 – 357: This seems to be the same information than in the beginning of the paragraph just in other words. Consider to delete this?

Line 358-359: Instead of “relationship” it would be better to use “effect” here. Also, not sure what you want to say here.

Line 361: “though” may be the wrong word here.

Line 362: Something is missing in this sentence, most ….?

Line 365: There is no S4 figure in the supplement.

Line 370: Do you mean increase or decrease? In the previous part of the sentence you write about a negative effect of clipping, so I assume it should be decrease. Again, it is confusing that you write clipping was not statistically significant but Fig.2 shows significant asterisks for the clipping treatment.

Line 381ff: You describe panel a) but not panel b) in your figure. The meaning of “…than clipping” is unclear, could be removed?

Discussion

Line 392: In the results you say that clipping has NO significant effect!

Line 394: Insert “increasing temperature” here, because OTCs are used to study effects of warming and not just temperature.

Line 404: enhanced

Line 405ff: Try to focus only on the warming treatment and not suddenly list discussion points related to the clipping treatment, since you will have a separate paragraph for it later. I suggest moving all discussions relevant to clipping treatment to the appropriate paragraph.

Line 435 ff: I suggest moving all information related to warming to the previous paragraph. Otherwise, it's confusing to read about the ethnobotanical survey and then jump back to warming.

Line 458: “… thus allowing natural selection to favor herbivores” does not sound right.

Line 463 – 464: The sentence starting with “therefore” is unclear.

Line 468: For consistency please decide whether you want to use AGB, aboveground biomass or biomass throughout the manuscript. If you use too many terms for the same variable, it becomes very confusing.

Line 477 -479: Sentence unclear.

Line 479 – 481: The reasoning here is unclear and not convincing. Your interaction shows that the positive warming effect is strongly reduced by clipping, so in my opinion you should discuss why grazing overrides the effect of warming.

Line 485: Figure 2 does not show the information you are referring to. Figure S2 shows species specific responses, but it is unclear whether these are in response to the combination of warming and clipping. Please, clarify both the figure and the text!

Line 501: What do you mean with indirect warming effects. Please explain in the text.

Line 503 – 505: Sentence unclear.

Line 511: Point missing before “Finally”.

References

Reference 29 and 31 are the same

Reference 75 is not related to your topic and does not support the point of the sentence in which the reference is cited.

Sometimes the references are reported in different styles e.g. number 54 includes editor names, number 74 has these lines | between names. Please check that the style of the references is consistent!

Figures

* Figure 1 Has improved, but there is still a confusing sense of scale, as the areas labeled "1m" are almost as large as the areas labeled 2.5m. Also was the species composition conducted in a hexagonal area or a square? Right now it looks inconsistent between the two treatments. What about the biomass extraction area, which is a weird shape? Where was the biomass extraction for the control and clipping-only plots?

Figure 2 Please, put the correct amount of * to each treatment. For now, the model output, the text and the figure are not consistent regarding the significance of the treatments.

Figure S2 The first graphs (a and b) are the same as Fig.2. Consider to delete these, because they take unnecessary space. Furthermore, it is unclear in the species-specific graphs whether these show responses to only one of the treatments or if the data are averages across all treatments.

Figure S3 This does not have a clear message. Consider deleting it.

Figure S4 Is missing, but referred to in the text.

Tables

Table 1 Good, but pay attention to consistent use of upper- and lower-case letters in the beginning of each word.

Table S2 Good, though in the caption “specie” is supposed to be “species”.

Table S4 Please see comment to line 240.

7. PLOS authors have the option to publish the peer review history of their article (what does this mean?). If published, this will include your full peer review and any attached files.

Reviewer #1: No

---

## [Author Response · Author response to Decision Letter 2]

24 Mar 2021

Authors Response to Reviewers 

Journal Requirements:

Please review your reference list to ensure that it is complete and correct. If you have cited papers that have been retracted, please include the rationale for doing so in the manuscript text or remove these references and replace them with relevant current references. Any changes to the reference list should be mentioned in the rebuttal letter that accompanies your revised manuscript. If you need to cite a retracted article, indicate the article’s retracted status in the References list and include a citation and full reference for the retraction notice.

Response: Reference list is updated 

Additional Editor Comments (if provided):

Dear Authors,

There are still some minor issues in statistical testing and language. Please address reviewer is all comment adequately and resubmit.

Reviewer #1: General comment

The manuscript has hugely improved and is much easier to read and understand. There are still some issues of confusion with the statistics, including the "Factor x species" model on line 240, use of t-test to assess the significances of the treatments, and understanding/reporting/interpretation of the interactive effect.

Response: Factor x species: The confusion in statistics is removed. Following details are mentioned. Please see line 243-254 in M&M. 

Here the factors are the treatments, (warming, clipping and warming x clipping). The model formulation was to check the individual response or species-specific response to each treatment. The summary statistics are shown in S4Table. (Please see supplementary information S4Table)

To estimate the fixed effects of the treatments, we used linear mixed effect models using lme4 package (Bates, Mächler, Bolker, & Walker, 2015) in R (The R Foundation, 2018). The model summary provides the estimation, standard error, and t values of each of the factors as shown in Table 2. It is shown that inferences for the fixed-effects parameters in linear mixed models are based on t-value distributions with suitably adjusted degrees of freedom. In mixed effect models, the main rationale is that while p-values can be misleading because of unclear null distribution, t values can still be useful as standardized parameters. The t-test used to evaluate the significance of overall treatments, the variance between the treatments and the response of each species to the specific treatment.

Reviewer comment: Interactive effects of warming and clipping treatments in OTC x clipping plot.

 Response: 

The overall interactive effects of both treatments have been described with more clarification. The interpretation is improved, and cross checked with the model tables and figures. Please see line 368-371, Table 2 and Figure 3. Details are as follows.

 The mixed effect model showed that the interactive effects of warming and clipping were comparatively weaker than the main effects, both on aboveground biomass and percentage cover. Whereas clipping has a consistent negative effect in main effect plots, it decreased the aboveground biomass of the species in interactive plots too. Thus, concluded that the negative clipping treatment effect decreased the cover and percentage also warming effect was not significant in these interactive plots. 

Also Please see section Discussion line 483

To better place our results in this context we explained the interactive effects as stronger clipping effect being an important finding than warming. The relative effects of warming and clipping on vegetation growth are not fully understood. The warming positive effect was not consistent in the interaction plot explaining the plants' sensitivity to combined stress.

Reviewer comment: Furthermore, the authors present results for the different sites for which the description of the analysis and model output tables are missing. Specific comments marked with * are particularly important.

Response: The results are not described for separate sites, but we consider effect of site as random. The model output table, where each factor (treatment, site, and species) considered as fixed effect was removed as suggested by previous reviewer. It has been now added to the supplementary file as S3Table b. Please see supplementary information S3Table b.

Reviewer comment There is still some language that is confusing or unclear in wording, especially in the results and discussion. Also, please check for upper- and lower-case letters throughout your figures and tables and chose a consistent style. Consistency is also needed concerning the terminology of the clipping/grazing treatment and aboveground biomass.

Response: Needful done. The terminology of clipping is used throughout the manuscript for consistency and careful revision of the manuscript is done for upper- and lower-case letters in figures and tables.

Reviewer comment: The authors state that "Yes - all data are fully available without restriction" and "All relevant data are within the manuscript and its Supporting Information files." but we see NO raw data in the supporting information file, just the supplemental tables with output. We hope this is an accidental error, but it needs to be corrected to meet the journal data requirements.

Response: Needful done. Data files of all relevant underlying data point is available as supplementary excel file.

Running title and abstract

Reviewer Comment: Line 13: Lowercase “H” in herbivory

Response: Needful done Please see line 14

Reviewer Comment: Line 22-23: This sentence may be easier to understand and read if you wrote: “Experimental warming increased biomass and percent cover throughout the experiment”. It is not necessary to report the p-value or that you used sites as a random factor. The abstract should highlight your main findings in a concise form, so avoid any unnecessary detail. 

Response: Needful done. P value are removed. Sentence is added as above. Please see line 22 in Abstract section.

Reviewer Comment: Line 27: Unclear what “due to its abundance at the site” means.

Response: Poa alpina was one of the dominant species at all experimental sites but we pooled the data of percentage cover for 3 years and represented the overall increase/decrease. The words “due to its abundance at the site” have been removed. Please see line 26 abstract section

Reviewer Comment: Line 28: Had instead of “has”.

Response: Needful done. Please see line 26

Reviewer Comment: Line 29: For consistency, it would be good to give the same names to your treatments. Clipping and simulated grazing are the same, right? Can you change this?

Response: Thank you for the suggestion. Clipping is now the consistent term used throughout the manuscript. 

Reviewer Comment: Line 31: Stronger effects, what does this mean? Please, report whether this refers to an increase or decrease.

Response: Needful done. The stronger effects of clipping refer to the decrease of percentage cover and biomass. Please see line 29-30

Introduction

Reviewer’s comment: Line 58-59: Sentence unclear, maybe the “to” needs to be replaced with “for”?

Response: Needful done. Please see line 56 in introduction section

Reviewer’s comment: Line 60: Delete the “s” at the end of “causes”.

Response: Needful done

Reviewer’s comment: Line 83: underrepresented?

Response: Needful done. Changed from underrepresented to marginalized. Please see line 81

Reviewer’s comment: Line 84-85: Sentence unclear.

Response: The sentence has improved as. “The incorporation of indigenous knowledge of medicinal plants with the current biological conservational practices is necessary to assess the future face of biodiversity and associated threats”. Please see line 82-84 

Reviewer’s comment: Line 101: This is the first time your mention KNP, please provide an explanation of the abbreviation.

Response: Needful done. Please see line 100 

Methods:

Reviewer’s comment: Line 117: No brackets needed when you mention the altitude.

Response: Needful done. Please see line 117

Reviewer’s comment: Line 126: Parking area??

Response: corrected as Park area. Please see line 126

Reviewer’s comment: Line 132: Comma missing between lanceolata and Saxifraga.

Response: Needful done. Please see line 132

Reviewer’s comment: Line 139: I think you mean individuals instead of species here.

Response: Needful done. Please see line 139

Reviewer’s comment: Line 190: Here you write that your clipping plot inside the fence is 2.5 m2, but from Figure 1 it looks like it is 1m2. Please clarify this.

Response: The subplot designated to the clipping treatment is 2.5m x 2.5m. By 1m x 1m dimensions means that we clipped the biomass from 1m x 1m area by quadrat method. Please see line 211 in data sampling section of material and methods

Reviewer comment: Line 193: Could you add the clipped area inside the OTC to Figure 1?

Response: Needful done. The area where species composition is mentioned was clipped and shown in Figure 1. Please see the Figure 1

Reviewer comment: Line 209: Lowercase “a” in “all”.

Response: Needful done. Please see line 209

Reviewer’s comment: Line 212: As pointed out in an earlier review, this reference seems not appropriate for plant metrics, but focusses on ground dwelling insects in riverbeds. You could cite for example Shaver and Chapin 1991: Production: biomass relationships and element cycling in contrasting arctic vegetation types, Ecological Monographs.

Response: Needful done. The updated reference is added as follows and updated in text too as[56]: Please see reference line 717 and line 212 in methods

Sanaei A, Ali A, Ahmadaali K, Jahantab E. Generalized and species-specific prediction models for aboveground biomass in semi-steppe rangelands. J Plant Ecol. 2019;12: 428–437. doi:10.1093/jpe/rty037

Reviewer’s comment: Line 223-224: What do you mean with “We clipped the vegetation from OTC”? This sentence needs clarification.

Response: Needful done. We measured aboveground biomass of each individual species in each treatment and OTC too by clipping the whole species using quadrat method, sort it out separately and weighed in laboratory by air drying at 37°C for 72 hours. Please see line 224

Reviewer’s comment: * Line 240: The model formulation for this second, species-specific model is not clear. What is "Factor" in this model? I think what you are intending to write is Factor * species + (1|site)?

*Response: Factor in the model is the treatment [warming, clipping and warm*clip (interactive treatment)]. We described the species-specific response to each treatment. (Please see S4Table). The model formulation is updated as follows. Perc cover ~Factor*species: (1|site). Please see line 243

Reviewer’s comment: Line 241 -244: Sentence unclear.

Response: The sentence is re-structured as” The high ethnomedicinal value (Table 1) and the response to the treatments illustrated by mixed effect models, the vulnerability and susceptibility of important plant species was determined. These species included Artemisia rupetris, Poa alpina, Primula macrophylla, Potentilla hololeuca and, Astragulus penduncularis. The combined effect of treatments on these species was illustrated as susceptibility zones to show the positive and negative effects of both the treatments and analyzed the occurrence of constitutive species. Please see line 256-263

Reviewer’s comment: *Line 244: “polygons- based susceptibility zones” unclear. If this is an established method, you should provide references.

Response: It is graphical illustration of the results obtained by linear mixed effect models (S4 Table). From the results the significance of interactive treatment effect was shown in figure 4 as susceptibility of the species to the combined effect of each treatment. The clipping and warming treatment effect was drawn as four susceptibility zones whose combined effect would increase or decrease the cover and biomass of highly important medicinal plant species. Please see line 256-263

Reviewer’s comment: * Line 246: You should add here which kind if test you used to assess the significances of the treatments, and what function or package you used for that test.

Response: Needful done. The given information is added to the text. The significances of treatments were assessed by linear mixed effect models for random (site, species) and fixed (treatments) effects using lme4 package, function lme(). The plot was made by ggplot2, function geom_polygon, package tidyverse. Please see above response also line 261-263

Reviewer’s comment: Line 250: Point missing between “groups” and “the”.

Response: Needful done. Please see line 255

Results

Reviewer’s comment: Line 267: Percentage for Asteraceae is missing.

Response: Needful done. Please see line 284

Reviewer’s comment: Line 295: You state that clipping is not significant, but in Figure 2a there are three asterisks to indicate high significance. Same for the interaction which is not significant according to Table2. Also based on Table 2 there should be two asterisks for the interaction the Figure 2b and only one for the clipping treatment.

Response: Needful done. The text and figure are updated according to the significance mentioned in Table 2. Please see Figure 2-part b and text line 310-316. The further details are given below:

Contrastingly clipping exerted significant negative effect on biomass and non-significant negative effect on percentage cover. Plant species decreased their aboveground biomass in clipping plots while there was a significant increase in biomass in warming plots. This positive effect was overridden by the clipping effect in interaction plots where the combined impact of both warming and clipping was found to be significantly negative on aboveground biomass (p<0.05) (Table 2).

Reviewer’s comment: Line 307: Why two models? What you show in the table can be derived from one model which contains both the main effects and the interaction.

Response: The sentence is structured as follows: 

Mixed effect model summary showing estimate, standard error, and t statistics is described in Table 2 explaining the significant fixed effects of experimental warming, clipping and the interaction treatment (Warm*Clip) on plant species. Please see line 320-322

Reviewer’s comment: Line 314 – 316: Species is missing an “s”, the sentence is unclear.

Response: Needful done. Species specific response to each treatment while keeping the effect of the site as random showed that elevated warming has significantly increased the percentage cover and aboveground biomass as shown in Figure 2, S4Table. Please see line 325-327

Reviewer’s comment: Line 318 -320: This sentence could be moved to the results but is not needed to describe the figure. You already do that in a) and b).

Response: This sentence is removed. Please see line 329-330

Reviewer’s comment: Line 319: Write the full name of the treatment, not just “warm” and “clip”. It is always good to be consistent in the use of treatment names/terminology!

Response: Needful done. This sentence is removed, please see the comment above. Moreover, the whole manuscript is revised for the warming and clipping term consistency.

Line 319 – 320: If clipping had a consistent negative effect you would see this as a main effect. Here you need to think about why the combination of warming and clipping decreases percentage cover.

Response: Our study reported consistent positive warming and negative clipping effects while in the interaction treatment plots clipping effect dampened the positive effects of warming because clipping usually counteracted the warming effect. Clipping has stronger effects than the warming also the alpine vegetation responses to combined stress suggested that clipping(grazing) is responsible driver for the vegetative productivity in the region. Warming enhanced the percentage cover but in this plot plant species are under the pressure of two main treatments, so consistent negative impact if clipping here has outcompete the positive effects on vegetation. It suggested it is difficult to predict the combined effects of change from single factor studies. This is later explained and discussed in results and discussion session. Please see line 482. 

Reviewer’s comment: Line 322: See the above comment. In general, you do not need to add interpretations of the results to the figure caption, reserve this for the discussion.

Response: Needful done. Please see line 331

Reviewer’s comment: * Line 326: You did not mention in your statistical analysis that you assessed differences in percentage cover and biomass between the different sites. Please add this information. Furthermore, you should also provide a table with the model output. Referring to the figure only is not sufficient because the reader cannot judge based on a figure whether something is significant or not.

Response: Needful done. In statistical analysis, factor effect plot analysis is mentioned. Please see line 228.

The factor effect plots of treatment site and species were added with the model output statistical summary in early draft. But because we took site as the random effect, reviewer has asked to remove these table. The individual effect of each factor was determined (S2Figure) to know the significance of each treatment on cover and biomass keeping each factor as fixed in mixed effect model. The model output is added as S3Table b, c&d where the significance of each factor was determined on percent cover and biomass. Please see supplementary information S3Table.

Reviewer’s comment: Line 330 – 333: Sentence unclear and confusing.

Response: We made the factor effect plots for each variable (site, species, and treatment). The main effect was of the treatments and site and species were taken as random effects. But we individually determined the effect of each factor as well (S2Figure). The response variability was high among the highest elevation site (M1) and the lowest (M5) suggesting that site may be a determining factor for difference in response magnitude of the species. Changes in the species composition along the elevational gradient as random effect can be described as predictor about the variance in response between different sites. Please see lines 337-340

Reviewer’s comment: Line 336 – 337: Higher biomass is a result of warming, but warming is not determined by higher biomass. Please, rephrase this.

Response: Needful done. The overall differences in response magnitude of all species' percentage cover and biomass can be seen in species effect plot where some plant species as Poa alpina (PA), Astragulus penduncularis (AS), Artemisia rupestris (AR), and Potentilla hololuca (PT) had higher percent cover and biomass in comparison to others (S3Table a, S2Fig). Please see line 344-348

Reviewers comment: Line 337: Figure S2 species-specific plots, are these plots in response to warming or in response to clipping or both? What does this figure add in comparison to figure 3?

Response: The effect of each factor was determined before selecting the fixed and rando factors in developing mixed effect models. Figure S2 explains the overall factor effect considering the species as fixed factors while the Figure 3 explains the species-specific response and effect of individual treatment on each species. It was added as main figure before, but it is moved to supplementary information as per reviewer’s suggestion

Reviewer’s comment: Line 343: Comma missing between ruprestis and Poa.

Response: Needful done. Please see line 353

Reviewer’s comment: Line 353: Using “suggest” here seems not fitting, rather use “show”.

Response: Needful done. Please see line 369

Reviewer’s comment: Line 356 – 357: This seems to be the same information than in the beginning of the paragraph just in other words. Consider deleting this?

Response: Needful done. Please see line 371-374

Reviewer’s comment: Line 358-359: Instead of “relationship” it would be better to use “effect” here. Also, not sure what you want to say here.

Response: Needful done. The whole sentence is re-structured for clarification of interactive experiment result. Please see line 369-375

Mixed effect model showed that the interactive effects of both warming and clipping were comparatively weaker than the main effects, both on aboveground biomass and percentage cover. Whereas clipping has a consistent negative effect in main effect plots, it decreased the aboveground biomass of the species in interactive plots too. Our finding showed that response magnitude of percentage cover and aboveground biomass to clipping was stronger than the experimental warming in interaction (W x CL) plots (Figure 2, Table 2). It has decreased the aboveground biomass of species significantly (<0.01) as shown in Figure 2 and Table 2.

 Reviewer’s comment: Line 361: “though” may be the wrong word here.

Response: Needful done. Please see comment above

Reviewer’s comment: Line 362: Something is missing in this sentence, most ….?

Response: The whole paragraph is re-structured. Please see response to comment line 358-359

Reviewer’s comment: Line 365: There is no S4 figure in the supplement.

Response: The S4 Figure is now S3 Figure. Please see supplementary figures S3Figure

Reviewer’s comment: Line 370: Do you mean increase or decrease? In the previous part of the sentence, you write about a negative effect of clipping, so I assume it should be decrease. Again, it is confusing that you write clipping was not statistically significant, but Fig.2 shows significant asterisks for the clipping treatment.

Response: The overall effect of treatment and species-specific response is given in two figures figure 2 and figure 3. It is corrected as our finding showed that response magnitude of percentage cover and aboveground biomass to clipping was stronger than the experimental warming in interaction (W x CL) plots (Figure 2, Table 2). It has decreased the aboveground biomass of species significantly (<0.01) as shown in Figure 2 and Table 2. Please see lines 372-375

Reviewer’s comment: Line 381ff: You describe panel a) but not panel b) in your figure. The meaning of “…than clipping” is unclear, could be removed?

Response: Needful done. Please see lines 390-391

Discussion

Reviewer’s comment: Line 392: In the results you say that clipping has NO significant effect!

Response: It is corrected in the results section “clipping effect” too. 

The clipping effect on biomass was significantly negative (Table 2) it means that in main clipping plot and interaction plot the biomass was reduced. Please see line 460-462

Reviewers comment Line 394: Insert “increasing temperature” here, because OTCs are used to study effects of warming and not just temperature.

Response: Needful done Please see line 408

Reviewer comment: Line 404: enhanced

Response: Needful done Please see line 423

Reviewer comment: Line 405ff: Try to focus only on the warming treatment and not suddenly list discussion points related to the clipping treatment, since you will have a separate paragraph for it later. I suggest moving all discussions relevant to clipping treatment to the appropriate paragraph.

Response: All discussion points related to warming have been moved to experimental warming section of discussion. Please see lines 430

Reviewer comment Line 435 ff: I suggest moving all information related to warming to the previous paragraph. Otherwise, it's confusing to read about the ethnobotanical survey and then jump back to warming.

Response: Needful done. Please see comment above

Reviewer’s comment: Line 458: “… thus allowing natural selection to favor herbivores” does not sound right.

Response: removed. Please see line 472-475

Reviewer’s comment: Line 463 – 464: The sentence starting with “therefore” is unclear.

Response: It is rephrased as: “our results supported this assumption, where clipping treatments had significant negative effects on plant biomass and non-significant negative effects on percentage cover. The general trend was decline this decrease was highly variable among the taxa also the random effect of site can also be a predictor of variability in response[77]. Please see line 478-482

Line 468: For consistency please decide whether you want to use AGB, aboveground biomass or biomass throughout the manuscript. If you use too many terms for the same variable, it becomes very confusing.

Response: The unified term “aboveground biomass” is used consistently throughout the manuscript and any other confusing term has removed

Reviewers comment: Line 477 -479: Sentence unclear.

Response: Please see line 496-501

Line 479 – 481: The reasoning here is unclear and not convincing. Your interaction shows that the positive warming effect is strongly reduced by clipping, so in my opinion you should discuss why grazing overrides the effect of warming.

Response: Needful done. Please see line 496-512

The magnitude of positive responses of species as increased percentage cover and aboveground biomass was in the zones of only warming treatment(+W,-G), in absence of clipping (Fig 4), thus predicting the availability of these species as preferred forage if the grazing is controlled. [78]. In contrast, above ground biomass has significantly decreased in clipping and interactive plots and strongly reduced the positive impacts of warming. This outcome was most likely recognized to the following mechanisms. The relatively small effects of experimental warming than clipping in interactive plots on vegetation biomass and cover can be related to experimental warming induced soil drying. The plant species are under the stress of two climatic drivers. Thus, warming has negligible positive effects on vegetation in presence of clipping. Grazing is reported to have strong effects on C storage than warming it can significantly decrease leaf area and plant photosynthesis[78–80]. This mechanism explained the relatively large effects of clipping on vegetation biomass which overrides the warming effect in interaction plots. Again the species specific response of these taxa is shown in biomass susceptibility zone where some species as A.rupestris, and P.alpina has non-significantly increased the biomass. Thus, predicting future vegetation in alpine ecosystem requires the knowledge of grazing interaction with climate variables which can influence the dynamics of alpine ecosystem. 

Reviewer comment: Line 485: Figure 2 does not show the information you are referring to. Figure S2 shows species specific responses, but it is unclear whether these are in response to the combination of warming and clipping. Please, clarify both the figure and the text!

Response: Needful done. The figure 3b is added in place of figure 2 Please see line 514.

Reviewer comment Line 501: What do you mean with indirect warming effects. Please explain in the text.

Response: Please see lines 530-531. ….. may differ in their ability to measure the indirect impacts (human and environment interaction, health risks through rising air temperatures and heat waves) of climate warming but can provide a very rapid estimation of short-term direct impacts.

Reviewer comment Line 503 – 505: Sentence unclear.

Response: Needful done. Please see lines 532-534.The future prediction of alpine biodiversity of culturally important species depends highly on the optimal grazing practices in the current climate warming scenario in HKKH

Reviewer comment Line 511: Point missing before “Finally”.

Response: Needful done. See line 541

References

Reviewer comment: Reference 29 and 31 are the same.

Response: Needful done. The repetitive reference is removed.

Reviewer comment Reference 75 is not related to your topic and does not support the point of the sentence in which the reference is cited.

Response: Needful done. This reference is removed.

Reviewer comment Sometimes the references are reported in different styles e.g. number 54 includes editor names, number 74 has these lines | between names. Please check that the style of the references is consistent!

Response: Needful done 

Figures

Reviewer comment: * Figure 1 Has improved, but there is still a confusing sense of scale, as the areas labeled "1m" are almost as large as the areas labeled 2.5m. Also was the species composition conducted in a hexagonal area or a square? Right now it looks inconsistent between the two treatments. What about the biomass extraction area, which is a weird shape? Where was the biomass extraction for the control and clipping-only plots?

Response: The biomass extraction was done using random quadrat method inside the control and clipping plots after the percentage cover estimation. The plant saplings were marked individually after clipping to a certain height and individual biomass was calculated for each species. Form OTC this destructive sampling of biomass extraction was done along the slopping sides of the OTCs.

Reviewer comment: Figure 2 Please, put the correct amount of * to each treatment. For now, the model output, the text and the figure are not consistent regarding the significance of the treatments.

Response: Needful done

Reviewer comment: Figure S2 The first graphs (a and b) are the same as Fig.2. Consider to delete these, because they take unnecessary space. Furthermore, it is unclear in the species-specific graphs whether these show responses to only one of the treatments or if the data are averages across all treatments.

Response: It is worth to mention that this was main figure in the first draft of the paper. But the reviewers suggested us to put this figure into supplementary information. By this S2Figure, we intend to show the factor effect on biomass and percentage cover considering each variable as fixed effect and according to previous revision Figure S2 and Figure 2 describe our results in a best way. Though it is in supplementary information, we request to include this information in our manuscript as supplementary information.

Reviewer comment Figure S3 This does not have a clear message. Consider deleting it.

Response: The relative abundance of each plant species biomass in the warming and warm*clip treatment was estimated through Non-metric multidimensional scaling (NMDS) plot with 95% confidence intervals for over a 3–years experimental period along the elevation gradient of (3590–4696m). The relative abundance of plant species biomass inside the OTCs on each elevation site was estimated for all study years and shown by paths of mean values in Non-metric multi-dimensional scaling (NMDS) using Bray and Curtis dissimilarity index in R. The plot shows no significant difference between the relative abundance as there is no significant change in the relative abundance of species inside both OTCs.

Reviewer comment Figure S4 Is missing but referred to in the text.

Response: Needful done. The number of the figures are updated

Tables

Reviewer comment: Table 1 Good but pay attention to consistent use of upper- and lower-case letters in the beginning of each word.

Response: Needful done

Reviewer comment: Table S2 Good, though in the caption “specie” is supposed to be “species”.

Response: Corrected

Reviewer comment: Table S4 Please see comment to line 240.

Response: Factor in the model is the treatment [warming, clipping and warm*clip (interactive treatment)]. We described the species-specific response to each treatment. (Please see S4Table). The model formulation is updated as follows. Perc cover ~Factor*species: (1|site)

---

## [Editor Report · Decision Letter 3]

30 Mar 2021

The Response of Culturally Important Plants to Experimental Warming and Clipping in Pakistan Himalayas

PONE-D-20-21262R3

Dear Dr. Ali,

We’re pleased to inform you that your manuscript has been judged scientifically suitable for publication and will be formally accepted for publication once it meets all outstanding technical requirements.

Kind regards,

Abid Hussain

Academic Editor

PLOS ONE

Additional Editor Comments (optional):

Dear Authors,

Thank you for addressing the concerns adequately.
---

## [Editor Report · Acceptance letter]

22 Apr 2021

PONE-D-20-21262R3 

The Response of Culturally Important Plants to Experimental Warming and Clipping in Pakistan Himalayas  

Dear Dr. Ali:

I'm pleased to inform you that your manuscript has been deemed suitable for publication in PLOS ONE. Congratulations! Your manuscript is now with our production department. 

Kind regards, 

on behalf of

Dr. Abid Hussain 

Academic Editor

PLOS ONE